# Mobility enhancement in heavily doped semiconductors via electron cloaking

Jiawei Zhou [1✉], Hangtian Zhu [2], Qichen Song [1], Zhiwei Ding [1], Jun Mao[2], Zhifeng Ren[2] & Gang Chen [1✉]

Doping is central for solid-state devices from transistors to thermoelectric energy converters. The interaction between electrons and dopants plays a pivotal role in carrier transport. Conventional theory suggests that the Coulomb field of the ionized dopants limits the charge mobility at high carrier densities, and that either the atomic details of the dopants are unimportant or the mobility can only be further degraded, while experimental results often show that dopant choice affects mobility. In practice, the selection of dopants is still mostly a trial-and-error process. Here we demonstrate, via first-principles simulation and comparison with experiments, that a large short-range perturbation created by selected dopants can in fact counteract the long-range Coulomb field, leading to electron transport that is nearly immune to the presence of dopants. Such "cloaking" of dopants leads to enhanced mobilities at high carrier concentrations close to the intrinsic electron–phonon scattering limit. We show that the ionic radius can be used to guide dopant selection in order to achieve such an electron-cloaking effect. Our finding provides guidance to the selection of dopants for solid-state conductors to achieve high mobility for electronic, photonic, and energy conversion applications.

[1] Department of Mechanical Engineering, Massachusetts Institute of Technology, Cambridge, MA 02139, USA. [2] Department of Physics and Texas Center for Superconductivity at the University of Houston (TcSUH), University of Houston, Houston, TX 77204, USA. ✉email: jwzhou@stanford.edu; gchen2@mit.edu

D oping is a fundamental strategy employed to control the electrical conductivity of semiconductors[1]. Conventional understanding based on the theory originally proposed by Brooks and Herring[2,3] states that electrons are strongly scattered by the long-range Coulomb field of the charged dopant, leading to reduced mobility. With further generalization to consider multiple scatterings, electron–electron interactions, and dielectric screening[4–6], the Brooks–Herring theory has been successfully used to explain the reduced charge mobility in conventional semiconductors such as silicon and III–V semiconductors at low to intermediate doping concentrations. An important consequence of this theory is that different dopants with the same charge have the same impact on the electron transport, as the theory neglects the atomic details[4]. While the effect of a dopant's chemical nature has been recognized in the past, most models have assumed an empirical potential profile and treated such a chemical effect as a perturbation to the Coulomb field, as indicated by the "central cell correction"[7–9]. A prevailing view is that such correction could only further reduce charge mobility due to the strong local interactions between electrons and defects. Experimentally, different impacts on carrier mobility resulting from different dopants are often observed[7,10,11], suggesting that their atomic details play a significant role in governing the electron-defect interactions (EDIs). The impact of dopant scattering on mobility can be particularly large at high carrier concentrations, as is often observed in solid oxide materials[12], transparent conductors[13], and thermoelectric compounds[14], with a carrier concentration close to or above ~$10^{20}$ cm$^{-3}$. Despite the crucial role of dopants in governing the carrier transport, dopant selection has thus far been mostly a trial-and-error process.

Recent advancements in ab initio simulations have enabled quantification of electron energy or potential changes induced by charged defects, allowing the engineering of defects from first principles[15–17]. These studies have revealed that a charged defect potential can significantly deviate from the conventional Coulomb field assumption[15–17]. However, how such atomic details impact charge mobility remains unknown, mainly due to the lack of capability to treat the long-range Coulomb field and short-range electron interactions on an equal footing. Here we employ a computational approach to treat Coulomb and short-range interactions simultaneously and apply it to heavily doped semiconductors. Building on the recent development in the formation energy calculations for charged defects[18], we take into account short-range perturbations from ab initio calculations while incorporating the long-range potential to far distances via analytic expression. Our approach overcomes the obstacles that have previously prevented direct quantification of electron interactions with charged defects, allowing us to examine the impact of short-range interactions on electron transport. We discovered that the deviation of defect potential from the Coulomb field at short range can lead to strong EDIs, particularly at high carrier concentrations. We further demonstrate, in contrast to the conventional belief that charge mobility in extrinsic semiconductors is limited by Coulomb scatterings, that the chemical nature of dopants can be harnessed to break this barrier, leading to effective electron cloaking and enhanced mobility close to the intrinsic electron–phonon scattering limit. While modeling suggested that a core-shell nanoparticle with proper potential can be cloaked[19], we found here that the central cell effect of selected dopants allows them to actually cloak themselves, achieving electron cloaking via doping. We show that ionic radius can guide the choice of proper dopants to realize the cloaking effect, hence providing direction in the dopant selection to achieve high mobility.

## Results

**Electron-defect interaction.** Central to the EDIs is the perturbed electronic potential $\triangle \widehat{V}$ resulting from the presence of defects. Representative defect potential profiles are shown in Fig. 1a, in which the potential at longer distances can be approximated by the Coulomb field of a point charge (here we use a representative n-type dopant, which gives rise to an attractive Coulomb potential, as an example), while deviations occur close to the defect (represented by a simplified rectangular profile). Two scenarios, corresponding to attractive and repulsive short-range potentials, are depicted. Such short-range deviations are traditionally treated using empirical potentials, known as central cell correction[4,7–9], to investigate their effect on electron dynamics. It has been found that the dominant EDIs are due to the long-range Coulomb field and the atomic details mostly introduce perturbations to the electron binding energy or scattering rates[4,7–9], which only further degrades the mobility. The central cell effects are usually weak, and have not been harnessed to help achieve high electron mobility.

However, if a large central cell potential exists and has a sign opposite to that of the Coulomb potential, it could counteract the scatterings due to the Coulomb field and enhance electron mobility. The reason is that the electron defect scattering is governed by the magnitude of the EDI matrix (to the first order), given by the spatial integration of the defect potential – $\langle \psi_{\mathbf{k}'} | \triangle \widehat{V} | \psi_{\mathbf{k}} \rangle$, where $\psi_{\mathbf{k}}$ is the electronic wavefunction with wave vector $\mathbf{k}$. Both central cell part and long-range Coulomb part of the defect potential $\triangle \widehat{V}$ contribute to the EDI matrix. When the central cell potential has an opposite sign to that of the Coulomb potential, it will lead to canceling terms in the above spatial integration, and correspondingly a small EDI matrix, which implies weak or negligible electron-defect scattering. Note that some of past literature calculated the scattering rate due to the central cell and the long-range Coulomb contribution separately, and added the two rates according to Matthiessen's rule. Such a treatment neglects the coherence effect of electron waves. In this treatment, the inclusion of the central cell potential always leads to a higher scattering rate. Conceptually, this is not the correct treatment since it is the net effect from the central cell and the long-range Coulomb interaction that impacts the electron scattering.

It is then clear that the sign of the central cell potential relative to that of Coulomb potential plays a large role in how charged defects scatter electrons. Here we show that the ionic radii of the dopant and host atoms can be used to indicate the sign of the central cell potential (Fig. 1a). The ionic radius here refers to the size of the ion without valence electrons. For example, a silicon atom can be considered as a silicon ion with +4 charge and four electrons, which we denote as Si$^{+4}$[2s2p]. Similarly, a phosphorus atom can be denoted as P$^{+5}$[2s3p]. The reason we separate the valence electrons from the core is that valence electrons participate in chemical bonding and will be more strongly affected by the lattice structure, while the core electrons are more localized. When silicon is doped with phosphorus (P), phosphorus atom will lose one electron (which becomes free) and becomes positively charged. The remaining four electrons in the outer shell of P atom would participate in the chemical bonding, just as the valence electrons of Si would do, except that the core potential is different from that of Si. The defect potential (perturbed potential due to replacement of one Si by one P) thus results from the difference in the core potential (P$^{+5}$ compared to Si$^{+4}$).

The first major difference between the core potential of P$^{+5}$ and Si$^{+4}$ is that their charges differ by one, which lead to a long-range Coulomb field when Si$^{+4}$ is replaced by P$^{+5}$. In addition, as

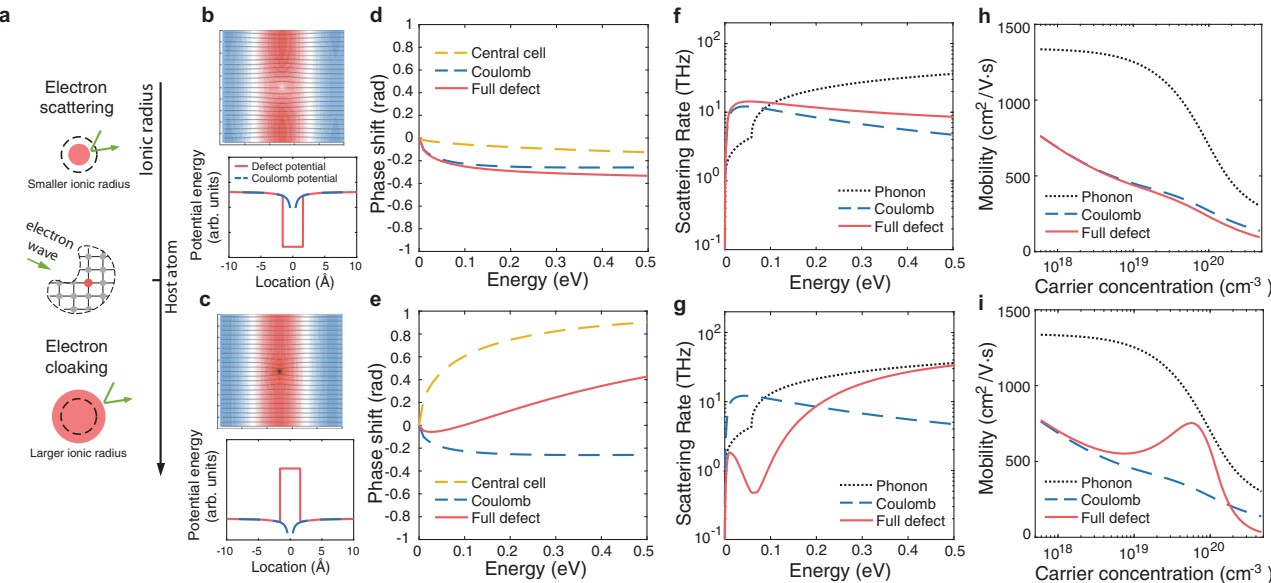

**Fig. 1 Impact of charged defects on electron transport. a** Illustration of a propagating electron wave scattered by the perturbed electronic potential of an n-type dopant. The defect potential contains two parts, a long-range part due to the attractive Coulomb field, and a short-range part that depends on the bonding environment of the defect (the central cell potential). The ionic radius can be used as a good indicator for the sign of the central cell potential. Depending on the ionic radius of the dopant atom, one of two general scenarios can play out. If the dopant atom has a smaller ionic radius than the host atom, the perturbation tends to create an additional attractive force for electrons (electron-scattering scenario); if the dopant atom has a larger ionic radius, it tends to create a repulsive force which then opposes the long-range Coulomb potential (electron-cloaking scenario). **b, c** Defect potentials and scattered electron wavefunctions corresponding to two different scenarios (**b**: electron scattering, **c**: electron cloaking). The central cell potential is represented by a simplified rectangular profile. The streamlines show the probability flux and the colors indicate the real part of the wavefunctions. In the case of electron cloaking (**c**), unperturbed streamlines are recovered away from the defect (located at the center). The domain size is 158.8 Å × 158.8 Å and the electron energy is 0.1 eV. **d, e** Phase shifts of scattered wavefunctions with angular momentum quantum number of $l = 1$ for two scenarios (**d**: electron scattering, **e**: electron cloaking). **f, g** Modeled intrinsic electron scattering rates due to electron–phonon interactions, in comparison to those considering electron-defect scatterings (**f**: electron scattering, **g**: electron cloaking). The calculation assumes a parabolic band to illustrate the general impact of defect scatterings and uses partial wave analysis to calculate the scattering rates due to defects (see details in Methods). **h, i** Electron mobility with respect to the carrier concentration (**h**: electron scattering, **i**: electron cloaking). While mobility at high carrier concentrations is traditionally believed to be limited by Coulomb scattering, a strong opposing central cell potential can be harnessed to break this limit, leading to high mobility limited only by the intrinsic electron–phonon interactions, as shown in (**i**).

the electron approaches close to the center, it will experience difference in its interactions with the core electrons as the latter now are being held by a different atom. Such interactions involve both Coulomb interactions and exchange–correlation interactions, the latter of which are inherent to the many-body electron system and a cause of this is Pauli exclusion. On the one hand, the Coulomb interactions between the nucleus and electrons will cause the core electrons to contract or expand depending on the atomic number of the dopant relative to the host, which would lead to variations in the core potential. In the case of P-doped Si, $P^{+5}$ has a higher proton number than $Si^{+4}$, which leads to a slight contraction of the core electrons. This core electron contraction (and correspondingly higher core electron density) will tend to give a slight positive defect potential near the center, as observed in Supplementary Fig. 3 (inset). On the other hand, Pauli exclusion (or exchange–correlation interactions) forces the valence electrons to stay outside the core region (in the sense that the valence electron wavefunctions are orthogonal to core electrons). Therefore, when the dopant ion has a smaller size than the host, propagating electron states (which are composed of mainly valence electrons) will have deeper penetration into the core, which is equivalent to creating an attractive short-range defect potential. For P-doped Si, because $P^{+5}$ has a smaller ionic size than $Si^{+4}$, the defect potential becomes negative slightly away from the center (Supplementary Fig. 3, inset).

The above-mentioned effects lead to different trends of defect potential profile in the periodic table. The variation of core

electron density mostly depends on the atomic number, while the Pauli exclusion effect is largely determined by the ionic size. While both will affect the defect potential, here we argue that the short-range core potential is better described by the ionic size difference between the dopant and host atom. This is because within and around the core region, the exchange–correlation interaction can be significant. When discussing the defect potential, especially the sign of its short-range part, we think Pauli exclusion is a more important governing factor, which is reasonably captured by the ionic radius picture to the first order. In general, when the dopant ion has a smaller size than the host, the defect potential tends to be dominantly negative (attractive potential to allow electrons to penetrate deeper into the core, Fig. 1b). Conversely, if a dopant ion has a larger size than the host, it will tend to repel electrons due to Pauli exclusion and tend to be dominantly positive (repulsive potential, Fig. 1c). We will later show that in most cases our simulation results agree better with this ionic size picture. We therefore will use the ionic radius as the major indicator to infer the sign of short-range defect potential.

Now we consider short-range and long-range parts together for the defect potential. For n-type dopant, the long-range Coulomb potential is attractive. A dopant with smaller ionic radius than the host would tend to create a short-range attractive potential, which adds to the Coulomb part and further increases EDI strength, leading to stronger electron-defect scatterings (electron scattering scenario). In contrast, dopants with larger ionic radius will tend

to create repulsive short-range potential, which counteracts the Coulomb potential. When the cancellation effect is maximal, the defects will appear to have negligible scatterings for electrons, and this realizes electron cloaking (electron cloaking scenario). As seen from the computed scattered electron wavefunction, the probability flux is distorted for the electron scattering scenario (Fig. 1b), while undistorted probability flux is recovered away from the defect for the electron cloaking scenario (Fig. 1c). In short, to achieve electron cloaking effect, dopants with larger ionic radius are favored for n-type materials. In addition to the electron cloaking, the ionic radius provides a useful guidance for understanding and selecting dopants in terms of the electron mobility. Effectively, the ionic radius can be seen as a scale bar for dopant selection: in n-type materials, dopants with larger ionic radius are more desired than those with smaller ionic radius because they are more repulsive to electrons and tend to counteract the Coulomb scatterings (Fig. 1a).

The above argument can also be applied to p-type materials, except that the Coulomb potential now becomes repulsive to electrons. As a result, the desired dopants that counteract the Coulomb scattering need to have attractive short-range potentials. The relation between ionic radius and short-range potentials remains the same. Therefore, dopants with smaller ionic radius are more desired for p-type materials (Supplementary Fig. 1).

To illustrate the consequence of different defect potentials on electron transport, we first study a model semiconductor as an example (see modeling details in Methods). Shown in Fig. 1d, e are scattered electron phase shifts computed for a n-type model semiconductor with different defect potentials. Phase shifts quantify how scattered electron waves differ from the incoming waves, and their magnitudes represent the strength of electron-defect scatterings[20]. In general, attractive potentials lead to negative phase shifts, while repulsive potentials lead to positive phase shifts. When attractive short-range potential coexists with the attractive long-range Coulomb potential in n-type materials, the phase shifts add up in magnitude (Fig. 1d), leading to increased electron scattering rates and reduced mobility (Fig. 1f, h). In contrast, if the short-range potential is repulsive, it counteracts the attractive long-range Coulomb field and reduces the overall phase shifts (Fig. 1e), decreasing the total electron-defect scattering rates (Fig. 1g). When the defect scatterings become weaker than the intrinsic electron–phonon scatterings, the charge mobility becomes nearly immune to the presence of defects and high mobility can be achieved (Fig. 1i).

**Electron scattering due to atomic distortions**. Having discussed the effects of different dopants on mobility in the model semiconductor, now we turn to practical materials. Below we will first present simulation results and comparison with experiments corresponding to the electron scattering scenario, and then discuss possible evidences that demonstrate the electron cloaking effect. First, we note that mobility variations due to different dopants become significant when the electron-dopant interactions from the central cell potential are strong and comparable to those from the Coulomb field. To find out the parameter space where dopant selection is more critical, Fig. 2a displays the ratio of the characteristic electron scattering rates due to central cell scattering and Coulomb scattering in the model semiconductor. In general, the central cell scattering becomes stronger for materials with larger dielectric constant and at higher carrier concentrations. This is because with larger dielectric constant the magnitude of Coulomb potential becomes relatively smaller and at higher carrier concentrations the Coulomb potential is weakened via screening. Conventional semiconductors usually have low to intermediate doping concentrations and small dielectric constant, which together indicate the dominance of the Coulomb potential in governing their

electron scatterings and the relative unimportance of the dopant selection (Fig. 2a). In contrast, materials with a high carrier density, e.g., thermoelectrics[14] and solid oxide conductors[12], are likely to be more sensitive to the central cell potential due to their large carrier concentrations. Among them, many oxide and thermoelectric materials also possess a high dielectric constant associated with soft phonon modes (see examples in Supplementary Table 1). Thus, for these materials, the central cell effect could be significant and thereby potentially be harnessed to counteract the Coulomb scatterings and break the conventional limit in charge mobility at high carrier densities.

Chalcogenide compounds are a class of materials that have received wide attention due to their potential for optoelectronic, photovoltaic, and thermoelectric applications. In particular, the pursuit of high thermoelectric energy conversion efficiency has driven studies of heavily doped chalcogenide semiconductors[14]. Experiments have shown that different dopants with the same ionization charge can lead to drastically different mobility values, suggesting strong central cell effects[11]. The significance of the central cell effect can be described by the short-range electron-defect interaction (sEDI) matrix, $\langle \psi_{\mathbf{k}'} | \triangle \hat{V}_{\text{cent}} | \psi_{\mathbf{k}} \rangle$, where $\triangle \hat{V}_{\text{cent}}$ is the central cell part of the defect potential. For comparison, we computed the defect potential for both Si-doped GaAs, a conventional semiconductor, and Bi-doped PbTe, a representative chalcogenide material. Significant deviations from the Coulomb potential are seen at short range in both cases (Supplementary Fig. 2). We note that the planar averaged defect potentials generally vary around 0.1–1 eV at the short range, on the same order of magnitude with previous reports[16,21]. Nonetheless, because of the strong covalent bond between Ga and As atoms and the larger spread of the charge density, the sEDI matrix in GaAs is generally small, as shown in Fig. 2b, which displays a three-dimensional contour color map of the sEDI matrix in the real space, with the shape exhibiting the hybridized $s$-$p$ orbital feature of the electron state. In comparison, PbTe consists of resonantly bonded $p$ orbitals, which are sensitive to the defect-induced local distortion that breaks the crystal symmetry. As a result, the sEDI matrix is significantly larger (Fig. 2c).

In order to evaluate the electron transport based on EDI, we need to account for the long-range Coulomb potential, which extends beyond the finite supercell calculations. The key observation that enabled our first-principles computation is that the long-range part of the defect potential can be well described by an analytic Coulomb potential profile (Supplementary Note and Supplementary Fig. 3, where the trend of short-range defect potentials of dopants in silicon are also found to agree with the ionic radius picture). Therefore, we are able to express the defect potential to far distances and to treat short-range and long-range potentials on an equal footing (see details in Methods). To validate our approach, Fig. 2d, e shows comparisons between calculations of electron mobility and experimental results for GaAs and PbTe, respectively, at different carrier densities. For each material, two dopants at different atomic sites are considered. In GaAs, we observed that Si doping reduces the mobility slightly more than Te doping does (Fig. 2d). The dopant effects are generally small and the major scattering is due to the Coulomb potential, consistent with past electron transport studies on conventional semiconductors[10]. On the other hand, the two dopants in PbTe have clearly different impacts. Bismuth (Bi) doping greatly reduces the electron mobility while iodine (I) doping is able to maintain a high mobility value (Fig. 2e). This large discrepancy occurs in part because electron states in the conduction band of PbTe are mostly formed by orbitals from Pb, and therefore I doping at the Te site only slightly disturbs the electron state while Bi doping at the Pb site overlaps with the electron states and has a large EDI matrix. In addition, because Bi has a smaller ionic radius

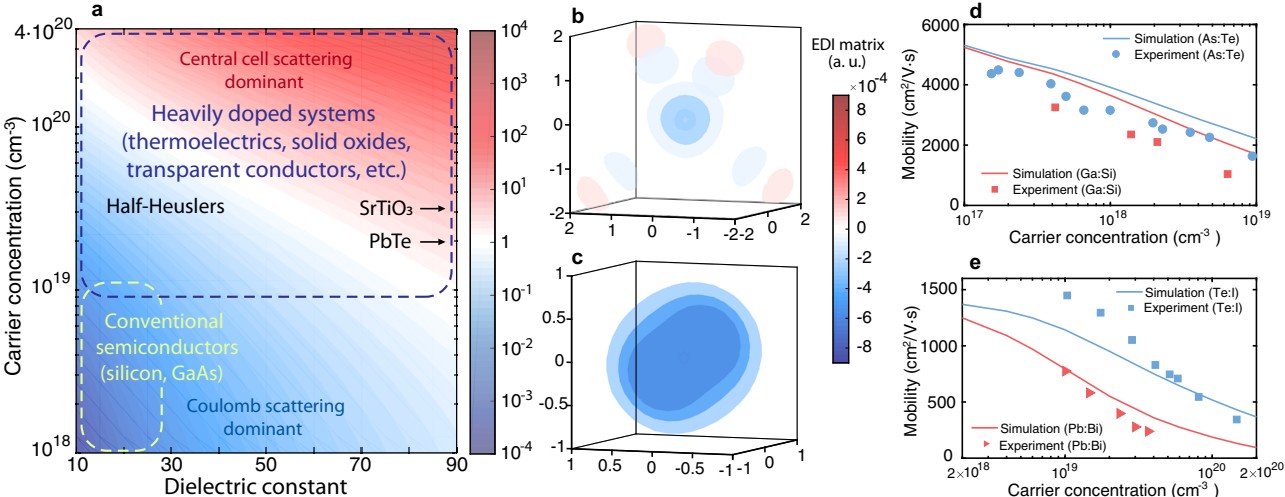

**Fig. 2 Electron-defect interaction. a** Ratio between characteristic electron scattering rates due to central cell scattering and those due to Coulomb scattering, $\eta = \gamma_{cent}/\gamma_{Coul}$, shown as a color map (with red indicating high $\eta$, blue indicating low $\eta$, and white indicating unity, as shown in the scale on the right), with respect to the dielectric constant and carrier concentration. The characteristic electron scattering rate $\gamma$ is defined based on the mobility $\mu$ as $\gamma = e/(m^*\mu)$. Studied materials in this work are also labeled in the plot. Both PbTe and SrTiO$_3$ have large dielectric constants and the arrows indicate their actual locations lie outside the given dielectric constant range. Regions with $\eta \approx 1$ represent cases in which the central cell potential has a comparable impact on charge transport to that of the Coulomb potential, and thus could be harnessed to counteract the Coulomb scatterings. The magnitude of the central cell potential is taken to be 6 eV (with a width of 6 Bohr radius) in this simulation. **b, c** Contour plots of the electron-defect interaction matrix $\langle\psi_{\mathbf{k}}|\triangle\hat{V}|\psi_{\mathbf{k}}\rangle$ for (**b**) Si-doped GaAs and (**c**) Bi-doped PbTe, respectively, where (**k**) is taken to be at the conduction band edge state (at the $\Gamma$ point for GaAs and at the L point for PbTe). Significant electron-defect interaction is seen for PbTe. The unit of the coordinates is Å and the center is at the defect location. **d, e** Computed electron mobility as a function of the carrier density for (**d**) n-type GaAs with two different dopants (Ga:Si, As:Te) and (**e**) n-type PbTe with two different dopants (Pb:Bi, Te:I) in comparison to experimental results [experimental sources: As:Te and Ga:Si[10]; Te:I[41]; Pb:Bi[42]]. For PbTe, Bi doping strongly reduces the mobility compared to I doping, whereas the dopant effects on GaAs are comparatively smaller.

than Pb[22], the short-range potential of Bi dopant is attractive and adds to the long-range Coulomb potential in n-type PbTe. Therefore, the large EDI from Bi doping contributes to strong electron-defect scattering, leading to decreased mobility as seen in Fig. 2e. For both GaAs and PbTe, good agreement between simulation and experiment has been achieved.

Mobility variation with dopants has also been observed in other materials, e.g., in SrTiO$_3$, an oxide with perovskite structure. The computed defect potentials and corresponding electron transport properties for lanthanum (La) doping (on Sr site) and niobium (Nb) doping (on Ti site) are shown in Supplementary Fig. 4. In order to correctly describe the phonon modes in SrTiO$_3$ with perovskite phase, ab initio molecular dynamics was used to extract effective force constants at finite temperature[23,24]. Neither dopant creates significant repulsive potentials that can counteract the Coulomb field. Instead, Nb dopant creates a strong attractive short-range potential that further enhances the electron-defect scatterings. We should mention that in both cases the dopant has an ionic size similar to the host, and therefore the sign of the defect potential is more sensitive to the actual electron density profile and needs to be determined by the calculation. In addition, La and Nb dopants are located at different atomic sites. In such case, the actual projection of valence electron wavefunctions on given atomic sites also influence the magnitude of the electron-defect scattering matrix (more discussions about this will be given later). For SrTiO$_3$, electrons near the conduction band edge mostly consist of $d$ orbitals on Ti (Supplementary Fig. 4), which strongly overlap with the perturbation caused by Nb dopant. This further increases the electron-defect scatterings from Nb dopant. As a result, the computed mobility for Nb-doped SrTiO$_3$ is lower than La-doped one. We caution that the computed mobilities are not accurate due to the use of quasiparticle picture in Boltzmann transport theory[25], which ignores the polaron nature of charge transport in SrTiO$_3$. Nonetheless, the general trend of mobility with different dopants

(La and Nb) agrees with experiments[26] (Supplementary Fig. 4e). The above examples confirmed the electron scattering scenario due to strong sEDIs and further demonstrate the ability of our computational approach to distinguish the impact of different charged defects on electron transport.

**Electron cloaking.** We now discuss in what circumstances the sEDIs can instead enhance the mobility and possible evidences that demonstrate electron cloaking, using half-Heusler materials as examples. Half-Heuslers are a promising material family for spintronic and thermoelectric applications. Their large compositional variability has opened up a wide space for exploring different dopants to optimize their charge transport. We will use the ionic radius as a scale bar for the sign of the short-range defect potential, which determines whether it would be detrimental or beneficial to the charge mobility. First, we establish the correlation between ionic radius and the sEDI matrix. As explained above, a dopant with a large (small) ionic radius compared to that of the host atom tends to create a strong repulsive (attractive) short-range potential $\triangle\hat{V}_{cent}$, and thereby a large positive (negative) value in the sEDI matrix $\langle\psi_{\mathbf{k}'}|\triangle\hat{V}_{cent}|\psi_{\mathbf{k}}\rangle$. We therefore expect the sEDI matrix to correlate with the ionic radius difference between the dopant and host atoms. Figure 3a shows a comparison between computed sEDI matrices in half-Heusler materials (for electrons/holes at the band edge) and the ionic radius difference between the dopant and host atoms, which indeed demonstrates this correlation. Here, the ionic radii are taken from theoretical calculations based on Slater orbitals excluding the valence electrons[22]. Supplementary Fig. 5 further shows the defect potentials for a few dopant/host pairs in different materials, demonstrating that indeed whenever the dopant has a larger ionic size the defect potential becomes relatively more positive (this trend is also observed comparing dopants before

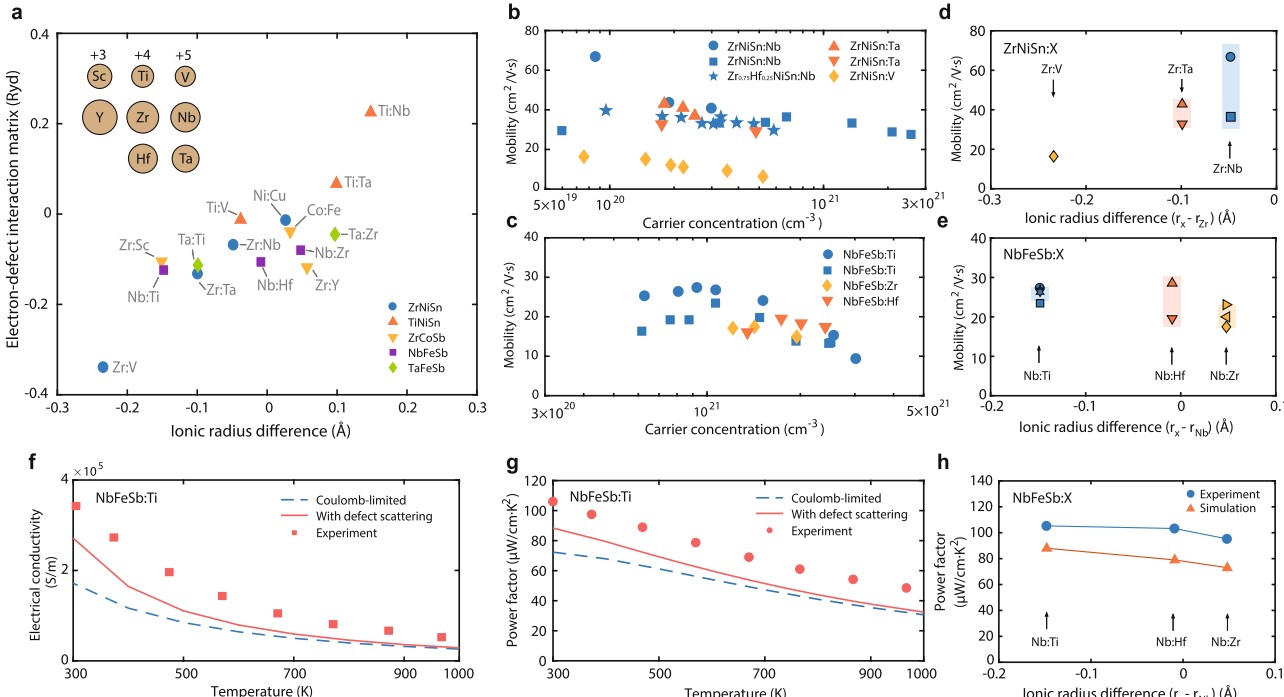

**Fig. 3 Defect-mediated electrical and thermoelectric transport. a** Correlation between ionic radius difference and the short-range electron-defect interaction matrix in half-Heusler materials with different dopants. Inset: relative sizes of the related atoms illustrated based on their ionic radius[22], with the corresponding charges shown at the top of each column. The electron-defect interaction matrix is calculated for the band edge state. **b, c** Compiled experimental data for mobility dependence on the carrier concentration in representative half-Heusler materials with different dopants: **b** n-type ZrNiSn [experimental sources: ZrNiSn:V[43]; ZrNiSn:Nb[27,44]; Zr$_{0.75}$Hf$_{0.25}$NiSn:Nb[45]; ZrNiSn:Ta[46,47]], and **c** p-type NbFeSb [experimental sources: NbFeSb: Ti[29,48]; NbFeSb:Zr and NbFeSb:Hf[49]]. **d, e** Compiled highest mobility data from past studies with respect to the ionic radius difference between the dopant and host atoms for (**d**) n-type ZrNiSn and (**e**) p-type NbFeSb [experimental sources: ZrNiSn:V[43]; ZrNiSn:Ta[46,47]; ZrNiSn:Nb[27,44]; NbFeSb:Ti[29,30,48,50]; NbFeSb:Hf[30,49]; NbFeSb:Zr[30,49,50]]. The ionic radius differences are taken from (**a**). The shaded regions indicate the standard deviation of these extracted mobility data. **f, g** Comparisons between simulations and experimental results for the (**f**) electrical conductivity and (**g**) thermoelectric power factor of Ti-doped NbFeSb from 300 to 1000 K. Consideration of Coulomb scatterings alone underestimates the power factor, while the consideration of full defect scattering with a partial cloaking effect leads to better agreement with the experiment. **h** Comparison between simulations and experimental results[30] for the optimal room-temperature thermoelectric power factor in p-type NbFeSb with respect to the ionic radius difference between dopant and host atoms. The trend of power factor with respect to the ionic radius agrees between the experiments and the simulations.

and after the lanthanide contraction, e.g., Ta and Nb). Here, we mention that because the sEDI matrix is a spatial product of defect potential and the electronic wavefunction, sEDI values will also depend on the projection of electronic wavefunctions on the dopant atom (see more discussions in Supplementary Note). For different materials, even when defect potentials are similar, the varying electronic wavefunctions and their spatial profiles would lead to variation in the sEDI values, and this is the reason why the correlation appears better within each material family. Nonetheless, based on Fig. 3a, we believe the variation in the ionic radius captures the major effect of short-range perturbation and provides a reasonable estimation for the sign and strength of the central cell potential. Based on this plot, dopants in the left region will lead to electron scatterings while dopants in the right region would favor electron cloaking (for n-type materials). Examples of defect potentials from dopants shown in Fig. 3a leading to electron-scattering or -cloaking scenarios can be found in Supplementary Fig. 6.

Classification of dopants based on their ionic radius allows us to further understand the electron transport behavior in heavily doped semiconductors, and is a potential guide in the selection of dopants to enhance mobility. This can be seen in the experimental results of two example compounds—ZrNiSn and NbFeSb, which are well known for their thermoelectric performance among n-type and p-type semiconductors, respectively. For n-type ZrNiSn, the typical

dopants (V, Nb, or Ta) each have a smaller ionic radius than the host atom (Zr, Fig. 3a), resulting in an attractive central cell potential. This attractive potential adds to the attractive Coulomb potential (for n-type material) and increases the defect scattering. As a result, the compiled experimental mobility data mostly show a monotonic decreasing trend with increasing carrier concentration (Fig. 3b). The non-monotonic trend in one data has been attributed to other extrinsic defect scatterings and is not directly related to dopants[27]. In Fig. 3d, we further show highest mobility values for V, Nb and Ta doping from past work with respect to the ionic radius difference between dopant and host ($r_{dopant} - r_{host}$, based on Fig. 3a). A general trend of increasing mobility with increasing dopant ionic radius is observed. This is consistent with our defect scattering picture (Fig. 1a), because for n-type materials a dopant with larger ionic radius is less attractive to electrons and therefore contributes less to total electron-defect scatterings.

On the other hand, for p-type NbFeSb, while the typical dopants (Ti, Hf) each still have a smaller ionic radius than the host atom (Nb), the Coulomb potential is now repulsive (for p-type material), so the two potentials counteract each other. In this case, there will be a carrier concentration range in which the dopant scattering is weak and the mobility approaches the intrinsic electron–phonon scattering limit, manifesting as a peak with increasing carrier concentration (Fig. 1i). Such peaks are indeed observed in the compiled experimental mobility data for

p-type NbFeSb materials (Fig. 3c), suggesting that a partial electron cloaking effect is at play. However, we acknowledge that other extrinsic effects, such as existence of compensated charged defects, may also lead to non-monotonic mobility variation. Another mechanism that is responsible for mobility peaks is the screening of polar optical phonon scattering, an intrinsic scattering mechanism[28]. Still, if the scattering from the dopant is strong, the mobility would instead be limited by EDIs and a monotonically decreasing trend would be expected. The general observation of mobility peaks in NbFeSb suggests electron cloaking effect likely exists. Another evidence supporting electron cloaking effect is shown in Fig. 3e, which plots the highest mobility values from past work for different dopants with respect to the ionic radius difference. In contrast to the case with ZrNiSn, dopant with the largest ionic radius difference from the host (Ti dopant on Nb) has highest mobility, consistent throughout many studies. This contradicts simple defect scattering picture, which would suggest such dopants should create strongest short-range electron-dopant scattering and decrease the mobility. However, because small ionic radius dopants actually create attractive short-range potential that counteracts the repulsive Coulomb field in p-type materials, the overall electron-defect scattering should decrease (and correspondingly the mobility increases) as the dopant's ionic radius becomes smaller, which is the trend observed in Fig. 3e. The electron dopant interaction picture based on the ionic radius is consistent with both n-type and p-type materials (Fig. 3c, e).

In order to quantify the extent to which electron cloaking can benefit the electron transport, and in particular the thermoelectric performance in the case of half-Heusler materials, we computed the electron transport properties in p-type NbFeSb. Ti-doped NbFeSb was recently reported to possess a high power factor at room temperature[29]. The analysis above suggests that Ti doping in p-type NbFeSb facilitates electron cloaking (Fig. 3c, e). Our calculations show that despite the large doping concentrations, the electron scattering rates are indeed close to the intrinsic limit (determined by electron–phonon scattering) due to the counteracting Coulomb potential and the short-range defect potential of Ti (Supplementary Fig. 7). This partial electron cloaking effect leads to higher electrical conductivity and a larger thermoelectric power factor in the range from 300 to 1000 K, bringing the simulation results into better agreement with the experiment (Fig. 3f, g). Moreover, the relative magnitudes of the optimal power factors in NbFeSb with the various dopants experimentally investigated thus far[30] are also consistent with our calculations (Fig. 3h), and the trend with the ionic radius agrees with our electron dopant interaction picture (compare Fig. 3h with Fig. 3e). The higher power factors achieved with Ti and Hf dopants can be understood by their defect potentials: their short-range potentials counterbalance the Coulomb potential and create a partial cloaking effect that enhances the charge mobility (Supplementary Fig. 7). In a similar compound (Ti-doped TaFeSb) we also observed a large power factor enhancement due to electron cloaking (Supplementary Fig. 8). While these enhancements may not seem large, the thermoelectric figure of merit zT is directly proportional to the power factor. In this regard, rational dopant selection that improves the charge mobility and power factor will be beneficial for the thermoelectric efficiency, whose improvement has been a challenging task. Besides, we note that such enhancement due to electron cloaking is expected to become stronger at lower temperatures, due to the increasing importance of defect scatterings compared to intrinsic electron–phonon scatterings as the temperature decreases. Our computation shows that Ti dopant in NbFeSb and TaFeSb can potentially lead to significant enhancement of the power factor by as much as 80% at 150 K compared to the conventional Coulomb-limited case due to

the cloaking effect of the dopant (Supplementary Fig. 9). This thus also provides an opportunity to optimize thermoelectric materials for cooling and refrigeration applications[31] through electron cloaking.

In summary, we have demonstrated that the chemical details of certain dopants can be harnessed to counteract the strong electron scatterings resulting from their long-range Coulomb field. Consequently, in contrast to the conventional belief that charge mobility is always limited by extrinsic Coulomb scatterings, we have shown how high intrinsic mobility can be achieved by rationally selecting dopants based on their ionic radius. While our study focuses on point defects, the first-principles computational approach can be applied to other short-range interactions such as those due to defect clusters, and even dislocations or grain boundaries, when the long-range Coulomb interactions and short-range perturbations are comparable in strength. Our results provide guidelines for dopant selection, which thus far has been mostly based on trial-and-error. The insights provided here on the impact of often-neglected atomic details of defects on charge transport in heavily doped materials will stimulate the search for high-efficiency thermoelectric materials, as well as the development of high-mobility materials for microelectronic and optoelectronic applications.

## Methods

**Model study of electron-defect scattering**. To illustrate the general impact of central cell potential on charge transport, we have used a model semiconductor with a parametrized isotropic band structure and scattering information corresponding to that of silicon, and we represent the defect potential by a simplified profile as shown in Fig. 1a. The charge mobility is given by

$$\mu_e = \left[ \frac{N_v e}{3} \int v^2 \tau \left( -\frac{\partial f^0}{\partial E} \right) D(E) dE \right] / n \qquad (1)$$

where the integration spans over electron states close to the Fermi level, $N_v$ is the band degeneracy, $e$ is the electronic charge, $v$ is the electron group velocity and is related to electron energy via the conductivity effective mass $m_{eff,c}$, as $v^2 = 2E/m_{eff,c}$, $\tau$ is the electron relaxation time, $f^0 = 1/(1 + \exp(\frac{E-\mu}{k_B T}))$ is the Fermi–Dirac distribution function with $\mu$ being the Fermi level, $E$ is the electron energy, $D(E)$ is the electronic density of states and is related to the density-of-states effective mass $m_{eff,DOS}$ via $D = (\frac{2m_{eff,DOS}}{\hbar^2})^{3/2} \frac{\sqrt{E}}{2\pi^2}$ with $\hbar$ being the reduced Planck constant, and $n$ is the carrier concentration.

The electron relaxation time $\tau$, the inverse of the scattering rate, is determined via Matthiessen's rule considering both intrinsic electron–phonon interactions and extrinsic EDIs: $1/\tau = 1/\tau_{e-ph} + 1/\tau_{e-d}$. The electron–phonon interactions consider both acoustic phonon and optical phonon scatterings via corresponding deformation potentials, and the full parametrization is given in Supplementary Note. The electron-defect scattering rate $1/\tau_{e-d}$ is calculated based on the partial wave analysis that evaluates the scattering of electrons by a spherically symmetric potential

$$\frac{1}{\tau_{e-d}} = N_d \frac{4\pi}{\hbar^2} \frac{1}{m_{eff,DOS}\sqrt{2m_{eff,DOS}E}} \sum_{l=0}^{\infty} (l+1)\sin^2(\delta_l - \delta_{l+1}) \qquad (2)$$

where $N_d$ is the volume density of the defect and $\delta_l$ is the phase shift of the electron wave with quantum number $l$. The defect potential (in units of energy) has been taken to have the form $\triangle \hat{V} = \begin{cases} V_0 & r \leq r_0 \\ -\frac{e^2}{4\pi\varepsilon_0 r} & r > r_0 \end{cases}$, with $r_0$ characterizing the range of the short-range potential, and where $V_0$ is the short-range potential energy and $\varepsilon_0$ is the vacuum permittivity. For the results in Fig. 1, we have assumed $V_0 = 9.7$ eV and $r_0 = 1.6$ A. The scattering rates and mobility in Fig. 1d–g are obtained assuming $\varepsilon = 11.7$ (dielectric constant corresponding to silicon) while the dielectric constant is varied in Fig. 2a.

Details of the calculation of the partial wave phase shift are provided in the Supplementary Note. In brief, because the defect potential is spherically symmetric, the orbital angular momentum operator becomes a constant of motion for electrons, and the electron wavefunctions can be represented by an additional quantum number $l$ in addition to its energy $E$, called partial waves. Each partial wave is scattered by the potential and the scattered wave acquires a phase shift $\delta_l$ compared to the case in which no defect is present. Intuitively, the phase shift represents how strongly the defect potential attracts or repels the electrons, both of which will lead to a large phase shift and thus strong scattering. The total scattering rates are obtained via the above formula by considering all values of $l$. The partial wave analysis considers multiple scatterings between the electron wave and the defect, and is thus exact under the

assumption that the electron wave is a plane wave[19]. Although the electron wave actually has a more complex profile modulated by the periodic potential in the crystal, the above results provide an estimation of the effect of the central cell potential on charge transport, particularly in comparison to the conventional Coulomb scatterings.

**First-principles calculation of defect potential.** The defect potential, $\triangle \widehat{V}$, is defined as the difference in the total electronic potential between the system containing one defect and the pristine bulk material, $\triangle \widehat{V} = V_d - V_0$, where the calculation is performed for a cubic supercell containing 96 atoms for half-Heusler materials. In the calculation of the structure with one defect, a net charge is given corresponding to the charged state of the dopant. In our study, the dopants were chosen from a column in the periodic table adjacent to that of the atom being substituted, and thus the net charge is assumed to be $+1$ for n-type and $-1$ for p-type. The extracted defect potential contains both the long-range Coulomb potential and the central cell part which deviates from the Coulomb potential profile. This defect potential cannot be directly used to compute the EDI matrix $\langle \psi_{k'}|\triangle\widehat{V}|\psi_k\rangle$ since the finite size of the supercell does not correctly capture the large span of the long-range Coulomb field. However, because the long-range tail of the defect potential agrees well with the analytic Coulomb potential profile (Fig. 1b, c), we can first subtract the long-range part from the defect potential, leaving only a short-range component $\triangle\widehat{V}_{sr}$. In figures where we show the defect potential extracted from first-principles simulation, we have also plotted the Coulomb potential to show that first-principles defect potential indeed recovers the correct asymptotic trend given by the Coulomb field when one moves away from the defect, as has been shown in previous work[21]. When evaluating the EDI matrix later, we add the long-range Coulomb potential back to the defect potential, $\triangle\widehat{V} = \triangle\widehat{V}_{sr} + \triangle\widehat{V}_{lr}$. Because the second contribution to EDI due to the long-range part can be computed using an analytic expression for the Coulomb potential extending to a longer distance (to be detailed below), we circumvent the difficulty of the finite size of the supercell and are able to treat both long-range and short-range potentials on an equal footing. This workflow is similar to the recent development in incorporating the long-range polar optical phonon scattering into the Wannier interpolation method for electron–phonon interaction calculations[32,33].

When subtracting the long-range part from the defect potential, we compute the long-range potential according to the following formula[34] based on the Ewald summation, which represents the Coulomb potential at location $\mathbf{r}$ due to an infinite periodic array of charge $Ze$ at locations $\mathbf{R}_i$ (given by the supercell size).

$$\triangle\widehat{V}_{lr} = -\sum_i \frac{Ze^2}{\sqrt{|\bar{\varepsilon}|}} \frac{\mathrm{erfc}\left(\gamma\sqrt{(\mathbf{R}_i - \mathbf{r})\cdot\bar{\varepsilon}^{-1}\cdot(\mathbf{R}_i - \mathbf{r})}\right)}{\sqrt{(\mathbf{R}_i - \mathbf{r})\cdot\bar{\varepsilon}^{-1}\cdot(\mathbf{R}_i - \mathbf{r})}}$$
$$-\sum_{\mathbf{G}_i}^{i\neq 0} \frac{4\pi Ze^2}{\Omega} \frac{\exp\left(-\mathbf{G}_i\cdot\bar{\varepsilon}\cdot\frac{\mathbf{G}_i}{4\gamma^2}\right)}{\mathbf{G}_i\cdot\bar{\varepsilon}\cdot\mathbf{G}_i}\exp\left(i\mathbf{G}_i\cdot\mathbf{r}\right) + \frac{\pi Ze^2}{\Omega\gamma^2} \qquad (3)$$

Here $\bar{\varepsilon}$ is the dielectric tensor computed from first principles, $\Omega$ is the supercell volume, and $\gamma$ is a convergence parameter for the Ewald summation. In Supplementary Note, we show that the remaining potential after the subtraction is indeed short-range. Because a short-range potential should approach zero at distances away from the defect, we further align the potential at far distances to zero.

**Electron transport calculation.** The first-principles electron transport properties for half-Heusler materials are calculated by summing over all electron states according to the Boltzmann transport theory[35]. For example, the electrical conductivity is given by

$$\sigma = \frac{e^2}{3\Omega_0 N_k}\sum_{k\alpha} v_{k\alpha}^2 \tau_{k\alpha}\left(-\frac{\partial f_{k\alpha}^0}{\partial E}\right) \qquad (4)$$

in which we have explicitly written out the summation of the discrete mesh points $\mathbf{k}$ in the Brillouin zone, and where $\Omega_0$ is the unit cell volume and $\alpha$ is the band index. Isotropic materials are assumed and the factor $1/3$ appears because the conductivity is the same in all three directions. The equilibrium properties of electrons are calculated from first principles using the QUANTUM ESPRESSO software package[36]. We use the generalized gradient approximation of Perdew et al.[37] with the Troullier–Martins-type norm-conserving semilocal pseudopotential (corresponding to pbe-mt.UPF in the QUANTUM ESPRESSO pseudopotential library). A cutoff energy of 120 Ryd and a $6 \times 6 \times 6$ k-mesh are used to determine the equilibrium lattice constant. The equilibrium properties of phonons and the electron–phonon interaction matrices are calculated using density functional perturbation theory[38] for a $6 \times 6 \times 6$ q-mesh (with a $6 \times 6 \times 6$ k-mesh for the electron–phonon interaction matrix). We then use the EPW software package[39] to interpolate the electronic information and the phonon information, as well as the electron–phonon coupling matrices to a fine mesh via the Wannier interpolation method[40]. In determining the Fermi level $\mu$, we assumed that the dopants are fully ionized and therefore the doping concentration $N_d$ equals the total carrier concentration, which is given by $n = \frac{1}{\Omega_0 N_k}\sum_{k\alpha} f_{k\alpha}^0$.

The electron relaxation time $\tau$ is determined via Matthiessen's rule considering both intrinsic electron–phonon interactions and extrinsic EDIs: $1/\tau = 1/\tau_{e-ph} + 1/\tau_{e-d}$. The electron–phonon interaction matrices are first calculated within density functional perturbation theory and then interpolated via the Wannier interpolation scheme to the fine mesh[39]. This includes electron scatterings by polar optical phonons[32,33], as well as taking the carrier screening effect into account. More details on this can be found in our previous work[35] and in Supplementary Note.

The calculation of electron-defect scattering rates considering the full defect potential is the key step in our study, and is given by the following formula under the momentum relaxation approximation

$$\frac{1}{\tau_k^{e-d}} = N_d\Omega_0\frac{2\pi}{\hbar}\frac{1}{N_k}\sum_{k'}\left(1 - \frac{\mathbf{v}_k\cdot\mathbf{v}_{k'}}{|\mathbf{v}_k||\mathbf{v}_{k'}|}\right)|g_{e-d}(\mathbf{k},\mathbf{k}')|^2\delta(E_k - E_{k'}) \qquad (5)$$

where $N_d$ is the volume density of dopants, $N_k$ is the number of $\mathbf{k}$ points, the factor $1 - \frac{\mathbf{v}_k\cdot\mathbf{v}_{k'}}{|\mathbf{v}_k||\mathbf{v}_{k'}|}$ takes into account the fact that scatterings between electrons with similar velocity directions do not contribute much to momentum loss and thus less to electrical resistance, and $\delta(E_k - E_{k'})$ indicates that the defect scattering is an elastic process. $g_{e-d}(\mathbf{k},\mathbf{k}') = \langle\psi_{k'}|\triangle\widehat{V}|\psi_k\rangle$ is the EDI matrix. As explained above, the defect potential $\triangle\widehat{V}$ contains both long-range and short-range parts, leading to two contributions to EDI. To compute these contributions, the EDI matrix is first rewritten as follows[17]

$$\langle\psi_k|\triangle\widehat{V}|\psi_{k'}\rangle = \int d^3r\, u_{k'}^* e^{-ik'\cdot r}\triangle\widehat{V} u_k e^{ik\cdot r} = \sum_G \triangle V(\mathbf{k}' - \mathbf{k} + \mathbf{G})\langle u_{k'}|e^{iG\cdot r}|u_k\rangle \qquad (6)$$

where the Fourier transform of the defect potential is defined as $\triangle V(\mathbf{q}) = \frac{1}{\Omega_0}\int d^3r \triangle\widehat{V}(\mathbf{r})e^{-i\mathbf{q}\cdot\mathbf{r}}$, with $\Omega_0$ being the unit cell volume and the integration spanning the entire space. This form separates the defect potential from the wavefunctions, and the factor containing wavefunctions can be computed readily once the periodic components $u_k$ of the wavefunctions are known[17]: $\langle u_{k'}|e^{iG\cdot r}|u_k\rangle = \int d^3r\, u_{k'}^* e^{-iG\cdot r}u_k$, where the integration spans over the unit cell. To evaluate the EDI matrix, we then compute the Fourier component of the defect potential $\triangle V(\mathbf{q})$, which again contains both long-range and short-range parts. The short-range part can be calculated readily within the supercell based on $\triangle V_{sr}(\mathbf{q}) = \frac{1}{\Omega_0}\int d^3r\triangle\widehat{V}_{sr}(\mathbf{r})e^{-i\mathbf{q}\cdot\mathbf{r}}$ since the potential has negligible contributions at far distances. The long-range part can be obtained by performing the integration analytically to infinity, yielding $\triangle V_{lr}(\mathbf{q}) = -\frac{Ze^2}{\Omega_0\varepsilon\varepsilon_0}\frac{1}{|\mathbf{q}|^2 + (1/L_D)^2}$. For this expression, we have assumed the Coulomb potential energy is given by $\triangle\widehat{V}_{lr}(\mathbf{r}) = -\frac{Ze^2}{4\pi\varepsilon\varepsilon_0}\frac{\exp(-r/L_D)}{r}$, where the factor $\exp(-r/L_D)$ considers the carrier screening at high carrier concentrations with the Debye screening length $L_D$ given by

$$L_D = \left(\frac{e^2}{\varepsilon\varepsilon_0}\int\left(-\frac{\partial f}{\partial E}\right)D(E)dE\right)^{-1/2} \qquad (7)$$

Adding both long-range and short-range components as $\triangle V(\mathbf{q}) = \triangle V_{sr}(\mathbf{q}) + \triangle V_{lr}(\mathbf{q})$ allows us to evaluate the EDI matrix completely. The electron-defect scattering rates thus obtained are then combined with electron–phonon scatterings to give the total electron scattering rates, which are used to evaluate the electron transport properties, as in Eq. (4).

**Reporting summary.** Further information on research design is available in the Nature Research Reporting Summary linked to this article.

## Data availability

All data needed to evaluate the conclusion in the paper are present in the paper and/or the Supplementary Information. Additional data related to this paper are available from the corresponding authors upon reasonable request.

## Code availability

The code for computing electron scattering rates through first-principles electron transport calculation is a modified version of the EPW code[39], originally released within the QUANTUM ESPRESSO package[36]. Our modified EPW code is available at https://doi.org/10.24435/materialscloud:5a-7s.

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

## Acknowledgements

We thank T.-H. Liu, M. Li, and Q. Zhang for helpful discussions on the first-principles calculations and electron-defect interactions. The work performed at MIT is supported by the DARPA MATRIX program under Grant No. HR0011-16-2-0041.

## Author contributions

J.Z. and G.C. conceived the project. J.Z. performed the theoretical analysis and first-principles computation. J.Z. and G.C. analyzed the data and wrote the manuscript, with contributions from H.Z., Q.S., Z.D., J.M., and Z.R. All authors commented on, discussed, and edited the manuscript.

## Competing interests

The authors declare no competing interests.
