## [Peer review file · Nature Communications]

REVIEWER COMMENTS

Reviewer #1 (Remarks to the Author):

The manuscript by Zhou Jiawei and co-workers attempted to show that electron cloaking can also be observed for charged defects, extending their earlier theoretical work on electron waves interacting with core-shell particles in which dimensions and electron wavelengths are carefully selected to suppress reflection wave of incident wave(PRL2012).

The present work claimed a similar effect occurs with charged defects. However, charged defects interact with electrons via the long-range Coulomb potential, unlike neutral nanoparticles, so I do not understand how cloaking can still occur. The authors claimed that the central cell correction portion, which in essence embodies the atomic electronic structure of the dopant atom, can control the probability of scattering of an incoming electron, including suppressing it completely, and purported to show some sample calculations that this is the case. How can this be true, if the electron is already scattered by the long-range portion of the charged defect? My doubt seems to be reinforced by the generally poor agreement between theory and experiment, for example, in PbTe:I and PbTe:Bi, see Fig 2e, where such an effect is claimed.

The defect potential plots computed in Figure S4 in fact show eV-scale sharp rises and troughs at the edge of the charge defect (1.5-2.0 Å). The very short spatial length scale of these features is very puzzling to me. Does atomic potential fluctuate so strongly and so rapidly? They appear to me to be an artifact of a large background dielectric constant used in calculations. If this is indeed the case, then the results would be wrong. Dielectric constant as a phenomenological concept is meaningful only for effects averaged over a unit cell or larger. The authors compared their defect potential to a Coulomb potential. The Coulomb potential indeed appears to have been wrongly scaled by the background dielectric constant over atomic dimensions.

The authors then described a dopant-selection criterion based on "ionic radius". The authors explained: "Here the ionic radius

refers to the size of the ion when valence electrons are not considered." I think this definition is very confusing. The authors seem to be comparing the size of the cores of the ions in different charge states, since the dopant and the host ions differ by one charge overall. In any case, if the size of the dopant is larger or smaller than the host, surely important effects due to polarization and deformation of the site must be taken into account to treat electron scattering?

Finally, the results of Figure 3 are too indirect to provide a test of theory. Thus, I'm unfortunately not persuaded that a new understanding has been obtained with this manuscript.

Reviewer #2 (Remarks to the Author):

In this paper, the authors propose that the nature of the dopant could influence the defect scattering process. Depending on the dopant size, mobility could be even increased due to a dopant cloaking effect.

This is an interesting concept and the theoretical analysis appears sound (with some open questions though, see further). However, for publication in a high impact journal such as Nature Communications, the authors should further demonstrate that the theoretical effect they suggest is indeed observed experimentally and that this effect is important. More specifically, while their model can be tuned to inject parameters leading to a large effect (as in Fig. 1 which only uses hypothetical parameters), it remains to be shown that real materials have such a peak in mobility vs carrier concentration and that this peak is large enough to be of practical importance. The examples provided especially on the heussler are not convincing.

1) the absence of a peak in mobility for ZnNiSn vs NbFeSb which shows one stressed by the authors appears to be inconsistent with previous work such as refs 22 which shows also a peak in ZnNiSn.

2) the cloaking effect is arguably quite marginal in these heusslers. This appears to change the power factor by around 10% only which is interesting but not very impactful.

3) is there any other explanation of a peak in mobility with carrier concentration. This should be clearly discussed.

I would suggest the authors to revise their manuscript and maybe choose another system where the effect could be more pronounced. Materials with higher dielectric constant could be a good target. There is for instance plenty of data for mobility on SrTiO₃ arguably on better single crystal sample which should lead to an important cloaking effect in view of its large dielectric constant.

I also a few more technical comments to make:

-the authors stress the ionic radius effect. Does electronegativity play a role too or is electronegativity correlated with ionic radius here?

-in the model do you have to consider the full (ionic + electronic) or only the electronic part of the dielectric constant and why?

-the authors mention "many oxide and thermoelectric materials also possess a high dielectric constant associated with soft phonon modes". Could they quantify this statement? If high dielectric constant materials are common (and what is "high" exactly). Their effect should be common? Maybe using a database of material property (computed or experimental) could help quantifying how many high dielectric constant materials there are. This is important to evaluate if the effect is truly common.

-I would recommend that the authors plot the materials they study (GaAs, PbTe and heuslers and/or others if they follow my suggestion) in the Fig. 2a

-why do the authors have a factor $1/3$ in their eq. (4)?

-"computed with DFT is vague" what functional? I know this is in SI but that's worth being in the main text

-related to my general comments on hypothetical parameters, why did you choose the given value for r_0 and V_0 for the model in Fig. 1 (see line 266)?

-line 189: Where is the point for Zr:Nb (Fig3a) for ZnNiSn?

There is a point for Nb:Zr but not for the same material

-As they reach very large doping the dielectric constant should be affected. The dielectric constant of a metal is infinite. Do the authors take this into account? And if not, why?

-the difference of the effect on p-type or n-type material could be better explained so the discussion on the heusler is easier to follow.

Revision report for MS# NCOMMS-21-20659-T

Mobility enhancement in heavily doped semiconductors *via* electron cloaking

Jiawei Zhou, Hangtian Zhu, Qichen Song, Zhiwei Ding, Jun Mao, Zhifeng Ren, Gang Chen

We thank all the reviewers for their thoughtful comments on the manuscript that have helped us to improve the manuscript. Our responses and revisions (in blue) are elaborated on below. In addition, the revised parts in the main manuscript and supplementary information are also highlighted (in yellow). The revised manuscript and supplementary materials are attached in this document, following our revision report.

Reviewer #1:

The manuscript by Zhou Jiawei and co-workers attempted to show that electron cloaking can also be observed for charged defects, extending their earlier theoretical work on electron waves interacting with core-shell particles in which dimensions and electron wavelengths are carefully selected to suppress reflection wave of incident wave(PRL2012).

Response: We thank the reviewer for his/her comments that have helped us to improve the manuscript. In the revised manuscript we have provided more detailed discussions to address the reviewer's questions. Furthermore, we provided additional evidence that support the electron cloaking effect in NbFeSb, which we will discuss below. In addition, we have also computed another material – SrTiO₃, and found qualitative agreement with experiment regarding the trend of mobility with different dopants, which further supports our computational method. We hope the reviewer will find our work suitable for publication. Our detailed responses are as follows.

The present work claimed a similar effect occurs with charged defects. However, charged defects interact with electrons via the long-range Coulomb potential, unlike neutral nanoparticles, so I do not understand how cloaking can still occur. The authors claimed that the central cell correction portion, which in essence embodies the atomic electronic structure of the dopant atom, can control the probability of scattering of an incoming electron, including suppressing it completely, and purported to show some sample calculations that this is the case. How can this be true, if the electron is already scattered by the long-range portion of the charged defect? My doubt seems to be reinforced by the generally poor agreement between theory and experiment, for example, in PbTe:I and PbTe:Bi, see Fig 2e, where such an effect is claimed.

Response: We would like to clarify the mechanism of electron cloaking in more detail to address the reviewer's question. First, the cloaking effect we describe is due to the interaction between electron waves and the electron potential created by defects in general. Cloaking indicates the condition when the potential creates negligible scattering for the electron waves. In the case of core-shell nanoparticles¹, the potential profile is created by electronic band structure. Potential profile created by defects can potentially lead to cloaking effect as long as the corresponding electron scattering cross sections due to the defect potential are small.

In our case, the potential is created by a charged defect. Potential from a charged defect includes both short range perturbations and long-range parts. These two parts are not separable when considering electron scatterings. In the conventional Brooks-Herring model, the short-range part is neglected and only the long-range Coulomb part is considered, and therefore electrons are always scattered. However, if one considers the true potential that includes both short-range and long-range parts, one may achieve smaller total electron scatterings than the case when only Coulomb scatterings are considered. This is because the short-range and long-range parts can coherently add up and cancel each other, which is essentially a cloaking effect. Some of past literature treated long-range and short-range scattering rates as additive, which will only lead to stronger scattering. Such a treatment is wrong, however, since it is the net potential profile that determines the electron scattering.

To quantitatively demonstrate this point, here we provide the calculated phase shifts for the electron wavefunctions when evaluating electron scatterings for the model semiconductor (Figure R1).

Figure R1. Phase shift analysis for electron-defect scattering in a model semiconductor. **a**, Phase shifts of the electron wave with angular momentum quantum number $l = 1$ for different defect potential profiles. **b**, Corresponding electron-defect scattering rates. In both cases, the carrier concentration is $6.3 \times 10^{19} \text{ cm}^{-3}$. The energy is plotted for electrons starting from the conduction band edge.

The phase shift analysis is a systematic method to calculate how waves are scattered by a given potential². The calculated phase shifts represent how scattered waves differ from the incoming waves due to the scattering potential. If no scattering occurs, phase shifts will be zero. When scattering happens, phase shifts will generally be negative for attractive potential and positive for repulsive ones. As can be seen in Fig. R1a, attractive Coulomb potential leads to negative phase shifts for electrons, while repulsive short-range (central cell) potential leads to positive phase shifts. In contrast, when we consider the full potential that includes both the short-range part and long-range Coulomb part, the phase shift is close to zero for low energy states (within about 0.1 eV above the conduction band edge). This directly translates to smaller total electron-defect scattering rates in the case of full defect potential (Fig. R1b, yellow curve).

We also want to mention that in our simulation for PbTe with I and Bi dopants, we do not observe electron cloaking in PbTe. This is because both I and Bi dopants do not create central cell potential that counteracts the Coulomb part. For I dopant, the central cell scattering is weak and the dominant scattering is due to the Coulomb potential. For Bi dopant, the central cell *adds to* Coulomb potential and further increases the scatterings. Besides, we think the agreement between simulation and experiment is reasonably good, considering that no fitting parameter has been used in our modeling. Even though mobility differences are often observed with different dopants, it has not been quantitatively simulated based on a microscopic theory. Our results present such efforts for the first time so that one can calculate electron mobilities in doped semiconductors and differentiate the effects from different dopants using first principles simulation.

Revision: To further clarify how short-range defect potential can counteract the Coulomb scattering and lead to smaller total electron-defect scattering rates, we have added following two paragraphs following the introduction section in the main text

“However, if a large central cell potential exists and has a sign opposite to that of the Coulomb potential, it could counteract the scatterings due to the Coulomb field and enhance electron mobility. The reason is that the electron defect scattering is governed by the magnitude of the electron-defect interaction (EDI) matrix (to the first order), given by the spatial integration of the defect potential $-\langle \psi_{k'} | \Delta \hat{V} | \psi_k \rangle$, where ψ_k is the electronic wavefunction with wave vector \mathbf{k} . The defect potential $\Delta \hat{V}$ includes both the central cell part and long-range Coulomb part, i.e., they add coherently to the EDI matrix. When the central cell potential has an opposite sign to that of the Coulomb potential, it will lead to cancelling terms in the above spatial integration, and correspondingly a small EDI matrix, which implies weak or negligible electron-defect scattering. Note that some of past literature calculated the scattering rate due to the central cell and the long-range Coulomb contribution separately, and added the two rates according to the Matthiessen’s rule. Such a treatment neglects the coherence effect of electron waves. In this treatment, the inclusion of the central cell potential always leads to a higher scattering rate. Conceptually, this is not the correct treatment since it is the net effect from the central cell and the long-range Coulomb interaction that impacts the electron scattering.”

We have also revised Figure 1 and included the phase shift results for both electron scattering and electron cloaking scenarios:

Revised Figure 1. Impact of charged defects on electron transport. **d-e**, Phase shifts of scattered wave functions with angular momentum quantum number of $l = 1$ for two scenarios (**d**: electron scattering, **e**: electron cloaking).

and added discussions about the phase shifts in the main text:

“To illustrate the consequence of different defect potentials on electron transport, we first study a model semiconductor as an example (see modeling details in Methods). Shown in Fig. 1d-e are scattered electron phase shifts computed for a n-type model semiconductor with different defect potentials. Phase shifts quantify how scattered electron waves differ from the incoming waves, and

their magnitudes represent the strength of electron-defect scatterings². In general, attractive potentials lead to negative phase shifts, while repulsive potentials lead to positive phase shifts. When the attractive short-range potential coexists with the attractive long-range Coulomb potential in n-type materials, the phase shifts add up in magnitude (Fig. 1d), leading to increased electron scattering rates and reduced mobility (Fig. 1f,h). In contrast, if the short-range potential is repulsive, it counteracts the attractive long-range Coulomb field and reduces the overall phase shifts (Fig. 1e), decreasing the total electron-defect scattering rates (Fig. 1g).”

In addition, we have added following discussion in the main text to clarify that the electron transport in PbTe belongs to the electron scattering scenario:

“In addition, because Bi has a smaller ionic radius than Pb³, the short-range potential of Bi dopant is attractive and adds to the long-range Coulomb potential in n-type PbTe. Therefore, the large EDI from Bi doping contributes to strong electron-defect scattering, leading to decreased mobility as seen in Fig. 2e.”

The defect potential plots computed in Figure S4 in fact show eV-scale sharp rises and troughs at the edge of the charge defect (1.5-2.0 Å). The very short spatial length scale of these features is very puzzling to me. Does atomic potential fluctuate so strongly and so rapidly? They appear to me to be an artifact of a large background dielectric constant used in calculations. If this is indeed the case, then the results would be wrong. Dielectric constant as a phenomenological concept is meaningful only for effects averaged over a unit cell or larger. The authors compared their defect potential to a Coulomb potential. The Coulomb potential indeed appears to have been wrongly scaled by the background dielectric constant over atomic dimensions.

Response: We would like to first clarify the calculation of the defect potential. First, in the first principles calculation of defect potential (e.g. those shown in Figure S4), no dielectric constant is given as the input. The obtained defect potential are direct results from the simulation by subtracting the potential of pristine material from that with the defect. Further, we agree with the reviewer that the dielectric constant as a mean field concept is not valid at atomic dimensions. However, as one moves away from the defect, the potential should asymptotically approach the one given by the screened Coulomb potential with the background dielectric constant, as shown before⁴ (defect formation energy correction schemes are also based on similar assumptions⁵). We compared the first principles defect potential with the screened Coulomb field mainly to show that the first principles generated potential indeed reproduces the correct asymptotic trend at longer distances, and did not mean that the potential should also follow the screened Coulomb profile at short range. In fact, the deviation from the screened Coulomb profile indicates the range where the conventional Coulomb scattering model breaks down. The exact length scale at which this happens will depend on the material and defect, but can be as small as a few Å to less than 20 Å^{4,5}.

Next, we would like to comment on the short-range deviations of the defect potential. The strong oscillations of defect-induced potentials at short range have been previously observed in first principles calculations⁴. These oscillations are not strange, and are direct results of the oscillating electron wavefunctions due to the fact that the outer electrons have to be orthogonal to the core electrons, leading to strong oscillation particularly near the core⁶. However, in previous studies on defects, defect potentials were often not reported. The reported quantity is usually the planar

averaged potential^{4,5} (the defect potential is averaged over a plane), which averages out the strong oscillation. Here, to fairly compare our results with previous simulation, we also calculate the planar averaged defect potential. In the case of a cubic supercell (as illustrated in Fig. R2a), this averaged potential is calculated by

$$\overline{\Delta V}(z) = \frac{1}{A} \iint_0^a dx dy \Delta \hat{V}(x, y, z)$$

where a is the lattice vector length of the supercell, and A is the area of the x-y plane of the supercell.

Figure R2. Planar average of defect potential. **a**, Schematic of the planar average in a cubic supercell. **b-c**, Planar averaged defect potential for Si-doped GaAs (b), and Bi-doped PbTe (c). The z axis starts from the plane containing the defect.

Figure R2b-c shows the planar averaged defect potential for Si-doped GaAs and Bi-doped PbTe, obtained from the three-dimensional defect potentials (Supplementary Fig. 2). The planar averaged defect potentials from our calculation are on the same order of magnitude with previously reported planar averaged defect potential^{4,5} (around 0.1 to 1 eV). Therefore, the behavior and magnitude of our calculated defect potential is consistent with previous studies. In fact, the true radial representation should show stronger oscillations. The oscillations of defect potential at short range are indications of strong perturbations from the defect.

Revision: In addition to discussion in the Supplementary Note, we have also added further clarifications in the Methods section, and the figure captions of Supplementary Fig. 2, 3 and 5, to indicate that the comparison of first principles defect potential with the Coulomb potential is only to justify that the defect potential indeed recovers the correct trend away from the defect.

In Methods, First-principles calculation of defect potential section, we added

“In figures where we show the defect potential extracted from first principles simulation, we have also plotted the Coulomb potential to show that first principles defect potential indeed recovers the correct asymptotic trend given by the Coulomb field when one moves away from the defect, as has been shown in previous work⁴.”

In the figure caption of Supplementary Fig. 2 and 3, we added

“The asymptotic trend given by the Coulomb field is recovered as one moves away from the defect.”

In the figure caption of Supplementary Fig. 5, we added

“Coulomb potentials are also plotted in (b-c) and (e-f), to show that the asymptotic trend given by the Coulomb field is recovered as one moves away from the defect.”

We have also included the planar averaged defect potentials for Si-doped GaAs and Bi-doped PbTe in Supplementary Fig. 2

Revised Supplementary Figure 2. c-d, Planar averaged defect potentials for Si-doped GaAs (c) and Bi-doped PbTe (d). The planar averaged potential is calculated by $\overline{\Delta V}(z) = \frac{1}{A} \iint_0^a dx dy \Delta \hat{V}(x, y, z)$ where a is the lattice vector length of the supercell, and A is the area of the x-y plane of the supercell, as indicated by the inset of (c). The z axis starts from the plane containing the defect.

and have included discussions about their magnitude in the main text:

“We note that the planar averaged defect potentials generally vary around 0.1 – 1 eV at the short range, on the same order of magnitude with previous reports^{4,5}.”

The authors then described a dopant-selection criterion based on "ionic radius". The authors explained: "Here the ionic radius refers to the size of the ion when valence electrons are not considered." I think this definition is very confusing. The authors seem to be comparing the size of the cores of the ions in different charge states, since the dopant and the host ions differ by one charge overall. In any case, if the size of the dopant is larger or smaller than the host, surely important effects due to polarization and deformation of the site must be taken into account to treat electron scattering?

Response: We thank the reviewer for raising this question. Here we want to give more detailed reasonings behind our use of the ionic radius concept to differentiate different dopants. We will use phosphorus-doped Si as an example. A silicon atom can be considered as a silicon ion with +4 charge and four electrons, which we denote as $\text{Si}^{+4}[2s2p]$. Similarly, a phosphorus atom can be denoted as $\text{P}^{+5}[2s3p]$. The reason we separate the valence electrons from the core is that valence electrons participate in chemical bonding and will be more strongly affected by the lattice structure, while the core electrons are more localized. When a Si atom is substituted by a P dopant, the P dopant becomes positively charged (with +1 charge), releasing one electron into the lattice. The remaining four electrons in the outer shell of P atom would participate in the chemical bonding, just as the valence electrons of Si would do, except that the core potential is different from that of Si. If the core potential of P^{+5} were the same as that of Si^{+4} , the outer four valence electrons would bond the same way as in pristine silicon, and no defect scatterings would occur. In other words, the defect potential (induced by replacement of one Si by one P) results from the difference in the core potential of P^{+5} and Si^{+4} .

The first major difference between the core potential of P^{+5} and Si^{+4} is that their charges differ by one, which lead to a long-range Coulomb field when Si^{+4} is replaced by P^{+5} , as the reviewer also pointed out. In addition to this, Si^{+4} and P^{+5} have different sizes (ionic radius), and thereby attract the valence electrons to different extents. When the dopant ion has a smaller size compared to the host ion, propagating electron states (which are composed of mainly valence electrons) will have deeper penetration into the core, which is equivalent to creating an attractive short-range defect potential. This is the case for P-doped silicon. Conversely, if a dopant ion has a larger size than the host ion, it will tend to repel electrons due to Pauli exclusion and leads to an equivalent repulsive short-range potential. This is the basis on which the ionic radius concept is used to infer the strength and sign of the short-range electron-defect interaction.

We agree with the reviewer that the ionic radius does not capture all the details related to electron-defect interactions. As mentioned by the reviewer, polarization and deformation of the site can occur because of the size difference. However, incorporating these into one indicator that describes electron-defect scattering is beyond the scope of our work. Rather, we hope to provide a simplified yet useful indicator to tell whether a dopant would be beneficial or not, and we find the ionic radius serves this purpose, as demonstrated by the favorable correlation between ionic radius and short-range electron-defect scattering matrix (Fig. 3a). We believe the details of the electron-defect interactions, including polarization and deformation of the electron charge density near the defect, are interesting directions to explore and could be studied in future work.

Revision: To further clarify the dopant selection based on the ionic radius concept, we have rewritten the relevant paragraphs about the ionic radius in the main text and have added more discussions:

“Here we show that the ionic radii of the dopant and host atoms can be used to indicate the sign of the central cell potential (Fig. 1a). The ionic radius here refers to the size of the ion without valence electrons. For example, a silicon atom can be considered as a silicon ion with +4 charge and four electrons, which we denote as $\text{Si}^{+4}[2s2p]$. Similarly, a phosphorus atom can be denoted as $\text{P}^{+5}[2s3p]$. The reason we separate the valence electrons from the core is that valence electrons participate in chemical bonding and will be more strongly affected by the lattice structure, while

the core electrons are more localized. When silicon is doped with phosphorus (P), phosphorus atom will lose one electron (which becomes free) and becomes positively charged. The remaining four electrons in the outer shell of P atom would participate in the chemical bonding, just as the valence electrons of Si would do, except that the core potential is different from that of Si. The defect potential (perturbed potential due to replacement of one Si by one P) thus results from the difference in the core potential (P^{+5} compared to Si^{+4}).

The first major difference between the core potential of P^{+5} and Si^{+4} is that their charges differ by one, which lead to a long-range Coulomb field when Si^{+4} is replaced by P^{+5} . In addition, Si^{+4} and P^{+5} have different sizes, and thereby attract the valence electrons to different extents. When the dopant ion has a smaller size compared to the host ion and replaces the host ion, propagating electron states (which are composed of mainly valence electrons) will have deeper penetration into the core, which is equivalent to creating an attractive short-range defect potential (Fig. 1b). This is the case for P-doped silicon. Conversely, if a dopant ion has a larger size than the host ion, it will tend to repel electrons due to Pauli exclusion and leads to an equivalent repulsive short-range potential (Fig. 1c).

Now we consider short-range and long-range parts together for the defect potential. For an n-type dopant, the long-range Coulomb potential is attractive. A dopant with smaller ionic radius than the host would tend to create a short-range attractive potential, which adds to the Coulomb part and further increases EDI strength, leading to stronger electron-defect scatterings (electron scattering scenario). In contrast, dopants with larger ionic radius will tend to create repulsive short-range potential, which counteracts the Coulomb potential. When the cancellation effect is maximal, the defects will appear to have negligible scatterings for electrons, and this realizes electron cloaking (electron cloaking scenario). As seen from the computed scattered electron wavefunction, the probability flux is distorted for the electron scattering scenario (Fig. 1b), while undistorted probability flux is recovered away from the defect for the electron cloaking scenario (Fig. 1c). In short, to achieve electron cloaking effect, dopants with larger ionic radius are favored for n-type materials. Thus, the ionic radius provides a useful guidance for understanding and selecting dopants in terms of the electron mobility. Effectively, the ionic radius can be seen as a scale bar for dopant selection: in n-type materials, dopants with larger ionic radius are more desired than those with smaller ionic radius because they are more repulsive to electrons and tend to counteract the Coulomb scatterings (Fig. 1a).”

Finally, the results of Figure 3 are too indirect to provide a test of theory. Thus, I'm unfortunately not persuaded that a new understanding has been obtained with this manuscript.

Response: We hope our responses above have addressed the reviewer's questions. Here, we want to further summarize the major findings in this work and provide additional evidence that support our electron-dopant interaction picture. First of all, we developed a first principles computational method that can quantify the charge mobility variation due to different dopants, which has been often observed but has not been quantitatively simulated based on a microscopic theory. With this method, the electron mobility differences with different dopants across a range of material systems, including GaAs, PbTe, and half Heuslers, can now all be understood based on their microscopic short-range electron-dopant scattering matrix, for which we have obtained reasonably good agreement between simulation and experiment (Fig. 2d-e, Fig. 3f-h). In addition to these materials,

we have included the simulation results of another material – SrTiO₃, a representative oxide with perovskite structure – in our revised manuscript (Fig. R3). Though our electron transport modeling based on Boltzmann transport equation is not accurate for SrTiO₃ because it ignores the polaron nature of its charge transport⁷, the general trend of mobility variation with different dopants (Fig. R3d) is reproduced⁸, again demonstrating the ability of our computational method to distinguish between different defects. We believe the ability to model and differentiate different dopant has its own significant merit and will benefit researchers working on materials related to electron transport.

Figure R3. Electron transport simulation of SrTiO₃. a-b, Defect potentials for (a) La dopant on Sr site, and (b) Nb dopant on Ti site. c, Comparison of electron-defect scattering rates for La and Nb dopants and the intrinsic electron-phonon scattering rates. d, Mobility considering different scattering conditions, and comparison between simulation and experiment⁸. The calculated mobilities assume a carrier concentration of $6 \times 10^{19} \text{ cm}^{-3}$.

Second, we demonstrated the correlation between the short-range electron-dopant interaction matrix and the ionic radius difference between dopant and host atoms (Fig. 3a), thereby establishing the ionic radius as a useful indicator for the strength and sign of the short-range electron-dopant interaction. Because the short-range electron-dopant interaction has a large impact on the charge mobility, we can then rationalize the dopant selection using the ionic radius concept.

Third and most importantly, based on our theoretical analysis, we show that the impact of short-range dopant potential on the electron transport does not only depend on the magnitude of the potential, but the sign matters as well. In particular, when the short-range perturbation counteracts the long-range Coulomb potential, electron cloaking can happen and the mobility is larger than the case if one solely considers Coulomb scattering. While the theory indicates that a large improvement is possible, we acknowledge that in practice this would depend on the exact potential profiles created by the defect and the material. Nonetheless, this picture provides a scale bar (which can be approximately represented by the ionic radius, as shown in revised Fig. 1a) that separates desirable dopants from undesirable ones. Conventional theory would suggest dopants with weak short-range potentials, or negligible ionic radius difference from the host, to be most desirable. However, our analysis shows this was not the full picture, because moving further along the ionic radius scale bar can achieve even higher mobility by counteracting the long-range Coulomb scatterings.

To further demonstrate the last point, we have provided another evidence in the revised manuscript. In Figure R4, we have collected previously reported highest mobility values from different studies, and organized them based on the dopants they use. Figure R4a shows n-type ZrNiSn doped with V, Ta, or Nb (all on Zr site), and Fig. R4b shows p-type NbFeSb doped with Ti, Hf, or Zr (all on Nb site). These include data already shown in Fig. 3, but also those that studied the dopant effect but did not report carrier concentration dependence. The mobilities for different dopants are plotted against the ionic radius difference between the dopant and the host atom (as given in Fig. 3a).

Figure R4. Trend of electron mobility and power factor with respect to ionic radius. a-b, Compiled highest mobility data from past studies with respect to the ionic radius difference between the dopant and host atoms for (a) n-type ZrNiSn and (b) p-type NbFeSb. The dopants are labeled inside the figures with arrows pointing to the mobility data. The ionic radius differences are taken Figure 3a. c, Comparison between simulation and experiment for the optimal room-temperature power factor in p-type NbFeSb with respect to the ionic radius difference between the dopant and host atoms.

Conventional theory for defect scattering would suggest that the central cell scattering due to the short-range defect potential should be minimized to favor high mobility. In other words, dopants with small ionic radius difference from the host atom are most desirable. This agrees with what is observed in n-type ZrNiSn (Fig. R4a), but not with p-type NbFeSb (Fig. R4b, the ionic radius difference between Ti dopant and Nb in fact is the largest). If one selects other length scale to represent the short-range perturbation (e.g. atomic radius instead of ionic radius), one would reach the same conclusion: the higher mobility seen in Ti-doped NbFeSb (compared to Hf and Zr dopants) is not consistent with the conventional theory that favors minimal short-range perturbations.

In contrast, the observed trends for both n-type ZrNiSn and p-type NbFeSb can be well explained by our electron-dopant interaction picture (Fig. 1a). For n-type materials, dopants with smaller ionic radius create more attractive short-range potentials, which add to the attractive Coulomb potential and increases the total defect scatterings. Therefore, the mobility generally decreases as the dopant's ionic radius becomes smaller, as observed in Fig. R4a. On the other hand, for p-type materials, the Coulomb potential is now repulsive. As a result, dopants with smaller ionic radius that create attractive short-range potentials will counteract the Coulomb field, leading to cancellation effect and smaller total defect scatterings. Thus, for p-type materials, the mobility instead increases as the dopant's ionic radius becomes smaller, an opposite trend to that of n-type materials. This is consistent with the trend observed in p-type NbFeSb (Fig. R4b). This trend is also shown in the power factors with respect to different dopants (Fig. R4c, where we have re-plotted Fig. 4f and organize the power factors based on the ionic radius difference), for which our simulation matches well with the experimental trend.

In summary, we show that the mobility dependence on carrier concentration as well as dopant choice cannot be explained based on the conventional theory, but instead can be explained based on our electron-dopant interaction picture. Furthermore, the observed trend of mobility with dopants (Fig. R4b) provides another evidence supporting the existence of partial electron cloaking effect. Together with the first two points, we hope that the reviewer will find our analysis convincing and our work suitable for the publication.

Revision: To further validate the ability of our computational method to differentiate the impact on electron transport between different dopants, we have included additional simulation results on SrTiO₃. Figure R3 is added into the supplementary materials, and corresponding discussions are added in the main text:

“Mobility variation with dopants has also been observed in other materials, e.g. in SrTiO₃, an oxide with perovskite structure. The computed defect potentials and corresponding electron transport properties for lanthanum (La) doping (on Sr site) and niobium (Nb) doping (on Ti site) are shown in Supplementary Fig. 4. In order to correctly describe the phonon modes in SrTiO₃ with perovskite phase, *ab initio* molecular dynamics was used to extract effective force constants at finite temperature^{9,10}. Neither dopant creates significant repulsive potentials that can counteract the Coulomb field. Instead, Nb dopant creates a strong attractive short-range potential that further enhances the electron-defect scatterings. As a result, the computed mobility for Nb-doped SrTiO₃ is lower than La-doped one. We caution that the computed mobilities are not accurate due to the use of quasiparticle picture in Boltzmann transport theory⁷, which ignores the polaron nature of charge transport in SrTiO₃. Nonetheless, the general trend of mobility with different dopants (La and Nb) agrees with experiments⁸ (Supplementary Fig. 4e). The above examples confirmed the electron scattering scenario due to strong short-range electron-defect interactions and further demonstrate the ability of our computational approach to distinguish the impact of different charged defects on electron transport.”

In the main text we have also revised Figure 3 to include the compilation of mobility data with different dopants based on their ionic radius:

Revised Figure 3. Defect-mediated electrical and thermoelectric transport. **d-e**, Compiled highest mobility data from past studies with respect to the ionic radius difference between the dopant and host atoms for **(d)** n-type ZrNiSn and **(e)** p-type NbFeSb. The ionic radius differences are taken from **(a)**. The shaded regions indicate the standard deviation of these extracted mobility data. **h**, Comparison between simulations and experimental results¹¹ for the optimal room-temperature thermoelectric power factor in p-type NbFeSb with respect to the ionic radius difference between dopant and host atoms. The trend of power factor with respect to the ionic radius agrees between the experiments and the simulations.

and further included discussions about how this trend of mobility with different dopants can be explained based on our electron-dopant interaction picture, for both n-type ZrNiSn

“In Fig. 3c, we further show highest mobility values for V, Nb and Ta doping from past work with respect to the ionic radius difference between dopant and host ($r_{dopant} - r_{host}$, based on Fig. 3a). A general trend of increasing mobility with increasing dopant ionic radius is observed. This is consistent with our defect scattering picture (Fig. 1a), because for n-type materials a dopant with larger ionic radius is less attractive to electrons and therefore contributes less to total electron-defect scatterings.”

and p-type NbFeSb

“The general observation of mobility peaks in NbFeSb suggests electron cloaking effect likely exists. Another evidence supporting electron cloaking effect is shown in Fig. 3e, which plots the highest mobility values from past work for different dopants with respect to the ionic radius difference. In contrast to the case with ZrNiSn, dopant with the largest ionic radius difference from the host (Ti dopant on Nb) has highest mobility, consistent throughout many studies. This contradicts simple defect scattering picture, which would suggest such dopants should create

strongest short-range electron-dopant scattering and decrease the mobility. However, because small ionic radius dopants actually create attractive short-range potential that counteracts the repulsive Coulomb field in p-type materials, the overall electron-defect scattering should decrease (and correspondingly the mobility increases) as the dopant's ionic radius becomes smaller, as observed in Fig. 3e. The electron dopant interaction picture based on the ionic radius is consistent with both n-type and p-type materials (Fig. 3c,e).”

Reviewer #2:

In this paper, the authors propose that the nature of the dopant could influence the defect scattering process. Depending on the dopant size, mobility could be even increased due to a dopant cloaking effect.

This is an interesting concept and the theoretical analysis appears sound (with some open questions though, see further). However, for publication in a high impact journal such as Nature Communications, the authors should further demonstrate that the theoretical effect they suggest is indeed observed experimentally and that this effect is important. More specifically, while their model can be tuned to inject parameters leading to a large effect (as in Fig. 1 which only uses hypothetical parameters), it remains to be shown that real materials have such a peak in mobility vs carrier concentration and that this peak is large enough to be of practical importance. The examples provided especially on the heussler are not convincing.

Response: First, we thank the reviewer for his/her positive comments on our work. Here, we would like to provide additional evidence to show that the electron-cloaking effect is indeed important in understanding the mobility in doped semiconductors.

In Fig. R5a-b below, we have collected previously reported highest mobility values from different studies, and organized them based on the dopants they use. Figure R5a shows n-type ZrNiSn doped with V, Ta, or Nb (all on Zr site), and Fig. R5b shows p-type NbFeSb doped with Ti, Hf, or Zr (all on Nb site). The mobilities for different dopants are plotted against the ionic radius difference between the dopant and the host atom (the ionic radius difference is taken from Fig. 3a). Conventional theory for defect scattering would suggest that the central cell scattering due to the short-range defect potential should be minimized to favor high mobility. In other words, dopants with small ionic radius difference from the host atom are most desirable. This agrees with what is observed in n-type ZrNiSn (Fig. R5a), but not with p-type NbFeSb (Fig. R5b, the ionic radius difference between Ti dopant and Nb in fact is the largest).

Figure R5. Trend of electron mobility and power factor with respect to ionic radius. a-b, Compiled highest mobility data from past studies with respect to the ionic radius difference between the dopant and host atoms for (a) n-type ZrNiSn and (b) p-type NbFeSb. The dopants are labeled inside the figures with arrows pointing to the mobility data. The ionic radius differences are taken Figure 3a. **c,** Comparison between simulation and experiment for the optimal room-temperature power factor in p-type NbFeSb with respect to the ionic radius difference between the dopant and host atoms.

In contrast, the observed trends for both n-type ZrNiSn and p-type NbFeSb can be explained well based on our electron-dopant interaction picture (Fig. 1a). For n-type materials, dopants with smaller ionic radius create more attractive short-range potentials, which add to the attractive Coulomb potential and increases the total defect scatterings. Therefore, the mobility generally decreases as the dopant's ionic radius becomes smaller (Fig. R5a). On the other hand, for p-type materials, the Coulomb potential is now repulsive. As a result, dopants with smaller ionic radius will create attractive short-range potentials, which counteract the Coulomb field, leading to the cancellation effect and smaller total defect scatterings. Thus, for p-type materials, the mobility instead increases as the dopant's ionic radius becomes smaller, an opposite trend to that of n-type materials. This is consistent with the trend observed in p-type NbFeSb (Fig. R5b). This trend is also further corroborated by the power factor trend with respect to different dopants (Fig. R5c, where we have re-plotted Fig. 4f and organize the power factors based on the ionic radius difference). The mobility and power factor variation with dopants thus provide another evidence that support the existence of electron cloaking effect in practical thermoelectric materials.

While we believe the electron-cloaking picture is important in understanding the mobility peaks and the mobility variation with dopants (Fig. R5), we do acknowledge that the mobility enhancement we observe at room temperature is small, only about 20% (e.g. Ti-doped NbFeSb compared to Zr-doped NbFeSb). However, we want to mention two examples where this effect can have more significant impact. First, thermoelectric figure of merit is directly proportional to the power factor. As we have seen, the optimal power factor trend in p-type NbFeSb follows the electron-cloaking picture (Fig. R5c). Considering that improving thermoelectric figure of merit has been an extremely challenging task, such improvement in power factor will be highly beneficial for finding more efficient thermoelectric materials. Second, the electron-cloaking effect can be more significant at lower temperatures, when dopant scatterings become stronger while intrinsic electron-phonon scatterings become weaker. In Fig. R6, we show that Ti dopant in NbFeSb and TaFeSb can lead to significant enhancement in power factor by as much as 80% (80% for NbFeSb, and 70% for TaFeSb) due to the cloaking effect of dopants. This provides another opportunity for developing efficient thermoelectric materials for low-temperature cooling and refrigeration applications. Above all, we believe the electron cloaking effect is important in understanding charge transport of practical doped materials, and also have impacts on improving thermoelectric device efficiencies, particularly for low-temperature applications.

Figure R6. Thermoelectric power factor at 150 K. a, p-type Ti-doped NbFeSb. **b,** p-type Ti-doped TaFeSb. For both plots, the red curve only considers electron scatterings by phonons and by the Coulomb potential, while the blue curve considers electron scatterings by phonons and the full defect potential (including the short-range part). Due to the counteraction between the strong central cell potential of the Ti dopant and the Coulomb potential, a larger enhancement in power factor is observed at lower temperatures.

Revision: We have revised Figure 3 in the main text to include the compiled mobility data with respect to the dopant's ionic radius, as shown below

Revised Figure 3. Defect-mediated electrical and thermoelectric transport. d-e, Compiled highest mobility data from past studies with respect to the ionic radius difference between the dopant and host atoms for **(d)** n-type ZrNiSn and **(e)** p-type NbFeSb. The ionic radius differences are taken from **(a)**. **h,** Comparison between simulations and experimental results¹¹ for the optimal room-temperature thermoelectric power factor in p-type NbFeSb with respect to the ionic radius difference between dopant and host atoms. The trend of power factor with respect to the ionic radius agrees between the experiments and the simulations.

and have included additional discussions about the mobility variation with dopants for n-type ZrNiSn:

“In Fig. 3c, we further show highest mobility values for V, Nb and Ta doping from past work with respect to the ionic radius difference between dopant and host ($r_{dopant} - r_{host}$, based on Fig. 3a). A general trend of increasing mobility with increasing dopant ionic radius is observed. This is consistent with our defect scattering picture (Fig. 1a), because for n-type materials a dopant with larger ionic radius is less attractive to electrons and therefore contributes less to total electron-defect scatterings.”

and for p-type NbFeSb:

“The general observation of mobility peaks in NbFeSb suggests electron cloaking effect likely exists. Another evidence supporting electron cloaking effect is shown in Fig. 3e, which plots the highest mobility values from past work for different dopants with respect to the ionic radius difference. In contrast to the case with ZrNiSn, dopant with the largest ionic radius difference from the host (Ti dopant on Nb) has highest mobility, consistent throughout many studies. This contradicts simple defect scattering picture, which would suggest such dopants should create strongest short-range electron-dopant scattering and decrease the mobility. However, because small ionic radius dopants actually create attractive short-range potential that counteracts the repulsive Coulomb field in p-type materials, the overall electron-defect scattering should decrease (and correspondingly the mobility increases) as the dopant’s ionic radius becomes smaller, as observed in Fig. 3e. The electron dopant interaction picture based on the ionic radius is consistent with both n-type and p-type materials (Fig. 3c,e).”

We also added discussion about the beneficial role the electron cloaking effect can play in improving thermoelectric efficiency:

“While these enhancements may not seem large, the thermoelectric figure of merit zT is directly proportional to the power factor. In this regard, rational dopant selection that improves the charge mobility and power factor will be beneficial for the thermoelectric efficiency, whose improvement has been a challenging task.”

Further, we have added Figure R6 as Supplementary Figure 8 in supplementary materials and discussed the larger enhancement in power factor at lower temperature in the main text:

“Our computation shows that Ti dopant in NbFeSb and TaFeSb can potentially lead to significant enhancement of the power factor by as much as 80% at 150 K compared to the conventional Coulomb-limited case due to the cloaking effect of the dopant (Supplementary Fig. 8).”

1) the absence of a peak in mobility for ZnNiSn vs NbFeSb which shows one stressed by the authors appears to be inconsistent with previous work such as refs 22 which shows also a peak in ZnNiSn.

Response: We would like to first explain the possible causes that can lead to a mobility peak, and the discrepancy between data shown in this work and those in Ref 22 for ZrNiSn. Usually, mobility

would decrease with carrier concentration for scattering mechanisms including acoustic/optical phonon deformation scattering and ionized impurity scattering. An intrinsic mechanism that can lead to increased mobility with carrier concentration is the screening effect of polar optical phonon scattering, as discussed in Ref 22. However, practical thermoelectric materials may also have other types of defects besides the dopants, such as compensated charged defects or charged vacancies, which can have varying degree of ionization when the Fermi level changes. It has been shown that extrinsic defects can alter the electron transport and lead to peaks in the mobility with carrier concentration [e.g. reference Xie et al (*Scientific Reports* 4, 6888 (2014)) discussed the case of alloy scattering in ZrNiSn where the mobility also shows a peak]. Therefore, while looking for the experimental signatures of electron cloaking, we have focused on materials with largest reported mobility (for each dopant element) because they should have minimal *extrinsic* defects and represent the closest example to an intrinsic semiconductor. Nonetheless, we have revised Figure 3 in the main text to include data from reference Xie et al.¹² and accordingly added discussions about other possible mechanisms that can lead to a mobility peak.

Revision: Figure 3 in the main text has been revised to include other data for n-type ZrNiSn (see revised Figure 3 in the response above). We also revised the discussion in the main text:

“As a result, the compiled experimental mobility data mostly show a monotonic decreasing trend with increasing carrier concentration (Fig. 3b). The non-monotonic trend in one data has been attributed to other extrinsic defect scatterings and is not directly related to dopants¹².”

In addition, we added discussions about other possible mechanisms that can lead to mobility peaks with respect to the carrier concentration:

“However, we acknowledge that other extrinsic effects, such as existence of compensated charged defects, may also lead to non-monotonic mobility variation. Another mechanism that is responsible for mobility peaks is the screening of polar optical phonon scattering, an intrinsic scattering mechanism¹³. Still, if the scattering from the dopant is strong, the mobility would instead be limited by electron-defect interactions and a monotonically decreasing trend would be expected. The general observation of mobility peaks in NbFeSb suggests electron cloaking effect likely exists.”

2) the cloaking effect is arguably quite marginal in these heusslers. This appears to change the power factor by around 10% only which is interesting but not very impactful.

Response: We acknowledge that the enhancements in mobility and power factor observed in NbFeSb are not particularly large, but would like to mention that these are still beneficial for improving the thermoelectric materials because the strategy is general and may potentially be applied to other systems to create larger enhancement. Nonetheless, here we provide another example where the cloaking effect can be much more significant (also discussed in the response above). The defect scattering generally becomes more significant compared to the intrinsic electron-phonon scatterings as the temperature decreases. Therefore, at lower temperatures, rational selection of dopants becomes even more important. We have simulated Ti-doped NbFeSb and TaFeSb (results are given in Fig. R6) and found that Ti dopant leads to enhancement in power factor by as much as 80% at 150 K compared to the case that is limited by Coulomb scattering. Such enhancement shows that the electron cloaking strategy can be a potentially impactful

approach to improve thermoelectric materials for low-temperature cooling and refrigeration applications.

Revision: We have added Fig. R6 into the supplementary materials, and included discussions about the larger enhancement in power factor at lower temperatures, as mentioned above.

3) is there any other explanation of a peak in mobility with carrier concentration. This should be clearly discussed.

Response: Other explanations include defect scatterings due to extrinsic defects, such as compensated charged defects, which can have varying degree of ionization when the Fermi level changes, or other types of defects such as interstitial impurities, which we have discussed in our responses to question 1 above.

I would suggest the authors to revise their manuscript and maybe choose another system where the effect could be more pronounced. Materials with higher dielectric constant could be a good target. There is for instance plenty of data for mobility on SrTiO₃ arguably on better single crystal sample which should lead to an important cloaking effect in view of its large dielectric constant.

Response: We thank the reviewer for his/her suggestion. Besides the revision we mentioned above, we have performed additional calculations for SrTiO₃, as shown in Fig. R7. Indeed, as the reviewer mentioned, materials with high dielectric constants may host larger electron cloaking effect. Experimentally, it was also found that mobilities in SrTiO₃ are strongly affected by the dopants. Among these dopants, La doping on Sr site and Nb doping on Ti site are the most well documented ones.

Our calculations suggest that La doped SrTiO₃ has a higher mobility than Nb-doped SrTiO₃, consistent with experimental findings (Fig. R7e). However, neither dopant shows electron cloaking effects. Instead, Nb dopant on Sr site creates a strong attractive potential (Fig. R7c) that further adds to the attractive Coulomb potential and increases the total defect scatterings, which explains the lower mobility of Nb-doped SrTiO₃ compared to La-doped one seen in experiments. In order to simulate SrTiO₃, *ab initio* molecular dynamics was used to extract effective force constants at finite temperature (Fig. R7a), otherwise one would obtain imaginary phonon modes because the perovskite structure is not stable at 0 K. Though our electron transport modeling based on Boltzmann transport equation is not accurate for SrTiO₃ because it ignores the polaron nature of charge transport in SrTiO₃⁷, the general trend of mobility with dopants is reproduced (Fig. R7e). At the least, this example demonstrates the ability of our computational method to distinguish between different dopants, and will be beneficial for researchers studying electron transport in doped semiconductors.

Figure R7. Electron transport simulation of SrTiO₃. **a**, Computed phonon dispersion of SrTiO₃. Dashed lines are results from density functional perturbation theory, which gives rise to imaginary phonon frequencies. The solid lines are obtained by fitting force constants to a force-displacement dataset from *ab initio* molecular dynamics study. The latter method correctly reproduces the stable phonon modes at 300 K. **b-c**, Defect potentials for **(b)** La dopant on Sr site, and **(c)** Nb dopant on Ti site. **d**, Comparison of electron-defect scattering rates for La and Nb dopants and the intrinsic electron-phonon scattering rates. **e**, Mobility considering different scattering conditions, and comparison between simulation and experiment⁸. The calculated mobilities assume a carrier concentration of $6 \times 10^{19} \text{ cm}^{-3}$.

Revision: We have included additional simulation results on SrTiO₃ and added Figure R7 into the supplementary materials as a Supplementary Figure, and also added corresponding discussions in the main text:

“Mobility variation with dopants has also been observed in other materials, e.g. in SrTiO₃, an oxide with perovskite structure. The computed defect potentials and corresponding electron transport properties for lanthanum (La) doping (on Sr site) and niobium (Nb) doping (on Ti site) are shown in Supplementary Fig. 4. In order to correctly describe the phonon modes in SrTiO₃ with perovskite phase, *ab initio* molecular dynamics was used to extract effective force constants at finite temperature^{9,10}. Neither dopant creates significant repulsive potentials that can counteract the Coulomb field. Instead, Nb dopant creates a strong attractive short-range potential that further enhances the electron-defect scatterings. As a result, the computed mobility for Nb-doped SrTiO₃ is lower than La-doped one. We caution that the computed mobilities are not accurate due to the use of quasiparticle picture in Boltzmann transport theory⁷, which ignores the polaron nature of charge transport in SrTiO₃. Nonetheless, the general trend of mobility with different dopants (La and Nb) agrees with experiments⁸ (Supplementary Fig. 4e). The above examples confirmed the electron scattering scenario due to strong short-range electron-defect interactions and further demonstrate the ability of our computational approach to distinguish the impact of different charged defects on electron transport.”

I also a few more technical comments to make:

-the authors stress the ionic radius effect. Does electronegativity play a role too or is electronegativity correlated with ionic radius here?

Response: The electron-defect scattering is affected by both the extent (or the range) of the defect potential and the strength of the defect potential. The former correlates more with the ionic radius,

while the latter is associated more with the electronegativity (ionic radius also plays a role here). However, ionic radius and electronegativity are connected quantities. Here we have mainly used ionic radius because we found it is a better indicator for whether the dopants tend to cloak electrons or scatter electrons (namely, the sign of the electron-defect scattering matrix).

-in the model do you have to consider the full (ionic + electronic) or only the electronic part of the dielectric constant and why?

Response: The full (ionic + electronic) dielectric constant needs to be used when evaluating the ionized impurity scattering based on the screened Coulomb potential, as in the conventional treatment of Brooks-Herring model. This is also used in our work to recover the long-range part of the full defect potential from the short-range perturbations.

-the authors mention "many oxide and thermoelectric materials also possess a high dielectric constant associated with soft phonon modes". Could they quantify this statement? If high dielectric constant materials are common (and what is "high" exactly). Their effect should be common? Maybe using a database of material property (computed or experimental) could help quantifying how many high dielectric constant materials there are. This is important to evaluate if the effect is truly common.

Response: The white regions in Fig. 2a indicate the approximate range of carrier concentration when central cell scattering is comparable to Coulomb scatterings. Though this would depend on the actual defect potential strength, this shows that at least certain level of carrier density is needed such that the central cell effect can potentially compete with Coulomb scattering. Based on this plot, dielectric constants above 30 can be regarded 'high' in our study, because in such case within a carrier concentration of 10^{20} cm^{-3} the central cell scattering would be already comparable to Coulomb scattering, and the sign of central cell potential then becomes important in determining the actual total defect scatterings.

In Table R1 below, we collected dielectric constants data from database and literature for representative oxides and thermoelectric materials. These materials, including some of the best thermoelectric compounds, all have dielectric constants close to or above 30. Therefore, we believe that central cell scattering is no longer negligible in these materials and understanding the dopant selection for their charge mobility is important.

Table 1. Dielectric constants of representative oxides and thermoelectric materials.

Material	Dielectric constant (relative)
ZrO ₂	29 ¹⁴
HfO ₂	25 ¹⁴
Ta ₂ O ₅	26 ¹⁴
La ₂ O ₃	30 ¹⁴
LaAlO ₃	30 ¹⁴
Nb ₂ O ₅	35 ¹⁴
TiO ₂	95 ¹⁴

Bi_2Te_3	$290 (\parallel, 15\text{K})^{15}$
PbTe	414^{15}
SnSe	$42 (\text{c axis}), 45 (\text{a axis})^{15}$
NbFeSb	45^{16}
Mg_3Sb_2	32^{16}

Revision: The above table has been included as Supplementary Table 1 in the supplementary materials, and has been cited in the main text.

-I would recommend that the authors plot the materials they study (GaAs, PbTe and heuslers and/or others if they follow my suggestion) in the Fig. 2a

Response: Following the reviewer's suggestion, we have labelled GaAs, PbTe and half-Heuslers in corresponding regions in Fig. 2a. We have also labeled the newly computed SrTiO_3 as well. We note that because PbTe and SrTiO_3 both have large dielectric constants that lie outside the dielectric range of Fig. 2a, they are labelled with arrows indicating their locations are at higher dielectric constant range.

Revision: We have revised Figure 2a as follows.

-why do the authors have a factor 1/3 in their eq. (4)?

Response: This is because we use v to denote the magnitude of the full velocity, while the conductivity is measured only along one direction. As half-Heuslers are isotropic materials, the conductivities along three directions are the same and therefore a factor of 1/3 appears. We have clarified this definition in the revised manuscript.

Revision: We have added a sentence in the Methods, Electron transport calculation section for clarification:

“Isotropic materials are assumed and the factor 1/3 appears because the conductivity is the same in all three directions.”

- "computed with DFT is vague" what functional? I know this is in SI but that's worth being in the main text

Response: For electron transport as well as defect potential calculations, we use the generalized gradient approximation (GGA) of Perdew, Burke and Ernzerhof¹⁷ with the Troullier-Martins-type norm-conserving semilocal pseudopotential. We have revised the Methods section and included this information.

Revision: In Methods, Electron transport calculation section, we added

“The equilibrium properties of electrons are calculated from first principles using the QUANTUM ESPRESSO software package¹⁸. We use the generalized gradient approximation (GGA) of Perdew, Burke and Ernzerhof¹⁷ with the Troullier-Martins-type norm-conserving semilocal pseudopotential (corresponding to pbe-mt.UPF in the QUANTUM ESPRESSO pseudopotential library). A cutoff energy of 120 Ryd and a $6 \times 6 \times 6$ **k**-mesh are used to determine the equilibrium lattice constant. The equilibrium properties of phonons and the electron-phonon interaction matrices are calculated *via* density functional perturbation theory¹⁹ for a $6 \times 6 \times 6$ **q**-mesh (with a $6 \times 6 \times 6$ **k**-mesh for the electron-phonon interaction matrix). We then use the EPW software package²⁰ to interpolate the electronic information and the phonon information, as well as the electron-phonon coupling matrices to a fine mesh *via* the Wannier interpolation method²¹.”

-related to my general comments on hypothetical parameters, why did you choose the given value for r_0 and V_0 for the model in Fig. 1 (see line 266)?

Response: The r_0 is chosen based on our general observation from first principles calculated defect potentials, which showed that the short-range potential usually extends about a few Angstroms. The V_0 is chosen to maximize the electron cloaking effect at the carrier concentration about $6 \times 10^{19} \text{ cm}^{-3}$. Because the electron-defect scattering ultimately is determined by the absolute phase shift, this V_0 corresponds to a phase shift that is close to zero for low-energy electrons (see panel e in revised Fig. 1), which lead to small total defect scattering rates. This V_0 was chosen to illustrate the enhancement the short-range defect potential could have on electron mobility if ideal dopants exist for a given semiconductor material.

-line 189: Where is the point for Zr:Nb (Fig3a) for ZnNiSn? There is a point for Nb:Zr but not for the same material

Response: We thank the reviewer for pointing out this typo. We have corrected it in the revised Figure 3.

-As they reach very large doping the dielectric constant should be affected. The dielectric constant of a metal is infinite. Do the authors take this into account? And if not, why?

Response: This effect has been considered in our simulation. The effect of increasing doping level on the dielectric response of a doped semiconductor is due to the additional screening effect from the free carriers. This adds an additional exponentially decaying term to the Coulomb potential

with a Debye screening length²², which is included in our modeling for the long-range part of charged defect scattering (section Electron transport calculation in Methods). This is equivalent to an effective dielectric constant of the form²³: $\epsilon_{eff}(k) = \epsilon(k) + \epsilon(0)/(R_D^2 k^2)$, where k is the spatial wave vector, $\epsilon(k)$ is the dielectric constant of the undoped semiconductor, and R_D is the Debye screening length. For high doping level (small Debye screening lengths), the effective dielectric constant for small k diverges like k^{-2} , as expected in a conductor.

-the difference of the effect on p-type or n-type material could be better explained so the discussion on the heusler is easier to follow.

Response: We thank the reviewer for his/her suggestion. In the revised version we have included more discussions on the difference between n-type and p-type materials to illustrate the impact of defects on electron transport.

Revision: Following the discussion on the electron scattering/cloaking in n-type materials, we added a paragraph to explain how the electron-dopant interaction in p-type material differs from that in n-type material:

“The above argument can also be applied to p-type materials, except that the Coulomb potential now becomes repulsive to electrons. As a result, the desired dopants that counteract the Coulomb scattering need to have attractive short-range potentials. The relation between ionic radius and short-range potentials remains the same. Therefore, dopants with smaller ionic radius are more desired for p-type materials (Supplementary Fig. 1).”

Reference

1. Liao, B., Zebarjadi, M., Esfarjani, K. & Chen, G. Cloaking core-shell nanoparticles from conducting electrons in solids. *Phys. Rev. Lett.* **109**, 126806 (2012).
2. Cohen-Tannoudji, C., Diu, B. & Laloe, F. *Quantum Mechanics*. (Wiley, 1992).
3. Ghosh, D. C. & Biswas, R. Theoretical calculation of absolute radii of atoms and ions. Part 2. The ionic radii. *International Journal of Molecular Sciences* **4**, 379–407 (2003).
4. Rurali, R., Markussen, T., Suñé, J., Brandbyge, M. & Jauho, A.-P. Modeling transport in ultrathin Si nanowires: Charged versus neutral impurities. *Nano Lett.* **8**, 2825–2828 (2008).
5. Freysoldt, C., Neugebauer, J. & Van de Walle, C. G. Fully ab initio finite-size corrections for charged-defect supercell calculations. *Phys. Rev. Lett.* **102**, 016402 (2009).
6. Martin, R. M. *Electronic Structure: Basic Theory and Practical Methods*. (Cambridge University Press, 2004).
7. Zhou, J.-J. & Bernardi, M. Predicting charge transport in the presence of polarons: The beyond-quasiparticle regime in SrTiO₃. *Phys. Rev. Research* **1**, 033138 (2019).
8. Han, W. *et al.* Spin injection and detection in lanthanum- and niobium-doped SrTiO₃ using the Hanle technique. *Nat Commun* **4**, 2134 (2013).
9. Hellman, O., Steneteg, P., Abrikosov, I. A. & Simak, S. I. Temperature dependent effective potential method for accurate free energy calculations of solids. *Phys. Rev. B* **87**, 104111 (2013).
10. Tadano, T., Gohda, Y. & Tsuneyuki, S. Anharmonic force constants extracted from first-principles molecular dynamics: applications to heat transfer simulations. *J. Phys.: Condens. Matter* **26**, 225402 (2014).
11. Ren, W. *et al.* Ultrahigh power factor in thermoelectric system Nb_{0.95}M_{0.05}FeSb (M = Hf, Zr, and Ti). *Advanced Science* **5**, 1800278 (2018).
12. Xie, H. *et al.* The intrinsic disorder related alloy scattering in ZrNiSn half-Heusler thermoelectric materials. *Scientific Reports* **4**, 6888 (2014).
13. Ren, Q. *et al.* Establishing the carrier scattering phase diagram for ZrNiSn-based half-Heusler thermoelectric materials. *Nature Communications* **11**, 3142 (2020).
14. Azadmanjiri, J. *et al.* A review on hybrid nanolaminate materials synthesized by deposition techniques for energy storage applications. *J. Mater. Chem. A* **2**, 3695–3708 (2014).
15. *Non-Tetrahedrally Bonded Elements and Binary Compounds I*. vol. 41C (Springer-Verlag, 1998).
16. J. Slade, T. *et al.* Understanding the thermally activated charge transport in NaPb_mSbQ_{m+2} (Q = S, Se, Te) thermoelectrics: weak dielectric screening leads to grain boundary dominated charge carrier scattering. *Energy & Environmental Science* **13**, 1509–1518 (2020).
17. Perdew, J. P., Burke, K. & Ernzerhof, M. Generalized gradient approximation made simple. *Phys. Rev. Lett.* **77**, 3865–3868 (1996).
18. Giannozzi, P. *et al.* QUANTUM ESPRESSO: a modular and open-source software project for quantum simulations of materials. *J. Phys.: Condens. Matter* **21**, 395502 (2009).
19. Baroni, S., de Gironcoli, S., Dal Corso, A. & Giannozzi, P. Phonons and related crystal properties from density-functional perturbation theory. *Rev. Mod. Phys.* **73**, 515–562 (2001).

20. Giustino, F., Cohen, M. & Louie, S. Electron-phonon interaction using Wannier functions. *Phys. Rev. B* **76**, 165108 (2007).
21. Marzari, N., Mostofi, A. A., Yates, J. R., Souza, I. & Vanderbilt, D. Maximally localized Wannier functions: Theory and applications. *Rev. Mod. Phys.* **84**, 1419–1475 (2012).
22. Lundstrom, M. *Fundamentals of Carrier Transport*. (Cambridge University Press, 2009).
23. Resta, R. Dielectric behavior of a doped semiconductor. *Phys. Rev. B* **19**, 3022–3026 (1979).

Mobility enhancement in heavily doped semiconductors *via* electron cloaking

Jiawei Zhou^{1*}, Hangtian Zhu², Qichen Song¹, Zhiwei Ding¹, Jun Mao², Zhifeng Ren², Gang Chen^{1*}

¹*Department of Mechanical Engineering, Massachusetts Institute of Technology, Cambridge, MA 02139, USA*

²*Department of Physics and Texas Center for Superconductivity at the University of Houston (TcSUH), University of Houston, Houston, TX 77204, USA*

Authors to whom correspondence should be addressed.

* Electronic mail: jwzhou@stanford.edu, gchen2@mit.edu

Doping is central for solid-state devices from transistors to thermoelectric energy converters^{1,2}. The interaction between electrons and dopants plays a pivotal role in carrier transport^{3,4}. Conventional theory suggests that the Coulomb field of the ionized dopants^{5,6} limits the charge mobility at high carrier densities, and that either the atomic details of the dopants are unimportant or the mobility can only be further degraded, while experimental results often show that dopant choice affects mobility. In practice, the selection of dopants is still mostly a trial-and-error process. Here we demonstrate, *via* first-principles simulation and comparison with experiments, that a large short-range perturbation created by selected dopants can in fact counteract the long-range Coulomb field, leading to electron transport that is nearly immune to the presence of dopants. Such “cloaking” of dopants leads to enhanced mobilities at high carrier concentrations close to the intrinsic electron-phonon scattering limit. We show that the ionic radius can be used to guide dopant selection in order to achieve such an electron-cloaking effect. Our finding provides guidance to the selection of dopants for solid-state conductors to achieve high mobility for electronic, photonic, and energy conversion applications.

Doping is a fundamental strategy employed to control the electrical conductivity of semiconductors¹. Conventional understanding based on the theory originally proposed by Brooks and Herring^{5,6} states that electrons are strongly scattered by the long-range Coulomb field of the charged dopant, leading to reduced mobility. With further generalization to consider multiple scatterings, electron-electron interactions, and dielectric screening^{3,7,8}, the Brooks-Herring theory has been successfully used to explain the reduced charge mobility in conventional semiconductors such as silicon and III-V semiconductors at low to intermediate doping concentrations. An

important consequence of this theory is that different dopants with the same charge have the same impact on the electron transport, as the theory neglects the atomic details³. While the effect of a dopant's chemical nature has been recognized in the past, most models have assumed an empirical potential profile and treated such a chemical effect as a perturbation to the Coulomb field, as indicated by the "central cell correction"⁹⁻¹¹. A prevailing view is that such correction could only further reduce charge mobility due to the strong local interactions between electrons and defects. Experimentally, different impacts on carrier mobility resulting from different dopants are often observed^{9,12,13}, suggesting that their atomic details play a significant role in governing the electron-defect interactions. The impact of dopant scattering on mobility can be particularly large at high carrier concentrations, as is often observed in solid oxide materials¹⁴, transparent conductors¹⁵, and thermoelectric compounds¹⁶, with a carrier concentration close to or above $\sim 10^{20} \text{ cm}^{-3}$. Despite the crucial role of dopants in governing the carrier transport, dopant selection has thus far been mostly a trial-and-error process.

Recent advancements in *ab initio* simulations have enabled quantification of electron energy or potential changes induced by charged defects, allowing the engineering of defects from first principles¹⁷⁻¹⁹. These studies have revealed that a charged defect potential can significantly deviate from the conventional Coulomb field assumption¹⁷⁻¹⁹. However, how such atomic details impact charge mobility remains unknown, mainly due to the lack of capability to treat the long-range Coulomb field and short-range electron interactions on an equal footing. Here we employ a new computational approach to treat Coulomb and short-range interactions simultaneously and apply it to heavily doped semiconductors. Building on the recent development in the formation energy calculations for charged defects⁴, we take into account short-range perturbations from *ab initio* calculations while incorporating the long-range potential to far distances *via* analytic

expression. Our approach overcomes the obstacles that have previously prevented direct quantification of electron interactions with charged defects, allowing us to examine the impact of short-range interactions on electron transport. We discovered that the deviation of defect potential from the Coulomb field at short range can lead to strong electron-defect interactions, particularly at high carrier concentrations. We further demonstrate, in contrast to the conventional belief that charge mobility in extrinsic semiconductors is limited by Coulomb scatterings, that the chemical nature of dopants can be harnessed to break this barrier, leading to effective electron cloaking and enhanced mobility close to the intrinsic electron-phonon scattering limit. While modeling suggested that a core-shell nanoparticle with proper potential can be cloaked²⁰, we found here that the central cell effect of selected dopants allows them to actually cloak themselves, achieving electron cloaking *via* doping. We show that ionic radius can guide the choice of proper dopants to realize the cloaking effect, hence providing direction in the dopant selection to achieve high mobility.

Electron-defect interaction

Central to the electron-defect interactions is the perturbed electronic potential $\Delta\hat{V}$ resulting from the presence of defects. Representative defect potential profiles are shown in Fig. 1a, in which the potential at longer distances can be approximated by the Coulomb field of a point charge (here we use a representative n-type dopant, which gives rise to an attractive Coulomb potential, as an example), while deviations occur close to the defect (represented by a simplified rectangular profile). Two scenarios, corresponding to attractive and repulsive short-range potentials, are depicted. Such short-range deviations are traditionally treated using empirical potentials, known as central cell correction^{3,9-11}, to investigate their effect on electron dynamics. It has been found that the dominant electron-defect interactions are due to the long-range Coulomb field and the

atomic details mostly introduce perturbations to the electron binding energy or scattering rates^{3,9-11}, which only further degrades the mobility. The central cell effects are usually weak, and have not been harnessed to help achieve high electron mobility.

However, if a large central cell potential exists and has a sign opposite to that of the Coulomb potential, it could counteract the scatterings due to the Coulomb field and enhance electron mobility. The reason is that the electron defect scattering is governed by the magnitude of the electron-defect interaction (EDI) matrix (to the first order), given by the spatial integration of the defect potential $-\langle \psi_{k'} | \Delta \hat{V} | \psi_k \rangle$, where ψ_k is the electronic wavefunction with wave vector k . Both central cell part and long-range Coulomb part of the defect potential $\Delta \hat{V}$ contributes to the EDI matrix. When the central cell potential has an opposite sign to that of the Coulomb potential, it will lead to cancelling terms in the above spatial integration, and correspondingly a small EDI matrix, which implies weak or negligible electron-defect scattering. Note that some of past literature calculated the scattering rate due to the central cell and the long-range Coulomb contribution separately, and added the two rates according to the Matthiessen's rule. Such a treatment neglects the coherence effect of electron waves. In this treatment, the inclusion of the central cell potential always leads to a higher scattering rate. Conceptually, this is not the correct treatment since it is the net effect from the central cell and the long-range Coulomb interaction that impacts the electron scattering.

It is then clear that the sign of the central cell potential relative to that of Coulomb potential plays a large role in how charged defects scatter electrons. Here we show that the ionic radii of the dopant and host atoms can be used to indicate the sign of the central cell potential (Fig. 1a). The ionic radius here refers to the size of the ion without valence electrons. For example, a silicon atom can be considered as a silicon ion with +4 charge and four electrons, which we denote as $\text{Si}^{+4}[2s2p]$.

Similarly, a phosphorus atom can be denoted as $P^{+5}[2s3p]$. The reason we separate the valence electrons from the core is that valence electrons participate in chemical bonding and will be more strongly affected by the lattice structure, while the core electrons are more localized. When silicon is doped with phosphorus (P), phosphorus atom will lose one electron (which becomes free) and becomes positively charged. The remaining four electrons in the outer shell of P atom would participate in the chemical bonding, just as the valence electrons of Si would do, except that the core potential is different from that of Si. The defect potential (perturbed potential due to replacement of one Si by one P) thus results from the difference in the core potential (P^{+5} compared to Si^{+4}).

The first major difference between the core potential of P^{+5} and Si^{+4} is that their charges differ by one, which lead to a long-range Coulomb field when Si^{+4} is replaced by P^{+5} . In addition, Si^{+4} and P^{+5} have different sizes, and thereby attract the valence electrons to different extents. When the dopant ion has a smaller size compared to the host ion and replaces the host ion, propagating electron states (which are composed of mainly valence electrons) will have deeper penetration into the core, which is equivalent to creating an attractive short-range defect potential (Fig. 1b). This is the case for P-doped silicon. Conversely, if a dopant ion has a larger size than the host ion, it will tend to repel electrons due to Pauli exclusion and leads to an equivalent repulsive short-range potential (Fig. 1c).

Now we consider short-range and long-range parts together for the defect potential. For n-type dopant, the long-range Coulomb potential is attractive. A dopant with smaller ionic radius than the host would tend to create a short-range attractive potential, which adds to the Coulomb part and further increases EDI strength, leading to stronger electron-defect scatterings (electron scattering scenario). In contrast, dopants with larger ionic radius will tend to create repulsive short-

range potential, which counteracts the Coulomb potential. When the cancellation effect is maximal, the defects will appear to have negligible scatterings for electrons, and this realizes electron cloaking (electron cloaking scenario). As seen from the computed scattered electron wavefunction, the probability flux is distorted for the electron scattering scenario (Fig. 1b), while undistorted probability flux is recovered away from the defect for the electron cloaking scenario (Fig. 1c). In short, to achieve electron cloaking effect, dopants with larger ionic radius are favored for n-type materials. In addition to the electron cloaking, the ionic radius provides a useful guidance for understanding and selecting dopants in terms of the electron mobility. Effectively, the ionic radius can be seen as a scale bar for dopant selection: in n-type materials, dopants with larger ionic radius are more desired than those with smaller ionic radius because they are more repulsive to electrons and tend to counteract the Coulomb scatterings (Fig. 1a).

The above argument can also be applied to p-type materials, except that the Coulomb potential now becomes repulsive to electrons. As a result, the desired dopants that counteract the Coulomb scattering need to have attractive short-range potentials. The relation between ionic radius and short-range potentials remains the same. Therefore, dopants with smaller ionic radius are more desired for p-type materials (Supplementary Fig. 1).

To illustrate the consequence of different defect potentials on electron transport, we first study a model semiconductor as an example (see modeling details in Methods). Shown in Fig. 1d-e are scattered electron phase shifts computed for a n-type model semiconductor with different defect potentials. Phase shifts quantify how scattered electron waves differ from the incoming waves, and their magnitudes represent the strength of electron-defect scatterings²¹. In general, attractive potentials lead to negative phase shifts, while repulsive potentials lead to positive phase shifts. When attractive short-range potential coexists with the attractive long-range Coulomb

potential in n-type materials, the phase shifts add up in magnitude (Fig. 1d), leading to increased electron scattering rates and reduced mobility (Fig. 1f,h). In contrast, if the short-range potential is repulsive, it counteracts the attractive long-range Coulomb field and reduces the overall phase shifts (Fig. 1e), decreasing the total electron-defect scattering rates (Fig. 1g). When the defect scatterings become weaker than the intrinsic electron-phonon scatterings, the charge mobility becomes nearly immune to the presence of defects and high mobility can be achieved (Fig. 1i).

Electron scattering due to atomic distortions

Having discussed the effects of different dopants on mobility in the model semiconductor, now we turn to practical materials. Below we will first present simulation results and comparison with experiments corresponding to the electron scattering scenario, and then discuss possible evidences that demonstrate the electron cloaking effect. First, we note that mobility variations due to different dopants become significant when the electron-dopant interactions from the central cell potential are strong and comparable to those from the Coulomb field. To find out the parameter space where dopant selection is more critical, Figure 2a displays the ratio of the characteristic electron scattering rates due to central cell scattering and Coulomb scattering in the model semiconductor. In general, the central cell scattering becomes stronger for materials with larger dielectric constant and at higher carrier concentrations. This is because with larger dielectric constant the magnitude of Coulomb potential becomes relatively smaller and at higher carrier concentrations the Coulomb potential is weakened *via* screening. Conventional semiconductors usually have low to intermediate doping concentrations and small dielectric constant, which together indicate the dominance of the Coulomb potential in governing their electron scatterings and the relative unimportance of the dopant selection (Fig. 2a). In contrast, materials with a high carrier density, *e.g.*, thermoelectrics¹⁶ and solid oxide conductors¹⁴, are likely to be more sensitive

to the central cell potential due to their large carrier concentrations. Among them, many oxide and thermoelectric materials also possess a high dielectric constant associated with soft phonon modes. Thus, for these materials, the central cell effect could be significant and thereby potentially be harnessed to counteract the Coulomb scatterings and break the conventional limit in charge mobility at high carrier densities.

Chalcogenide compounds are a class of materials that have received wide attention due to their potential for optoelectronic, photovoltaic and thermoelectric applications. In particular, the pursuit of high thermoelectric energy conversion efficiency has driven studies of heavily doped chalcogenide semiconductors¹⁶. Experiments have shown that different dopants with the same ionization charge can lead to drastically different mobility values, suggesting strong central cell effects¹³. The significance of the central cell effect can be described by the **short-range** electron-defect interaction (**sEDI**) matrix, $\langle \psi_{\mathbf{k}'} | \Delta \hat{V}_{cent} | \psi_{\mathbf{k}} \rangle$, where $\Delta \hat{V}_{cent}$ is the central cell part of the defect potential. For comparison, we computed the defect potential for both Si-doped GaAs, a conventional semiconductor, and Bi-doped PbTe, a representative chalcogenide material. Significant deviations from the Coulomb potential are seen at short range in both cases (Supplementary Fig. 2). **We note that the planar averaged defect potentials generally vary around 0.1 – 1 eV at the short range, on the same order of magnitude with previous reports^{18,22}.** Nonetheless, because of the strong covalent bond between Ga and As atoms and the larger spread of the charge density, the **sEDI** matrix in GaAs is generally small, as shown in Fig. 2b, which displays a three-dimensional contour colormap of the **sEDI** matrix in the real space, with the shape exhibiting the hybridized *s-p* orbital feature of the electron state. In comparison, PbTe consists of resonantly bonded *p* orbitals, which are sensitive to the defect-induced local distortion that breaks the crystal symmetry. As a result, the **sEDI** matrix is significantly larger (Fig. 2c).

In order to evaluate the electron transport based on EDI, we need to account for the long-range Coulomb potential, which extends beyond the finite supercell calculations. The key observation that enabled our first-principles computation is that the long-range part of the defect potential can be well described by an analytic Coulomb potential profile (Supplementary Note and Supplementary Fig. 3). Therefore, we are able to express the defect potential to far distances and to treat short-range and long-range potentials on an equal footing (see details in Methods). To validate our approach, Fig. 2d-e show comparisons between calculations of electron mobility and experimental results for GaAs and PbTe, respectively, at different carrier densities. For each material, two dopants at different atomic sites are considered. In GaAs, we observed that Si doping reduces the mobility slightly more than Te doping does (Fig. 2d). The dopant effects are generally small and the major scattering is due to the Coulomb potential, consistent with past electron transport studies on conventional semiconductors¹². On the other hand, the two dopants in PbTe have clearly different impacts. Bismuth (Bi) doping greatly reduces the electron mobility while iodine (I) doping is able to maintain a high mobility value (Fig. 2e). This large discrepancy occurs in part because electron states in the conduction band of PbTe are mostly formed by orbitals from Pb, and therefore I doping at the Te site only slightly disturbs the electron state while Bi doping at the Pb site overlaps with the electron states and has a large EDI matrix. In addition, because Bi has a smaller ionic radius than Pb²³, the short-range potential of Bi dopant is attractive and adds to the long-range Coulomb potential in n-type PbTe. Therefore, the large EDI from Bi doping contributes to strong electron-defect scattering, leading to decreased mobility as seen in Fig. 2e. For both GaAs and PbTe, good agreement between simulation and experiment has been achieved.

Mobility variation with dopants has also been observed in other materials, e.g. in SrTiO₃, an oxide with perovskite structure. The computed defect potentials and corresponding electron

transport properties for lanthanum (La) doping (on Sr site) and niobium (Nb) doping (on Ti site) are shown in Supplementary Fig. 4. In order to correctly describe the phonon modes in SrTiO₃ with perovskite phase, *ab initio* molecular dynamics was used to extract effective force constants at finite temperature^{24,25}. Neither dopant creates significant repulsive potentials that can counteract the Coulomb field. Instead, Nb dopant creates a strong attractive short-range potential that further enhances the electron-defect scatterings. As a result, the computed mobility for Nb-doped SrTiO₃ is lower than La-doped one. We caution that the computed mobilities are not accurate due to the use of quasiparticle picture in Boltzmann transport theory²⁶, which ignores the polaron nature of charge transport in SrTiO₃. Nonetheless, the general trend of mobility with different dopants (La and Nb) agrees with experiments²⁷ (Supplementary Fig. 4e). The above examples confirmed the electron scattering scenario due to strong short-range electron-defect interactions and further demonstrate the ability of our computational approach to distinguish the impact of different charged defects on electron transport.

Electron cloaking

We now discuss in what circumstances the short-range electron-defect interactions can instead enhance the mobility and possible evidences that demonstrate electron cloaking, using half-Heusler materials as examples. Half-Heuslers are a promising material family for spintronic and thermoelectric applications. Their large compositional variability has opened up a wide space for exploring different dopants to optimize their charge transport. We will use the ionic radius as a scale bar for the sign of the short-range defect potential, which determines whether it would be detrimental or beneficial to the charge mobility. First, we establish the correlation between ionic radius and the sEDI matrix. As explained above, a dopant with a large (small) ionic radius compared to that of the host atom tends to create a strong repulsive (attractive) short-range

potential $\Delta\hat{V}_{cent}$, and thereby a large positive (negative) value in the sEDI matrix $\langle\psi_{k'}|\Delta\hat{V}_{cent}|\psi_k\rangle$. We therefore expect the sEDI matrix to correlate with the ionic radius difference between the dopant and host atoms. Figure 3a shows a comparison between computed sEDI matrices in half-Heusler materials (for electrons/holes at the band edge) and the ionic radius difference between the dopant and host atoms, which indeed demonstrates this correlation. Here, the ionic radii are taken from theoretical calculations based on Slater orbitals excluding the valence electrons²³. Clearly, the sEDI matrix also depends on perturbations of valence electrons, as well as on the projection of electronic wavefunctions on the dopant atom. Nonetheless, the variation in the ionic radius captures the dominant short-range perturbation and provides a reasonable estimation for the sign and strength of the central cell potential. Based on this plot, dopants in the left region will lead to electron scatterings while dopants in the right region would favor electron cloaking (for n-type materials). Examples of defect potentials from dopants shown in Fig. 3a leading to electron-scattering or -cloaking scenarios can be found in Supplementary Fig. 5.

Classification of dopants based on their ionic radius allows us to further understand the electron transport behavior in heavily doped semiconductors, and is a potential guide in the selection of dopants to enhance mobility. This can be seen in the experimental results of two example compounds – ZrNiSn and NbFeSb, which are well known for their thermoelectric performance among n-type and p-type semiconductors, respectively. For n-type ZrNiSn, the typical dopants (V, Nb, or Ta) each have a smaller ionic radius than the host atom (Zr, Fig. 3a), resulting in an attractive central cell potential. This attractive potential adds to the attractive Coulomb potential (for n-type material) and increases the defect scattering. As a result, the compiled experimental mobility data mostly show a monotonic decreasing trend with increasing carrier concentration (Fig. 3b). The non-monotonic trend in one data has been attributed to other

extrinsic defect scatterings and is not directly related to dopants²⁸. In Fig. 3d, we further show highest mobility values for V, Nb and Ta doping from past work with respect to the ionic radius difference between dopant and host ($r_{dopant} - r_{host}$, based on Fig. 3a). A general trend of increasing mobility with increasing dopant ionic radius is observed. This is consistent with our defect scattering picture (Fig. 1a), because for n-type materials a dopant with larger ionic radius is less attractive to electrons and therefore contributes less to total electron-defect scatterings.

On the other hand, for p-type NbFeSb, while the typical dopants (Ti, Hf) each still have a smaller ionic radius than the host atom (Nb), the Coulomb potential is now repulsive (for p-type material), so the two potentials counteract each other. In this case, there will be a carrier concentration range in which the dopant scattering is weak and the mobility approaches the intrinsic electron-phonon scattering limit, manifesting as a peak with increasing carrier concentration (Fig. 1i). Such peaks are indeed observed in the compiled experimental mobility data for p-type NbFeSb materials (Fig. 3c), suggesting that a partial electron cloaking effect is at play. However, we acknowledge that other extrinsic effects, such as existence of compensated charged defects, may also lead to non-monotonic mobility variation. Another mechanism that is responsible for mobility peaks is the screening of polar optical phonon scattering, an intrinsic scattering mechanism²⁹. Still, if the scattering from the dopant is strong, the mobility would instead be limited by electron-defect interactions and a monotonically decreasing trend would be expected. The general observation of mobility peaks in NbFeSb suggests electron cloaking effect likely exists. Another evidence supporting electron cloaking effect is shown in Fig. 3e, which plots the highest mobility values from past work for different dopants with respect to the ionic radius difference. In contrast to the case with ZrNiSn, dopant with the largest ionic radius difference from the host (Ti dopant on Nb) has highest mobility, consistent throughout many studies. This

contradicts simple defect scattering picture, which would suggest such dopants should create strongest short-range electron-dopant scattering and decrease the mobility. However, because small ionic radius dopants actually create attractive short-range potential that counteracts the repulsive Coulomb field in p-type materials, the overall electron-defect scattering should decrease (and correspondingly the mobility increases) as the dopant's ionic radius becomes smaller, which is the trend observed in Fig. 3e. The electron dopant interaction picture based on the ionic radius is consistent with both n-type and p-type materials (Fig. 3c,e).

In order to quantify the extent to which electron cloaking can benefit the electron transport, and in particular the thermoelectric performance in the case of half-Heusler materials, we computed the electron transport properties in p-type NbFeSb. Ti-doped NbFeSb was recently reported to possess a high power factor at room temperature³⁰. The analysis above suggests that Ti doping in p-type NbFeSb facilitates electron cloaking (Fig. 3c,e). Our calculations show that despite the large doping concentrations, the electron scattering rates are indeed close to the intrinsic limit (determined by electron-phonon scattering) due to the counteracting Coulomb potential and the short-range defect potential of Ti (Supplementary Fig. 6). This partial electron cloaking effect leads to higher electrical conductivity and a larger thermoelectric power factor in the range from 300 K to 1000 K, bringing the simulation results into better agreement with the experiment (Fig. 3f-g). Moreover, the relative magnitudes of the optimal power factors in NbFeSb with the various dopants experimentally investigated thus far³¹ are also consistent with our calculations (Fig. 3h), and the trend with the ionic radius agrees with our electron dopant interaction picture (compare Fig. 3h with Fig. 3e). The higher power factors achieved with Ti and Hf dopants can be understood by their defect potentials: their short-range potentials counterbalance the Coulomb potential and create a partial cloaking effect that enhances the charge mobility (Supplementary Fig. 6). In a

similar compound (Ti-doped TaFeSb) we also observed a large power factor enhancement due to electron cloaking (Supplementary Fig. 7). While these enhancements may not seem large, the thermoelectric figure of merit zT is directly proportional to the power factor. In this regard, rational dopant selection that improves the charge mobility and power factor will be beneficial for the thermoelectric efficiency, whose improvement has been a challenging task. Besides, we note that such enhancement due to electron cloaking is expected to become stronger at lower temperatures, due to the increasing importance of defect scatterings compared to intrinsic electron-phonon scatterings as the temperature decreases. Our computation shows that Ti dopant in NbFeSb and TaFeSb can potentially lead to significant enhancement of the power factor by as much as 80% at 150 K compared to the conventional Coulomb-limited case due to the cloaking effect of the dopant (Supplementary Fig. 8). This thus also provides an opportunity to optimize thermoelectric materials for cooling and refrigeration applications³² through electron cloaking.

In summary, we have demonstrated that the chemical details of certain dopants can be harnessed to counteract the strong electron scatterings resulting from their long-range Coulomb field. Consequently, in contrast to the conventional belief that charge mobility is always limited by extrinsic Coulomb scatterings, we have shown how high intrinsic mobility can be achieved by rationally selecting dopants based on their ionic radius. While our study focuses on point defects, the first-principles computational approach can be applied to other short-range interactions such as those due to defect clusters, and even dislocations or grain boundaries, when the long-range Coulomb interactions and short-range perturbations are comparable in strength. Our results provide guidelines for dopant selection, which thus far has been mostly based on trial-and-error. The insights provided here on the impact of often-neglected atomic details of defects on charge transport in heavily doped materials will stimulate the search for high-efficiency thermoelectric

materials, as well as the development of high-mobility materials for microelectronic and optoelectronic applications.

Methods

Model study of electron-defect scattering

To illustrate the general impact of central cell potential on charge transport, we have used a model semiconductor with a parametrized isotropic band structure and scattering information corresponding to that of silicon, and we represent the defect potential by a simplified profile as shown in Fig. 1a. The charge mobility is given by

$$\mu_e = \left[\frac{N_v e}{3} \int v^2 \tau \left(-\frac{\partial f^0}{\partial E} \right) D(E) dE \right] / n \quad (1)$$

where the integration spans over electron states close to the Fermi level, N_v is the band degeneracy, e is the electronic charge, v is the electron group velocity and is related to electron energy *via* the conductivity effective mass $m_{eff,c}$, as $v^2 = 2E/m_{eff,c}$, τ is the electron relaxation time, $f^0 = 1/(1 + \exp(\frac{E-\mu}{k_B T}))$ is the Fermi-Dirac distribution function with μ being the Fermi level, E is the electron energy, $D(E)$ is the electronic density of states and is related to the density-of-states effective mass $m_{eff,DOS}$ *via* $D = (\frac{2m_{eff,DOS}}{\hbar^2})^{3/2} \frac{\sqrt{E}}{2\pi^2}$ with \hbar being the reduced Planck constant, and n is the carrier concentration.

The electron relaxation time τ , the inverse of the scattering rate, is determined *via* Matthiessen's rule considering both intrinsic electron-phonon interactions and extrinsic electron-defect interactions: $1/\tau = 1/\tau_{e-ph} + 1/\tau_{e-d}$. The electron-phonon interactions consider both acoustic phonon and optical phonon scatterings *via* corresponding deformation potentials, and the full parametrization is given in Supplementary Note. The electron-defect scattering rate $1/\tau_{e-d}$ is calculated based on the partial wave analysis which evaluates the scattering of electrons by a spherically symmetric potential

$$\frac{1}{\tau_{e-d}} = N_d \frac{4\pi}{\hbar^2} \frac{1}{m_{eff,DOS} \sqrt{2m_{eff,DOS} E}} \sum_{l=0}^{\infty} (l+1) \sin^2(\delta_l - \delta_{l+1}) \quad (2)$$

where N_d is the volume density of the defect and δ_l is the phase shift of the electron wave with quantum number l . The defect potential (in units of energy) has been taken to have the form $\Delta\hat{V} = \begin{cases} V_0 & r \leq r_0 \\ -\frac{e^2}{4\pi\epsilon\epsilon_0 r} & r > r_0 \end{cases}$, with r_0 characterizing the range of the short-range potential, and where V_0 is the short-range potential energy and ϵ_0 is the vacuum permittivity. For the results in Fig. 1, we have assumed $V_0 = 9.7$ eV and $r_0 = 1.6$ Å. The scattering rates and mobility in Fig. 1d-g are obtained assuming $\epsilon = 11.7$ (dielectric constant corresponding to silicon) while the dielectric constant is varied in Fig. 2a.

Details of the calculation of the partial wave phase shift are provided in the Supplementary Note. In brief, because the defect potential is spherically symmetric, the orbital angular momentum operator becomes a constant of motion for electrons, and the electron wavefunctions can be represented by an additional quantum number l in addition to its energy E , called partial waves.

Each partial wave is scattered by the potential and the scattered wave acquires a phase shift δ_l compared to the case in which no defect is present. Intuitively, the phase shift represents how strongly the defect potential attracts or repels the electrons, both of which will lead to a large phase shift and thus strong scattering. The total scattering rates are obtained *via* the above formula by considering all values of l . The partial wave analysis considers multiple scatterings between the electron wave and the defect, and is thus exact under the assumption that the electron wave is a plane wave²⁰. Although the electron wave actually has a more complex profile modulated by the periodic potential in the crystal, the above results provide an estimation of the effect of the central cell potential on charge transport, particularly in comparison to the conventional Coulomb scatterings.

First-principles calculation of defect potential

The defect potential, $\Delta\hat{V}$, is defined as the difference in the total electronic potential between the system containing one defect and the pristine bulk material, $\Delta\hat{V} = V_d - V_0$, where the calculation is performed for a cubic supercell containing 96 atoms for half-Heusler materials. In the calculation of the structure with one defect, a net charge is given corresponding to the charged state of the dopant. In our study, the dopants were chosen from a column in the periodic table adjacent to that of the atom being substituted, and thus the net charge is assumed to be +1 for n-type and -1 for p-type. The extracted defect potential contains both the long-range Coulomb potential and the central cell part which deviates from the Coulomb potential profile. This defect potential cannot be directly used to compute the EDI matrix $\langle\psi_{k'}|\Delta\hat{V}|\psi_k\rangle$ since the finite size of the supercell does not correctly capture the large span of the long-range Coulomb field. However, because the long-range tail of the defect potential agrees well with the analytic Coulomb potential profile (Fig. 1b-c), we can first subtract the long-range part from the defect potential, leaving only a short-range component $\Delta\hat{V}_{sr}$. **In figures where we show the defect potential extracted from first principles simulation, we have also plotted the Coulomb potential to show that first principles defect potential indeed recovers the correct asymptotic trend given by the Coulomb field when one moves away from the defect, as has been shown in previous work²².** When evaluating the EDI matrix later, we add the long-range Coulomb potential back to the defect potential, $\Delta\hat{V} = \Delta\hat{V}_{sr} + \Delta\hat{V}_{lr}$. Because the second contribution to EDI due to the long-range part can be computed using an analytic expression for the Coulomb potential extending to a longer distance (to be detailed below), we circumvent the difficulty of the finite size of the supercell and are able to treat both long-range and short-range potentials on an equal footing. This workflow is similar to the recent development in incorporating the long-range polar optical phonon scattering into the Wannier interpolation method for electron-phonon interaction calculations^{33,34}.

When subtracting the long-range part from the defect potential, we compute the long-range potential according to the following formula³⁵ based on the Ewald summation, which represents the Coulomb potential at location \mathbf{r} due to an infinite periodic array of charge Ze at locations \mathbf{R}_i (given by the supercell size).

$$\Delta\hat{V}_{lr} = -\sum_i \frac{ze^2}{\sqrt{|\bar{\epsilon}|}} \frac{\text{erfc}\left(\gamma\sqrt{(\mathbf{R}_i-\mathbf{r})\cdot\bar{\epsilon}^{-1}\cdot(\mathbf{R}_i-\mathbf{r})}\right)}{\sqrt{(\mathbf{R}_i-\mathbf{r})\cdot\bar{\epsilon}^{-1}\cdot(\mathbf{R}_i-\mathbf{r})}} - \sum_{\mathbf{G}_i \neq 0} \frac{4\pi Ze^2}{\Omega} \frac{\exp\left(-\mathbf{G}_i\cdot\bar{\epsilon}\cdot\frac{\mathbf{G}_i}{4\gamma^2}\right)}{\mathbf{G}_i\cdot\bar{\epsilon}\cdot\mathbf{G}_i} \exp(i\mathbf{G}_i\cdot\mathbf{r}) + \frac{\pi Ze^2}{\Omega\gamma^2} \quad (3)$$

Here $\bar{\epsilon}$ is the dielectric tensor computed from first principles, Ω is the supercell volume, and γ is a convergence parameter for the Ewald summation. In Supplementary Note, we show that the remaining potential after the subtraction is indeed short-range. Because a short-range potential should approach zero at distances away from the defect, we further align the potential at far distances to zero.

Electron transport calculation

The first-principles electron transport properties for half-Heusler materials are calculated by summing over all electron states according to the Boltzmann transport theory³⁶. For example, the electrical conductivity is given by

$$\sigma = \frac{e^2}{3\Omega_0 N_k} \sum_{\mathbf{k}\alpha} v_{\mathbf{k}\alpha}^2 \tau_{\mathbf{k}\alpha} \left(-\frac{\partial f_{\mathbf{k}\alpha}^0}{\partial E} \right) \quad (4)$$

in which we have explicitly written out the summation of the discrete mesh points \mathbf{k} in the Brillouin zone, and where Ω_0 is the unit cell volume and α is the band index. Isotropic materials are assumed and the factor 1/3 appears because the conductivity is the same in all three directions. The equilibrium properties of electrons are calculated from first principles using the QUANTUM ESPRESSO software package³⁷. We use the generalized gradient approximation (GGA) of Perdew, Burke and Ernzerhof³⁸ with the Troullier-Martins-type norm-conserving semilocal pseudopotential (corresponding to pbe-mt.UPF in the QUANTUM ESPRESSO pseudopotential library). A cutoff energy of 120 Ryd and a $6 \times 6 \times 6$ \mathbf{k} -mesh are used to determine the equilibrium lattice constant. The equilibrium properties of phonons and the electron-phonon interaction matrices are calculated *via* density functional perturbation theory³⁹ for a $6 \times 6 \times 6$ \mathbf{q} -mesh (with a $6 \times 6 \times 6$ \mathbf{k} -mesh for the electron-phonon interaction matrix). We then use the EPW software package⁴⁰ to interpolate the electronic information and the phonon information, as well as the electron-phonon coupling matrices to a fine mesh *via* the Wannier interpolation method⁴¹. In determining the Fermi level μ , we assumed that the dopants are fully ionized and therefore the Fermi level is such that the doping concentration N_d equals the total carrier concentration, which is given by $n = \frac{1}{\Omega_0 N_k} \sum_{\mathbf{k}\alpha} f_{\mathbf{k}\alpha}^0$. The electron relaxation time τ is determined *via* Matthiessen's rule considering both intrinsic electron-phonon interactions and extrinsic electron-defect interactions: $1/\tau = 1/\tau_{e-ph} + 1/\tau_{e-d}$. The electron-phonon interaction matrices are first calculated within density functional perturbation theory and then interpolated *via* the Wannier interpolation scheme to the fine mesh⁴⁰. This includes electron scatterings by polar optical phonons^{33,34}, as well as taking the carrier screening effect into account. More details on this can be found in our previous work³⁶ and in Supplementary Note.

The calculation of electron-defect scattering rates considering the full defect potential is the key step in our study, and is given by the following formula under the momentum relaxation approximation

$$\frac{1}{\tau_k^{e-d}} = N_d \Omega_0 \frac{2\pi}{\hbar} \frac{1}{N_k} \sum_{\mathbf{k}'} \left(1 - \frac{\mathbf{v}_k \cdot \mathbf{v}_{\mathbf{k}'}}{|\mathbf{v}_k| |\mathbf{v}_{\mathbf{k}'}} \right) |g_{e-d}(\mathbf{k}, \mathbf{k}')|^2 \delta(E_k - E_{\mathbf{k}'}) \quad (5)$$

where N_d is the volume density of dopants, N_k is the number of \mathbf{k} points, the factor $1 - \frac{\mathbf{v}_k \cdot \mathbf{v}_{\mathbf{k}'}}{|\mathbf{v}_k| |\mathbf{v}_{\mathbf{k}'}}$ takes into account the fact that scatterings between electrons with similar velocity directions do not contribute much to momentum loss and thus less to electrical resistance, and $\delta(E_k - E_{\mathbf{k}'})$ indicates that the defect scattering is an elastic process. $g_{e-d}(\mathbf{k}, \mathbf{k}') = \langle \psi_{\mathbf{k}'} | \Delta \hat{V} | \psi_{\mathbf{k}} \rangle$ is the EDI matrix. As explained above, the defect potential $\Delta \hat{V}$ contains both long-range and short-range parts, leading to two contributions to EDI. To compute these contributions, the EDI matrix is first rewritten as follows¹⁹

$$\langle \psi_{\mathbf{k}} | \Delta \hat{V} | \psi_{\mathbf{k}'} \rangle = \int d^3r u_{\mathbf{k}'}^* e^{-i\mathbf{k}' \cdot \mathbf{r}} \Delta \hat{V} u_{\mathbf{k}} e^{i\mathbf{k} \cdot \mathbf{r}} = \sum_{\mathbf{G}} \Delta V(\mathbf{k}' - \mathbf{k} + \mathbf{G}) \langle u_{\mathbf{k}'} | e^{i\mathbf{G} \cdot \mathbf{r}} | u_{\mathbf{k}} \rangle \quad (6)$$

where the Fourier transform of the defect potential is defined as $\Delta V(\mathbf{q}) = \frac{1}{\Omega_0} \int d^3r \Delta \hat{V}(\mathbf{r}) e^{-i\mathbf{q} \cdot \mathbf{r}}$, with Ω_0 being the unit cell volume and the integration spanning the entire space. This form separates the defect potential from the wave functions, and the factor containing wavefunctions can be computed readily once the periodic components $u_{\mathbf{k}}$ of the wavefunctions are known¹⁹: $\langle u_{\mathbf{k}'} | e^{i\mathbf{G} \cdot \mathbf{r}} | u_{\mathbf{k}} \rangle = \int d^3r u_{\mathbf{k}'}^* e^{-i\mathbf{G} \cdot \mathbf{r}} u_{\mathbf{k}}$, where the integration spans over the unit cell. To evaluate the EDI matrix, we then compute the Fourier component of the defect potential $\Delta V(\mathbf{q})$, which again contains both long-range and short-range parts. The short-range part can be calculated readily within the supercell based on $\Delta V_{sr}(\mathbf{q}) = \frac{1}{\Omega_0} \int d^3r \Delta \hat{V}_{sr}(\mathbf{r}) e^{-i\mathbf{q} \cdot \mathbf{r}}$ since the potential has negligible contributions at far distances. The long-range part can be obtained by performing the integration analytically to infinity, yielding $\Delta V_{lr}(\mathbf{q}) = -\frac{Ze^2}{\Omega_0 \epsilon \epsilon_0} \frac{1}{|\mathbf{q}|^2 + (1/L_D)^2}$. For this expression, we have assumed the Coulomb potential energy is given by $\Delta \hat{V}_{lr}(\mathbf{r}) = -\frac{Ze^2}{4\pi \epsilon \epsilon_0} \frac{\exp(-r/L_D)}{r}$, where the factor $\exp(-r/L_D)$ considers the carrier screening at high carrier concentrations with the Debye screening length L_D given by

$$L_D = \left(\frac{e^2}{\epsilon \epsilon_0} \int \left(-\frac{\partial f}{\partial E} \right) D(E) dE \right)^{-1/2} \quad (7)$$

Adding both long-range and short-range components as $\Delta V(\mathbf{q}) = \Delta V_{sr}(\mathbf{q}) + \Delta V_{lr}(\mathbf{q})$ allows us to evaluate the EDI matrix completely. The electron-defect scattering rates thus obtained are then combined with electron-phonon scatterings to give the total electron scattering rates, which are used to evaluate the electron transport properties, as in Eq. (4).

Data Availability

The data that support the findings of this study are available from the corresponding authors on reasonable request.

Code Availability

The code for computing electron scattering rates through first principles electron transport calculation is a modified version of the EPW code⁴⁰, originally released within the QUANTUM ESPRESSO package³⁷. Our modified EPW code is available at <http://doi.org/10.24435/materialscloud:5a-7s>.

Acknowledgement

We thank T.-H. Liu, M. Li, and Q. Zhang for helpful discussions on the first-principles calculations and electron-defect interactions. The work performed at MIT is supported by the DARPA MATRIX program under Grant No. HR0011-16-2-0041.

Author Contributions

J.Z. and G.C. conceived the project. J.Z. performed the theoretical analysis and first-principles computation. J.Z., H.Z., Q.S., Z.R., and G.C. contributed to the data analysis. J.Z. and G.C. wrote the manuscript. All authors commented on, discussed, and edited the manuscript.

Competing Financial Interests

The authors declare no competing financial interests.

Reference

1. Sze, S. M. *Physics of Semiconductor Devices*. (John Wiley & Sons, 1981).
2. Minnich, A. J., Dresselhaus, M. S., Ren, Z. F. & Chen, G. Bulk nanostructured thermoelectric materials: current research and future prospects. *Energy Environ. Sci.* **2**, 466–479 (2009).
3. Chattopadhyay, D. & Queisser, H. J. Electron scattering by ionized impurities in semiconductors. *Rev. Mod. Phys.* **53**, 745–768 (1981).
4. Freysoldt, C. *et al.* First-principles calculations for point defects in solids. *Rev. Mod. Phys.* **86**, 253–305 (2014).
5. Brooks, H. Theory of the electrical properties of germanium and silicon. in *Advances in Electronics and Electron Physics* (ed. Marton, L.) vol. 7 85–182 (Academic Press, 1955).
6. Herring, C. Transport properties of a many-valley semiconductor. *The Bell System Technical Journal* **34**, 237–290 (1955).
7. Fischetti, M. V. Effect of the electron-plasmon interaction on the electron mobility in silicon. *Phys. Rev. B* **44**, 5527–5534 (1991).
8. Sanborn, B. A., Allen, P. B. & Mahan, G. D. Theory of screening and electron mobility: Application to n-type silicon. *Phys. Rev. B* **46**, 15123–15134 (1992).
9. Ralph, H. I., Simpson, G. & Elliott, R. J. Central-cell corrections to the theory of ionized-impurity scattering of electrons in silicon. *Phys. Rev. B* **11**, 2948–2956 (1975).
10. El-Ghanem, H. M. A. & Ridley, B. K. Impurity scattering of electrons in non-degenerate semiconductors. *J. Phys. C: Solid State Phys.* **13**, 2041 (1980).
11. Sankey, O. F., Dow, J. D. & Hess, K. Theory of resonant scattering in semiconductors due to impurity central-cell potentials. *Appl. Phys. Lett.* **41**, 664–666 (1982).
12. Szymyd, D. M., Hanna, M. C. & Majerfeld, A. Heavily doped GaAs:Se. II. Electron mobility. *Journal of Applied Physics* **68**, 2376–2381 (1990).
13. Wang, H., Cao, X., Takagiwa, Y. & Snyder, G. J. Higher mobility in bulk semiconductors by separating the dopants from the charge-conducting band – a case study of thermoelectric PbSe. *Mater. Horiz.* **2**, 323–329 (2015).
14. Reagor, D. W. & Butko, V. Y. Highly conductive nanolayers on strontium titanate produced by preferential ion-beam etching. *Nature Materials* **4**, 593–596 (2005).
15. Hitosugi, T., Yamada, N., Nakao, S., Hirose, Y. & Hasegawa, T. Properties of TiO₂-based transparent conducting oxides. *physica status solidi (a)* **207**, 1529–1537 (2010).
16. Liu, W. *et al.* New trends, strategies and opportunities in thermoelectric materials: A perspective. *Materials Today Physics* **1**, 50–60 (2017).
17. Lany, S. & Zunger, A. Assessment of correction methods for the band-gap problem and for finite-size effects in supercell defect calculations: Case studies for ZnO and GaAs. *Phys. Rev. B* **78**, 235104 (2008).
18. Freysoldt, C., Neugebauer, J. & Van de Walle, C. G. Fully ab initio finite-size corrections for charged-defect supercell calculations. *Phys. Rev. Lett.* **102**, 016402 (2009).
19. Lu, I.-T., Zhou, J.-J. & Bernardi, M. Efficient ab initio calculations of electron-defect scattering and defect-limited carrier mobility. *Phys. Rev. Materials* **3**, 033804 (2019).

20. Liao, B., Zebarjadi, M., Esfarjani, K. & Chen, G. Cloaking core-shell nanoparticles from conducting electrons in solids. *Phys. Rev. Lett.* **109**, 126806 (2012).
21. Cohen-Tannoudji, C., Diu, B. & Laloe, F. *Quantum Mechanics*. (Wiley, 1992).
22. Rurali, R., Markussen, T., Suñé, J., Brandbyge, M. & Jauho, A.-P. Modeling transport in ultrathin Si nanowires: Charged versus neutral impurities. *Nano Lett.* **8**, 2825–2828 (2008).
23. Ghosh, D. C. & Biswas, R. Theoretical calculation of absolute radii of atoms and ions. Part 2. The ionic radii. *International Journal of Molecular Sciences* **4**, 379–407 (2003).
24. Hellman, O., Steneteg, P., Abrikosov, I. A. & Simak, S. I. Temperature dependent effective potential method for accurate free energy calculations of solids. *Phys. Rev. B* **87**, 104111 (2013).
25. Tadano, T., Gohda, Y. & Tsuneyuki, S. Anharmonic force constants extracted from first-principles molecular dynamics: applications to heat transfer simulations. *J. Phys.: Condens. Matter* **26**, 225402 (2014).
26. Zhou, J.-J. & Bernardi, M. Predicting charge transport in the presence of polarons: The beyond-quasiparticle regime in SrTiO₃. *Phys. Rev. Research* **1**, 033138 (2019).
27. Han, W. *et al.* Spin injection and detection in lanthanum- and niobium-doped SrTiO₃ using the Hanle technique. *Nat Commun* **4**, 2134 (2013).
28. Xie, H. *et al.* The intrinsic disorder related alloy scattering in ZrNiSn half-Heusler thermoelectric materials. *Scientific Reports* **4**, 6888 (2014).
29. Ren, Q. *et al.* Establishing the carrier scattering phase diagram for ZrNiSn-based half-Heusler thermoelectric materials. *Nature Communications* **11**, 3142 (2020).
30. He, R. *et al.* Achieving high power factor and output power density in p-type half-Heuslers Nb_{1-x}Ti_xFeSb. *PNAS* **113**, 13576–13581 (2016).
31. Ren, W. *et al.* Ultrahigh power factor in thermoelectric system Nb_{0.95}M_{0.05}FeSb (M = Hf, Zr, and Ti). *Advanced Science* **5**, 1800278 (2018).
32. Mao, J. *et al.* High thermoelectric cooling performance of n-type Mg₃Bi₂-based materials. *Science* **365**, 495–498 (2019).
33. Sjakste, J., Vast, N., Calandra, M. & Mauri, F. Wannier interpolation of the electron-phonon matrix elements in polar semiconductors: Polar-optical coupling in GaAs. *Phys. Rev. B* **92**, 054307 (2015).
34. Verdi, C. & Giustino, F. Fröhlich Electron-Phonon Vertex from First Principles. *Phys. Rev. Lett.* **115**, 176401 (2015).
35. Kumagai, Y. & Oba, F. Electrostatics-based finite-size corrections for first-principles point defect calculations. *Phys. Rev. B* **89**, 195205 (2014).
36. Zhou, J., Liao, B. & Chen, G. First-principles calculations of thermal, electrical, and thermoelectric transport properties of semiconductors. *Semicond. Sci. Technol.* **31**, 043001 (2016).
37. Giannozzi, P. *et al.* QUANTUM ESPRESSO: a modular and open-source software project for quantum simulations of materials. *J. Phys.: Condens. Matter* **21**, 395502 (2009).
38. Perdew, J. P., Burke, K. & Ernzerhof, M. Generalized gradient approximation made simple. *Phys. Rev. Lett.* **77**, 3865–3868 (1996).

39. Baroni, S., de Gironcoli, S., Dal Corso, A. & Giannozzi, P. Phonons and related crystal properties from density-functional perturbation theory. *Rev. Mod. Phys.* **73**, 515–562 (2001).
40. Giustino, F., Cohen, M. & Louie, S. Electron-phonon interaction using Wannier functions. *Phys. Rev. B* **76**, 165108 (2007).
41. Marzari, N., Mostofi, A. A., Yates, J. R., Souza, I. & Vanderbilt, D. Maximally localized Wannier functions: Theory and applications. *Rev. Mod. Phys.* **84**, 1419–1475 (2012).
42. LaLonde, A. D., Pei, Y. & Snyder, G. J. Reevaluation of $\text{PbTe}_{1-x}\text{I}_x$ as high performance n-type thermoelectric material. *Energy Environ. Sci.* **4**, 2090–2096 (2011).
43. Gelbstein, Y., Dashevsky, Z. & Dariel, M. P. Synthesis of n-type PbTe by powder metallurgy. in *Proceedings ICT2001 20th International Conference on Thermoelectrics* 143–149 (2001).
44. Rogacheva, E. I., Lyubchenko, S. G., Vodoretz, O., Kuzmenko, A. M. & Dresselhaus, M. Thermoelectric properties of PbTe crystals and thin films. in *2006 25th International Conference on Thermoelectrics* 656–661 (2006).
45. Chauhan, N. S. *et al.* Vanadium-doping-induced resonant energy levels for the enhancement of thermoelectric performance in Hf-free ZrNiSn half-Heusler alloys. *ACS Appl. Energy Mater.* **1**, 757–764 (2018).
46. Muta, H., Kanemitsu, T., Kurosaki, K. & Yamanaka, S. High-temperature thermoelectric properties of Nb-doped MNiSn (M=Ti, Zr) half-Heusler compound. *Journal of Alloys and Compounds* **469**, 50–55 (2009).
47. Zhang, H. *et al.* Thermoelectric properties of n-type half-Heusler compounds $(\text{Hf}_{0.25}\text{Zr}_{0.75})_{1-x}\text{Nb}_x\text{NiSn}$. *Acta Materialia* **113**, 41–47 (2016).
48. Yang, X. *et al.* Enhanced thermoelectric performance of $\text{Zr}_{1-x}\text{Ta}_x\text{NiSn}$ half-Heusler alloys by diagonal-rule doping. *ACS Appl. Mater. Interfaces* **12**, 3773–3783 (2020).
49. Zhao, D., Zuo, M., Wang, Z., Teng, X. & Geng, H. Synthesis and thermoelectric properties of tantalum-doped ZrNiSn half-Heusler alloys. *Funct. Mater. Lett.* **07**, 1450032 (2014).
50. Fu, C., Zhu, T., Liu, Y., Xie, H. & Zhao, X. Band engineering of high performance p-type FeNbSb based half-Heusler thermoelectric materials for figure of merit $zT > 1$. *Energy Environ. Sci.* **8**, 216–220 (2015).
51. Fu, C. *et al.* Realizing high figure of merit in heavy-band p-type half-Heusler thermoelectric materials. *Nat Commun* **6**, 8144 (2015).
52. Tavassoli, A. *et al.* On the Half-Heusler compounds $\text{Nb}_{1-x}\{\text{Ti,Zr,Hf}\}_x\text{FeSb}$: Phase relations, thermoelectric properties at low and high temperature, and mechanical properties. *Acta Materialia* **135**, 263–276 (2017).

Figure Legends

Figure 1. Impact of charged defects on electron transport. **a**, Illustration of a propagating electron wave scattered by the perturbed electronic potential due to the presence of charged defects. The example of an n-type dopant is depicted. The defect potential in general contains two parts, a long-range part due to the Coulomb field of the charge which is attractive to electrons for a n-type dopant, and a short-range part that depends on the bonding environment of the defect, known as the central cell potential. **The ionic radius can be used as a good indicator for the sign of the central cell potential.** Depending on the ionic radius of the dopant atom, one of two general scenarios can play out. If the dopant atom has a smaller ionic radius than the host atom, the perturbation tends to create an additional attractive force for electrons (electron-scattering scenario); if the dopant atom has a larger ionic radius, it tends to create a repulsive force which then opposes the long-range Coulomb potential (electron-cloaking scenario). The above discussions pertain to the case of n-type dopant. **b-c**, Defect potentials and scattered electron wavefunctions corresponding to two different scenarios (**b**: electron scattering, **c**: electron cloaking). Here the central cell potential is represented by a simplified rectangular profile. The streamlines show the probability flux and the colors indicate the real part of the wavefunctions. In the case of electron cloaking (**c**), unperturbed streamlines are recovered away from the defect (located at the center). The domain size is $158.8 \text{ \AA} \times 158.8 \text{ \AA}$ and the electron energy is 0.1 eV . **d-e**, Phase shifts of scattered wave functions with angular momentum quantum number of $l = 1$ for two scenarios (**d**: electron scattering, **e**: electron cloaking). **f-g**, Modeled intrinsic electron scattering rates due to electron-phonon interactions, in comparison to those considering electron-defect scatterings (**f**: electron scattering, **g**: electron cloaking). The calculation assumes a parabolic band to illustrate the general impact of defect scatterings and uses partial wave analysis to calculate the scattering rates due to defects (see details in Methods). In (**g**), because the central cell potential opposes the Coulomb potential, their effects on electron scatterings are partially canceled. **h-i**, Electron mobility with respect to the carrier concentration (**h**: electron scattering, **i**: electron cloaking). While mobility at high carrier concentrations is traditionally believed to be limited by Coulomb scattering, a strong opposing central cell potential can be harnessed to break this limit, leading to high mobility limited only by the intrinsic electron-phonon interactions, as shown in (**i**).

Figure 2. Electron-defect interaction. **a**, Ratio between characteristic electron scattering rates due to central cell scattering and those due to Coulomb scattering, $\eta = \gamma_{cent}/\gamma_{Coul}$, shown as a color map (with red indicating high η , blue indicating low η , and white indicating unity, as shown in the scale at right), with respect to the dielectric constant and carrier concentration. The characteristic electron scattering rate γ is defined based on the mobility μ as $\gamma = e/(m^*\mu)$. **Studied materials in this work are also labeled in the plot.** Both PbTe and SrTiO₃ have large dielectric constants and the arrows indicate their actual locations lie outside the given dielectric constant range. Regions with $\eta \approx 1$ represent cases in which the central cell potential has a comparable impact on charge transport to that of the Coulomb potential, and thus could be harnessed to counteract the Coulomb scatterings. The magnitude of the central cell potential is taken to be 6 eV (with a width of 6 Bohr radius) in this simulation. **b-c**, Contour plots of the electron-defect interaction matrix $\langle \psi_{\mathbf{k}} | \Delta \hat{V} | \psi_{\mathbf{k}} \rangle$ for **(b)** Si-doped GaAs and **(c)** Bi-doped PbTe, respectively, where \mathbf{k} is taken to be at the conduction band edge state (at the Γ point for GaAs and at the L point for PbTe). Significant electron-defect interaction is seen for PbTe. The unit of the coordinates is \AA and the center is at the defect location. **d-e**, Computed electron mobility as a function of the carrier density for **(d)** n-type GaAs with two different dopants (Ga:Si, As:Te) and **(e)** n-type PbTe with two different dopants (Pb:Bi, Te:I) in comparison to experimental results [experimental sources: As:Te and Ga:Si¹²; Te:I (squares)⁴²; Te:I (diamonds)⁴³; and Pb:Bi⁴⁴]. For PbTe, Bi doping strongly reduces the mobility compared to I doping, whereas the dopant effects on GaAs are comparatively smaller.

Figure 3. Defect-mediated electrical and thermoelectric transport. **a**, Correlation between ionic radius difference and the **short-range** electron-defect interaction matrix in half-Heusler materials with different dopants. Inset: relative sizes of the related atoms illustrated based on their ionic radius²³, with the charge of each shown at the top of each column. The electron-defect interaction matrix is calculated for the band edge state. **b-c**, Compiled experimental data for mobility dependence on the carrier concentration in representative half-Heusler materials with different dopants: **(b)** n-type ZrNiSn [experimental sources: ZrNiSn:V⁴⁵; ZrNiSn:Nb⁴⁶; Zr_{0.75}Hf_{0.25}NiSn:Nb⁴⁷; ZrNiSn:Ta (violet)⁴⁸; and ZrNiSn:Ta (green)⁴⁹], and **(c)** p-type NbFeSb [experimental sources: NbFeSb: Ti (blue)³⁰; NbFeSb:Ti (red)⁵⁰; NbFeSb:Zr and NbFeSb:Hf⁵¹]. **d-e**, Compiled highest mobility data from past studies with respect to the ionic radius difference between the dopant and host atoms for **(d)** n-type ZrNiSn and **(e)** p-type NbFeSb [experimental sources: ZrNiSn:V⁴⁵; ZrNiSn:Ta^{48,49}; ZrNiSn:Nb^{28,46}; NbFeSb:Ti^{30,31,50,52}; NbFeSb:Hf^{31,51}; NbFeSb:Zr^{31,51,52}]. The ionic radius differences are taken from **(a)**. The shaded regions indicate the standard deviation of these extracted mobility data. **f-g**, Comparisons between simulations and experimental results for the **(f)** electrical conductivity and **(g)** thermoelectric power factor of Ti-doped NbFeSb from 300 K to 1000 K. Consideration of Coulomb scatterings alone underestimates the power factor, while the consideration of full defect scattering with a partial cloaking effect leads to better agreement with the experiment. **(h)** Comparison between simulations and experimental results³¹ for the optimal room-temperature thermoelectric power factor in p-type NbFeSb **with respect to the ionic radius difference between dopant and host atoms**. The trend of power factor with respect to the ionic radius agrees between the experiments and the simulations.

Supplementary Information

Mobility enhancement in heavily doped semiconductors *via* electron cloaking

Jiawei Zhou¹, Hangtian Zhu², Qichen Song¹, Zhiwei Ding¹, Jun Mao², Zhifeng Ren², Gang
Chen¹

¹*Department of Mechanical Engineering, Massachusetts Institute of Technology, Cambridge,
MA 02139, USA*

²*Department of Physics and Texas Center for Superconductivity, University of Houston,
Houston, TX 77204, USA*

Supplementary Note

Model study of defect scattering

Figure 1 of the manuscript showed our electron transport calculation results for a model semiconductor, which illustrates the general effect of electron-defect interaction on electron transport, and demonstrates the possibility of achieving ideal electron cloaking (Fig. 1c,e,g). The electron mobility is calculated as¹

$$\mu_e = \left[\frac{N_v e}{3} \int v^2 \tau \left(-\frac{\partial f^0}{\partial E} \right) D(E) dE \right] / n \quad (1)$$

Here the integration spans over electron states close to the Fermi level, $N_v = 6$ is the band degeneracy and e is the electronic charge. E is the electron energy. v is the electron group velocity and is related to the conductivity effective mass $m_{eff,c}$ ($0.268 m_e$, where m_e is the free electron mass) via $v^2 = 2E/m_{eff,c}$. τ is the electron relaxation time. $f^0 = 1/(1 + \exp(\frac{E-\mu}{k_B T}))$ is the Fermi-Dirac distribution function with μ being the Fermi level. $D(E)$ is the electronic density of states, related to the density-of-states effective mass $m_{eff,DOS}$ ($0.33 m_e$) via $D = (\frac{2m_{eff,DOS}}{\hbar^2})^{3/2} \frac{\sqrt{E}}{2\pi^2}$ with \hbar being the reduced Planck constant. n is the carrier concentration.

The electron relaxation time τ , the inverse of the scattering rate, is determined via Matthiessen's rule considering both intrinsic electron-phonon interactions and extrinsic electron-defect interactions: $1/\tau = 1/\tau_{e-ph} + 1/\tau_{e-d}$. The intrinsic electron-phonon interactions consider both acoustic phonon and optical phonon scatterings via corresponding deformation potentials¹:

$$\frac{1}{\tau_{e-ph,acoustic}} = \frac{\pi D_A^2 k_B T}{\hbar C_l} D(E) \quad (2)$$

where $D_A = 9.6$ eV is the acoustic deformation potential and $C_l = 190.7$ GPa is the elastic constant, and

$$\frac{1}{\tau_{e-ph,optical}} = (N_v - 1) \frac{\pi D_O^2}{2\rho\omega_O} \left[\frac{1}{e^{\frac{\hbar\omega_O}{k_B T}} - 1} D(E + \hbar\omega_O) + \left(\frac{1}{e^{\frac{\hbar\omega_O}{k_B T}} - 1} + 1 \right) D(E - \hbar\omega_O) \right] \quad (3)$$

where $D_O = 6.5$ eV/Å is the optical deformation potential, $\rho = 2330$ kg/m³ is the material density, and ω_O is the angular frequency of the optical phonon ($\omega_O = 2\pi f_O$, with $f_O = 472$ cm⁻¹). The sum of these two scattering rates leads to the intrinsic electron-phonon scattering rates: $1/\tau_{e-ph} = 1/\tau_{e-ph,acoustic} + 1/\tau_{e-ph,optical}$.

The electron-defect scattering rate $1/\tau_{e-d}$ is calculated based on the partial wave analysis which evaluates the scattering of electrons by a spherically symmetric potential²

$$\frac{1}{\tau_{e-d}} = N_d \frac{4\pi}{\hbar^2} \frac{1}{m_{eff,DOS} \sqrt{2m_{eff,DOS} E}} \sum_{l=0}^{\infty} (l+1) \sin^2(\delta_l - \delta_{l+1}) \quad (4)$$

where N_d is the volume density of the defect and δ_l is the phase shift of the electron wave with quantum number l . The maximum number of l in the summation is taken to be 10, which is found to be sufficient to achieve convergence. The scattering of electron waves is obtained by solving the Schrödinger equation. For spherically symmetric potential V , the wavefunction takes the form of $\Psi(\mathbf{r}) = \sum_{l,m} a_{l,m} \frac{u_l(r)}{r} Y_l^m(\theta, \phi)$, where $a_{l,m}$ are constants, and $Y_l^m(\theta, \phi)$ are the normalized spherical harmonics. The radial function $u_l(r)$ satisfies²

$$\frac{d^2 u_l}{dx^2} + \left[\frac{2ma_0^2 E}{\hbar^2} - \frac{2ma_0^2 V(r)}{\hbar^2} - \frac{l(l+1)}{x^2} \right] u_l = 0 \quad (5)$$

where the coordinate has been re-scaled by a characteristic length a_0 ($x = r/a_0$). Two forms of defect potential have been considered. The first corresponds to a pure Coulomb potential

$$\Delta \hat{V} = -\frac{e^2}{4\pi\epsilon\epsilon_0 r} \quad (6)$$

where $\epsilon = 11.7$ is the dielectric constant, and ϵ_0 is the vacuum permittivity. The second corresponds to a more practical case in which the central part is replaced with a central potential

$$\Delta \hat{V} = \begin{cases} V_0 & r \leq r_0 \\ -\frac{e^2}{4\pi\epsilon\epsilon_0 r} & r > r_0 \end{cases} \quad (7)$$

with $r_0 = 1.6 \text{ \AA}$ characterizing the range of the short-range potential, and where $V_0 = 9.7 \text{ eV}$ is the short-range potential energy. The solution to Eq. (5) is obtained by numerical integration based on the Gauss-Jackson method³. The phase shift is determined by matching the obtained solution to the expected asymptotic form of $u_l \approx x B_l \cos(\delta_l) \left[j_l \left(\sqrt{\frac{2ma_0^2 E}{\hbar^2}} x \right) - \tan(\delta_l) y_l \left(\sqrt{\frac{2ma_0^2 E}{\hbar^2}} x \right) \right]$, where $j_l(z)$ and $y_l(z)$ are the regular and irregular spherical Bessel function of order l , respectively. Once the phase shifts are determined, the electron-defect scattering rates can be readily computed using Eq. (4).

Extraction of central cell defect potential

The impurity potential is defined as the difference in the total electronic potential from first-principles calculations between the system with the defect and the original pristine system $\Delta \hat{V} = V_d - V_{bulk}$. One typically builds a large supercell and calculates the defect potential for the pure and defected systems separately. However, as the supercell size is often limited to be no more than a few lattice vectors long, the long-range Coulomb potential is cut off at the supercell boundary and cannot be correctly represented. The challenge of the electron-defect scattering calculation is thus to obtain a correct full profile of the defect potential including both its long-range and short-range parts.

While the central cell potential can vary significantly with the defect and lattice type, we recognize that the long-range portion of the defect potential can be well described by an analytic Coulomb potential profile. This fact has been utilized to correct the defect formation energy in finite size supercell calculations⁴. Here we use this fact to recover the short-range part of the defect potential, which essentially is the central cell potential. To illustrate this, we consider n-type silicon as an example. We built a silicon supercell with a $3 \times 3 \times 3$ conventional unit cell and replaced one silicon atom with a dopant atom (phosphorous, arsenic, or antimony). The defect potential is obtained by subtracting the potential of pure silicon from the one with the dopant atom. For the latter calculation, the total charge of the supercell is taken to be $+1e$, making the dopant positively charged (corresponding to n-type). The defect potentials corresponding to the different dopants in silicon are shown in Fig. S3a.

If the defect potential is short-ranged, the potential profile far away from the defect atom (corresponding to the middle point in the plot) should be flat. However, it is clear from Fig. S3a that while different dopants show different short-range profiles, all exhibit a gradually decaying profile far away from the defect. This gradually decaying profile is due to the long-range Coulomb potential of the defect charge. If we consider the Coulomb potential of an infinite periodic array of charge Ze at locations \mathbf{R}_i corresponding to the corners of the periodic supercell⁵, the Coulomb potential at \mathbf{r} is

$$\Delta\hat{V}_{lr} = -\sum_i \frac{Ze^2}{\sqrt{|\bar{\epsilon}|}} \frac{\text{erfc}(\gamma\sqrt{(\mathbf{R}_i-\mathbf{r})\cdot\bar{\epsilon}^{-1}\cdot(\mathbf{R}_i-\mathbf{r})})}{\sqrt{(\mathbf{R}_i-\mathbf{r})\cdot\bar{\epsilon}^{-1}\cdot(\mathbf{R}_i-\mathbf{r})}} - \sum_{\mathbf{G}_i \neq 0} \frac{4\pi Ze^2}{\Omega} \frac{\exp(-\mathbf{G}_i\cdot\bar{\epsilon}\cdot\frac{\mathbf{G}_i}{4\gamma^2})}{\mathbf{G}_i\cdot\bar{\epsilon}\cdot\mathbf{G}_i} \exp(i\mathbf{G}_i\cdot\mathbf{r}) + \frac{\pi Ze^2}{\Omega\gamma^2} \quad (8)$$

Here $\bar{\epsilon}$ is the dielectric tensor computed from first principles, Ω is the supercell volume, $Z = 1$ is the defect charge, and γ is a convergence parameter for the Ewald summation. This long-range Coulomb potential is also plotted in Fig. S3a as a reference and matches well with the asymptotic trend of all defect potentials extracted from first principles. If we subtract this long-range term from the defect potential, we are then left with only a short-range component (Fig. S3b), which becomes flat away from the defect. In Fig. S3b we have also aligned the potential at the farthest distance from the defect to zero, based on its short-range nature. This shows that the above procedure enabled the extraction of the short-range defect potential from the finite-size supercell calculations. The effects of long-range Coulomb potential on electron-defect interactions can then be later added into the scattering matrix *via* an analytic expression, as explained in detail in Methods.

First-principles electron transport calculation

First-principles calculations of electron transport properties (specifically, the electrical conductivity σ and the Seebeck coefficient S) are based on the Boltzmann transport theory¹:

$$\begin{cases} \sigma = \frac{e^2}{3\Omega_0 N_k} \sum_{\mathbf{k}\alpha} v_{\mathbf{k}\alpha}^2 \tau_{\mathbf{k}\alpha} \left(-\frac{\partial f_{\mathbf{k}\alpha}^0}{\partial E} \right) \\ S = \frac{e}{3\sigma\Omega_0 N_k T} \sum_{\mathbf{k}\alpha} (E - \mu) v_{\mathbf{k}\alpha}^2 \tau_{\mathbf{k}\alpha} \left(-\frac{\partial f_{\mathbf{k}\alpha}^0}{\partial E} \right) \end{cases} \quad (9)$$

where e is the electronic charge, Ω_0 is the unit cell volume, $N_{\mathbf{k}}$ is the number of \mathbf{k} points, α is the band index, $v_{\mathbf{k}\alpha}$ is the electron group velocity, $\tau_{\mathbf{k}\alpha}$ is the electron relaxation time, E is the electron energy, μ is the Fermi level, and $f_{\mathbf{k}\alpha}^0$ is the Fermi-Dirac distribution. The electron energy and group velocity are derived from the electronic band structure. The equilibrium properties of electrons of half-Heusler materials are calculated from first principles using the QUANTUM ESPRESSO software package⁶. We use the generalized gradient approximation (GGA) of Perdew, Burke and Ernzerhof with the Troullier-Martins-type norm-conserving semilocal pseudopotential⁷ (corresponding to pbe-mt.UPF in the QUANTUM ESPRESSO pseudopotential library). A cutoff energy of 120 Ryd and a $6 \times 6 \times 6$ \mathbf{k} -mesh are used to determine the equilibrium lattice constant. The equilibrium properties of phonons and the electron-phonon interaction matrices are calculated *via* density functional perturbation theory⁸ for a $6 \times 6 \times 6$ \mathbf{q} -mesh (with a $6 \times 6 \times 6$ \mathbf{k} -mesh for the electron-phonon interaction matrix). We then use the EPW software package⁹ to interpolate the electronic information and the phonon information, as well as the electron-phonon coupling matrices to a fine mesh. A fine mesh is required to ensure that the calculation of transport properties based on Eq. (9) is converged. In Eq. (9), the electron relaxation time is determined *via* Matthiessen's rule considering both intrinsic electron-phonon scattering rates and extrinsic electron-defect scattering rates: $1/\tau = 1/\tau_{e-ph} + 1/\tau_{e-d}$. The intrinsic electron-phonon scattering rates are related to the electron-phonon interaction matrix $g(\mathbf{k}, \mathbf{k} + \mathbf{q}, \mathbf{q})$ *via*¹⁰

$$\frac{1}{\tau_k^{e-ph}} = \frac{2\pi}{\hbar} \frac{1}{N_q} \sum_{\mathbf{q}} |g(\mathbf{k}, \mathbf{k} + \mathbf{q}, \mathbf{q})|^2 \cdot \left[\begin{array}{l} (n_{\mathbf{q}} + f_{\mathbf{k}+\mathbf{q}}) \delta(E_{\mathbf{k}} - E_{\mathbf{k}+\mathbf{q}} + \hbar\omega_{\mathbf{q}}) \\ + (n_{\mathbf{q}} + 1 - f_{\mathbf{k}+\mathbf{q}}) \delta(E_{\mathbf{k}} - E_{\mathbf{k}+\mathbf{q}} - \hbar\omega_{\mathbf{q}}) \end{array} \right] \quad (10)$$

which sums over all possible scattering processes that satisfy momentum and energy conservations using a tetrahedral integration method, where $N_{\mathbf{q}}$ is the number of \mathbf{q} points, $n_{\mathbf{q}}$ is the Bose-Einstein distribution for phonons, and the delta functions indicate the energy conservation. The electron-defect scattering rates are determined by the electron-defect interaction matrix $g_{e-d}(\mathbf{k}, \mathbf{k}')$ *via*

$$\frac{1}{\tau_k^{e-d}} = N_d \Omega_0 \frac{2\pi}{\hbar} \frac{1}{N_{\mathbf{k}'}} \sum_{\mathbf{k}'} \left(1 - \frac{\mathbf{v}_{\mathbf{k}} \cdot \mathbf{v}_{\mathbf{k}'}}{|\mathbf{v}_{\mathbf{k}}| |\mathbf{v}_{\mathbf{k}'}} \right) |g_{e-d}(\mathbf{k}, \mathbf{k}')|^2 \delta(E_{\mathbf{k}} - E_{\mathbf{k}'}) \quad (11)$$

where N_d is the volume density of defects. The calculation of $g_{e-d}(\mathbf{k}, \mathbf{k}')$ has been detailed in Methods. By adding all scattering rates together, we obtain the total electron scattering rates, which are then inserted into Eq. (9) to yield the electron transport properties.

Supplementary Figure

Figure S1. Illustration of a propagating electron wave scattered by the perturbed electronic potential due to the presence of a charged p-type dopant. For a p-type dopant, the long-range Coulomb potential is repulsive to electrons (contrary to the n-type case). In this case, if the dopant atom has a smaller ionic radius than the host atom, the perturbation tends to create an attractive force for electrons, which counteracts the Coulomb potential (electron-cloaking scenario). On the other hand, if the dopant atom has a larger ionic radius, it tends to create a repulsive force, which then adds to the long-range Coulomb potential (electron-scattering scenario).

Figure S2. Defect potential extracted from first principles calculations. **a-b**, Directly extracted defect potentials for **(a)** Si-doped GaAs, and **(b)** Bi-doped PbTe, plotted from the location of the defect. In each case, the long-range part can be approximately described by the Coulomb potential. The distortions observed in the range between ~ 2 Å and ~ 7 Å are due to relaxation of atomic positions in the presence of the defect. Significant deviation from the Coulomb potential can be seen at short range (< 1 Å). The asymptotic trend given by the Coulomb field is recovered as one moves away from the defect. **c-d**, Planar averaged defect potentials for Si-doped GaAs **(c)** and Bi-doped PbTe **(d)**. The planar averaged potential is calculated by $\overline{\Delta V}(z) = \frac{1}{A} \iint_0^a dx dy \Delta \hat{V}(x, y, z)$ where a is the lattice vector length of the supercell, and A is the area of the x - y plane of the supercell, as indicated by the inset of **(c)**. The z axis starts from the plane containing the defect.

Figure S3. Illustration of subtraction of the long-range Coulomb potential from *ab initio* defect potential profiles. **a**, Uncorrected defect potentials for silicon doped with different n-type dopants (P, As, or Sb, as indicated in the figure legend) directly extracted from first-principles calculations, compared with the analytic long-range Coulomb potential calculated based on Eq. (8) in Supplementary Note. The three-dimensional defect potentials are projected as a function of the distance from the defect, located at the center of the supercell. **The asymptotic trend given by the Coulomb field is recovered as one moves away from the defect.** **b**, Corrected defect potentials, showing their short-range nature. Here, in calculating the defect potential, we have ignored the atomic relaxation to emphasize the long-range decaying part. The potentials at the farthest location away from the defect have also been aligned to zero.

Figure S4. Electron transport simulation of SrTiO₃. **a**, Computed phonon dispersion of SrTiO₃. Dashed lines are results from density functional perturbation theory, which gives rise to imaginary phonon frequencies. The solid lines are obtained by fitting force constants to a force-displacement dataset from *ab initio* molecular dynamics study¹¹ using VASP package. The fitting was performed using ALAMODE package¹². The latter method correctly reproduces the stable phonon modes at 300 K. **b-c**, Defect potentials for **(b)** La dopant on Sr site, and **(c)** Nb dopant on Ti site. **d**, Comparison of electron-defect scattering rates for La and Nb dopants at the carrier concentration of $6 \times 10^{19} \text{ cm}^{-3}$. The defect potential includes both the short-range perturbation and long-range Coulomb potential. **e**, Mobilities considering different scattering conditions (Coulomb scattering only, and those that consider the full defect potential corresponding to La and Nb dopants), and comparison between simulation and experiment¹³. The calculated mobilities assume a carrier concentration of $6 \times 10^{19} \text{ cm}^{-3}$.

Figure S5. Illustrations of defect potentials in n-type and p-type materials leading to electron-scattering and electron-cloaking scenarios. **a**, The electron-cloaking effect is achieved in n-type material when the Coulomb potential is counterbalanced by a repulsive central cell potential from a dopant. **b-c**, Calculated defect potential for n-type **(b)** V-doped ZrNiSn (electron-scattering scenario) and **(c)** Nb-doped TiNiSn (electron-cloaking scenario). **d**, The electron-cloaking effect is achieved in p-type material when the Coulomb potential is counterbalanced by an attractive central cell potential from a dopant. **e-f**, Calculated defect potential for p-type **(e)** Sc-doped ZrCoSb (electron-cloaking scenario) and **(f)** Y-doped TiCoSb (electron-scattering scenario). Here, in presenting the defect potential, we have ignored the atomic relaxation to focus on the short-range defect potential. Coulomb potentials are also plotted in **(b-c)** and **(e-f)**, to show that the asymptotic trend given by the Coulomb field is recovered as one moves away from the defect.

Figure S6. Electron transport details in p-type NbFeSb. **a**, Scattering rates for holes in p-type Ti-doped NbFeSb, showing that the short-range defect potential leads to a significant reduction in the scattering rates compared to those limited by Coulomb scattering. The energy is relative to the valence band edge. **b**, Charge carrier mean free paths as a function of energy in Ti-doped NbFeSb at a carrier concentration of $2 \times 10^{20} \text{ cm}^{-3}$. While Coulomb scatterings severely limit the mean free paths, the central cell potential can counteract this effect and increase the carrier mean free paths by almost a factor of two near the band edge. **c**, Calculated defect potential of p-type dopants (Ti, Zr, and Hf) in NbFeSb. The larger power factors obtained with Ti and Hf dopants can be understood based on their attractive short-range potential which counterbalances the long-range Coulomb scatterings.

Figure S7. Thermoelectric power factor of p-type Ti-doped TaFeSb at room temperature. Both curves consider intrinsic electron-phonon interactions. The dashed curve includes electron scatterings by the Coulomb potential, while the solid curve considers electron scatterings by the full defect potential. Due to the counteraction between the strong central cell potential of the Ti dopant and the Coulomb potential, the full defect potential leads to an enhanced optimal power factor.

Figure S8. Thermoelectric power factor at 150 K. a, p-type Ti-doped NbFeSb. b, p-type Ti-doped TaFeSb. For both plots, the red curve only considers electron scatterings by phonons and by the Coulomb potential, while the blue curve considers electron scatterings by phonons and the full defect potential (including the short-range part). Due to the counteraction between the strong central cell potential of the Ti dopant and the Coulomb potential, a larger enhancement in power factor is observed at lower temperatures.

Table S1. Dielectric constants of representative oxides and thermoelectric materials.

Material	Dielectric constant (relative)
ZrO ₂	29 ¹⁴
HfO ₂	25 ¹⁴
Ta ₂ O ₅	26 ¹⁴
La ₂ O ₃	30 ¹⁴
LaAlO ₃	30 ¹⁴
Nb ₂ O ₅	35 ¹⁴
TiO ₂	95 ¹⁴
Bi ₂ Te ₃	290 (, 15K) ¹⁵
PbTe	414 ¹⁵
SnSe	42 (c axis), 45 (a axis) ¹⁵
NbFeSb	45 ¹⁶
Mg ₃ Sb ₂	32 ¹⁶

Reference

1. Lundstrom, M. *Fundamentals of Carrier Transport*. (Cambridge University Press, 2009).
2. Cohen-Tannoudji, C., Diu, B. & Laloe, F. *Quantum Mechanics*. (Wiley, 1992).
3. Fetterman, W., Osborne, E. & Saxon, D. S. A numerical solution of Schrodinger's equation in the continuum. *Journal of Research of the National Bureau of Standards* **52**, (1954).
4. Freysoldt, C. *et al.* First-principles calculations for point defects in solids. *Rev. Mod. Phys.* **86**, 253–305 (2014).
5. Kumagai, Y. & Oba, F. Electrostatics-based finite-size corrections for first-principles point defect calculations. *Phys. Rev. B* **89**, 195205 (2014).
6. Giannozzi, P. *et al.* QUANTUM ESPRESSO: a modular and open-source software project for quantum simulations of materials. *J. Phys.: Condens. Matter* **21**, 395502 (2009).
7. Perdew, J. P., Burke, K. & Ernzerhof, M. Generalized gradient approximation made simple. *Phys. Rev. Lett.* **77**, 3865–3868 (1996).
8. Baroni, S., de Gironcoli, S., Dal Corso, A. & Giannozzi, P. Phonons and related crystal properties from density-functional perturbation theory. *Rev. Mod. Phys.* **73**, 515–562 (2001).
9. Giustino, F., Cohen, M. & Louie, S. Electron-phonon interaction using Wannier functions. *Phys. Rev. B* **76**, 165108 (2007).
10. J. M. Ziman. *Electrons and Phonons: The Theory of Transport Phenomena in Solids*. (Clarendon Press, 1960).
11. Hellman, O., Steneteg, P., Abrikosov, I. A. & Simak, S. I. Temperature dependent effective potential method for accurate free energy calculations of solids. *Phys. Rev. B* **87**, 104111 (2013).
12. Tadano, T., Gohda, Y. & Tsuneyuki, S. Anharmonic force constants extracted from first-principles molecular dynamics: applications to heat transfer simulations. *J. Phys.: Condens. Matter* **26**, 225402 (2014).
13. Han, W. *et al.* Spin injection and detection in lanthanum- and niobium-doped SrTiO₃ using the Hanle technique. *Nat. Comm.* **4**, 2134 (2013).
14. Azadmanjiri, J. *et al.* A review on hybrid nanolaminate materials synthesized by deposition techniques for energy storage applications. *J. Mater. Chem. A* **2**, 3695–3708 (2014).
15. *Non-Tetrahedrally Bonded Elements and Binary Compounds I*. vol. 41C (Springer-Verlag, 1998).
16. J. Slade, T. *et al.* Understanding the thermally activated charge transport in NaPb_mSbQ_{m+2} (Q = S, Se, Te) thermoelectrics: weak dielectric screening leads to grain boundary dominated charge carrier scattering. *Energy & Environmental Science* **13**, 1509–1518 (2020).

REVIEWER COMMENTS

Reviewer #1 (Remarks to the Author):

The revised manuscript by Jiawei Zhou and co-workers has greatly improved in clarity. I'm now persuaded that signatures of the cancellation of scattering by a reversed short-range defect potential can be found in real materials. These ideas are important. They will attract follow-on work and further debate in the literature, so I can recommend publication.

However, I'd still suggest the authors to reconsider the following in their future revision:

While it is nice to search for a simple parameter to correlate with the extent of cancellation, I'm not sure that 'core size' (which the author calls 'ionic size') is the best one.

There seem to me at least two distinct effects:

(a) Size effect. While the cartoon of Fig 1 and Fig S1 depicts a larger core size gives rise to more negative core potential, and vice versa, I don't think this must be generally true. The sign of the defect potential in the core region depends on the difference in core electron density of the defect compared to host atom.

(b) Electronic structure effect. Thus, there should be stronger atomic number effect. For n-doping, I think it may go something like this. In general, the n-dopant is one periodic table column to the right of the host. (i) If the n-dopant is on same row, proton number increases by one, accompanied by a slight contraction of core electron density. This causes core size to decrease, which builds up a slight negative defect potential in the core. (ii) If however the dopant is one row above host, the number of protons decreases by 17 (in the transition block), accompanied by outward expansion in the counterbalancing core electron density, but reduction in core size, which causes a strong positive potential spike at the nucleus that is counteracted by a tendency for negative defect potential outside of the nucleus. (iii) If the dopant is one row below host, the number of protons increases by 17, accompanied by inward contraction of the counterbalancing core density, but expansion in core size, which causes a strong negative potential spike at the nucleus that is counteracted by a tendency for positive defect potential outside. Whether the counteraction is sufficient to reverse the potential depends on the atom itself.

This description appears consistent with the calculations in Figs S2 to S5.

Thus, the defect potential trend may be mainly an atomic number effect, which confounds with size, but not always in same direction. Further, the calculations of Fig S5 compares a dopant before the lanthanide series contraction, and a dopant after, which may miss this effect.

Consider the following from the manuscript:

(i) n-doped Si with P or As. P^{5+} core is smaller than Si^{4+} but As^{5+} core is bigger. By the rule in the manuscript (bigger core makes better dopant for n-type), the latter should give higher mobility at high doping. In actual case, the former gives the higher mobility (marginally), and Fig S3b does show P gives the positive defect potential.

(ii) n-doped $SrTiO_3$ with $Sr^{2+}:La^{3+}$ or $Ti^{4+}:Nb^{5+}$. La^{3+} is smaller than Sr^{2+} (103A vs 118A; Shannon effective radii, 1976), but Nb^{5+} is larger than Ti^{4+} (64A vs 60.5A). By the rule in the manuscript, Nb doping should be better, but in fact, La doping is better.

(iii) Fig 3a shows very weak global correlation between electron-defect interaction matrix with ionic size difference, apart from the two extreme points, but correlation is better within each semicond family...

Revision report for MS# NCOMMS-21-20659A

Mobility enhancement in heavily doped semiconductors *via* electron cloaking

Jiawei Zhou, Hangtian Zhu, Qichen Song, Zhiwei Ding, Jun Mao, Zhifeng Ren, Gang Chen

We thank the reviewer for his/her further suggestions that have helped us to improve the manuscript. Our responses and revisions (in blue) are elaborated on below. In addition, the revised parts in the main manuscript and supplementary information are also highlighted (in yellow). The revised manuscript and supplementary materials are attached in this document, following our revision report.

Reviewer #1:

The revised manuscript by Jiawei Zhou and co-workers has greatly improved in clarity. I'm now persuaded that signatures of the cancellation of scattering by a reversed short-range defect potential can be found in real materials. These ideas are important. They will attract follow-on work and further debate in the literature, so I can recommend publication.

Response: We thank the reviewer for his positive comments and his further suggestions regarding the physical interpretation of the defect potential. In this revised version we have provided additional data on the defect potential, and further clarified our reasoning behind using the ionic radius as the major indicator to distinguish between defects.

However, I'd still suggest the authors to reconsider the following in their future revision:

While it is nice to search for a simple parameter to correlate with the extent of cancellation, I'm not sure that 'core size' (which the author calls 'ionic size') is the best one.

There seem to me at least two distinct effects:

(a) Size effect. While the cartoon of Fig 1 and Fig S1 depicts a larger core size gives rise to more negative core potential, and vice versa, I don't think this must be generally true. The sign of the defect potential in the core region depends on the difference in core electron density of the defect compared to host atom.

(b) Electronic structure effect. Thus, there should be stronger atomic number effect. For n-doping, I think it may go something like this. In general, the n-dopant is one periodic table column to the right of the host. (i) If the n-dopant is on same row, proton number increases by one, accompanied by a slight contraction of core electron density. This causes core size to decrease, which builds up a slight negative defect potential in the core. (ii) If however the dopant is one row above host, the number of protons decreases by 17 (in the transition block), accompanied by outward expansion

in the counterbalancing core electron density, but reduction in core size, which causes a strong positive potential spike at the nucleus that is counteracted by a tendency for negative defect potential outside of the nucleus. (iii) If the dopant is one row below host, the number of protons increases by 17, accompanied by inward contraction of the counterbalancing core density, but expansion in core size, which causes a strong negative potential spike at the nucleus that is counteracted by a tendency for positive defect potential outside. Whether the counteraction is sufficient to reverse the potential depends on the atom itself.

Response: We thank the reviewer for raising these questions and here we provide more detailed reasonings behind using ionic radius as the major indicator. As the reviewer pointed out, in addition to the size effect, the variation of core electron density also affects the defect potential, and there should be a strong atomic number effect on the sign of the defect potential.

First of all, we generally agree with the reviewer's picture about the dependence of defect potential on the core electron density. As the reviewer mentioned, due to contraction (or expansion) of core electron density for different atomic numbers, the defect potential (if the dopant is not at the same row with the host) would tend to have a spike near the nucleus and reverses its sign slightly away from the nucleus. In fact, in some simulation results we have observed such trends (e.g. SrTiO₃:Nb in Fig. S4d and TiCoSb:Y in Fig. S5f) – the defect potential does appear to have a spike near the nucleus (which tend to be positive for dopants above the host, and negative for dopants below the host).

Nonetheless, we think between the ionic size effect and the core electron density effect, the former plays a more important role in governing the electron-defect scattering. First, while we agree with the general picture the reviewer mentioned, our simulation results in most cases do not show a strong spike near the nucleus in the potential profile, and instead show a profile that correlates better with the ionic size effect. Specifically, if the core electron density effect as the reviewer mentioned were a dominant effect, the defect potential for dopants with smaller atomic numbers would have a positive spike near the center and develop negative values slightly away from the center (case ii mentioned by the reviewer). While we do sometimes observe a strong positive spike followed by negative potential for dopants with smaller atomic numbers (e.g. Ti:Nb in SrTiO₃ in Fig. S4d), usually we only observed a negative potential (e.g. Si-doped GaAs in Fig. S2a; Nb-doped TiNiSn in revised Fig. S6c), or dominantly negative potential with scattered small positive values near the center (V-doped ZrNiSn in revised Fig. S6b). Similarly, for dopants with larger atomic numbers (case iii mentioned by the reviewer), we usually only observed a positive potential (As- and Sb-doped Si in the inset of revised Fig. S3a, also see responses below; Nb-doped TiNiSn in revised Fig. S6c), or dominantly positive potential with scattered small negative values near the center (e.g. Y-doped TiCoSb in revised Fig. S6f). For dopants on the same row with the host (case i mentioned by the reviewer), while the increase of proton number by one would result in a more negative potential, this is the conventional long-term Coulomb potential of the charged defect and should not be considered as the short-range perturbation. In fact, the short-range perturbation would come from the slight contraction of the core electron density that would give a slight positive potential outside the nucleus. However, for such case we usually only observed a dominant negative potential (see examples of V-doped TiNiSn and Nb-doped ZrNiSn in Fig. R1b-c below), which is consistent with the ionic size picture. In general, the sign of the defect potential in the

examples above do not follow the picture suggested by the core electron density effect, but is consistent with the ionic size picture.

To further demonstrate the validity of ionic size in understanding the defect potential, here we plotted the defect potentials for some dopant/host pairs in the half Heusler material family (Fig. R1). For each material, we presented defect potentials for two different dopants for the same element. Within each material, we can see that whenever the dopant has a larger ionic size, the defect potential becomes relatively more positive (or less negative). The same can also be found for defect potentials we already presented in the supplementary materials (Figure S7c, p-type NbFeSb).

Figure R1. Calculated defect potential for n-type Ta-doped ZrNiSn (a), Nb-doped ZrNiSn (b), V-doped TiNiSn (c), Nb-doped TiNiSn (d), and p-type Sc-doped ZrCoSb (e), Y-doped ZrCoSb (f), Sc-doped TiCoSb (g), and Y-doped TiCoSb (h). The ionic sizes are drawn in the inset (same as Figure 3a). Here, in presenting the defect potential, we have ignored the atomic relaxation to focus on the short-range defect potential.

We believe the examples above indicate that the ionic size is a good indicator for the sign of the defect potential. One possible reason that the core electron density variation does not exhibit a stronger effect on the defect potential is that inside the core the defect potential is not only affected by the usual Coulomb interaction between the nucleus and electrons, but also exchange-correlation interactions that are inherent in the many-body electron systems (which result from Pauli exclusion). When discussing the defect potential near the core, especially the sign of its short-range part, we think the exchange-correlation interaction is a more important governing factor than the changes caused by the variation in core electron density due to Coulomb interactions. Because the Pauli exclusion effect is better correlated with the ionic size (as we illustrated in Fig. 1), a stronger exchange-correlation interaction inside the core may lead to defect potential profiles that are better correlated with the ionic size.

Revision: On page 6-7 we have added more discussions on the reasoning behind using ionic radius as the major indicator for the sign of the defect potential:

“In addition, as the electron approaches close to the center, it will experience difference in its interactions with the core electrons as the latter now are being held by a different atom. Such interactions involve both Coulomb interactions and exchange-correlation interactions, the latter of which are inherent to the many-body electron system and a cause of this is Pauli exclusion. On one hand, the Coulomb interactions between the nucleus and electrons will cause the core electrons to contract or expand depending on the atomic number of the dopant relative to the host, which would lead to variations in the core potential. In the case of P-doped Si, P^{+5} has a higher proton number than Si^{+4} , which leads to a slight contraction of the core electrons. This core electron contraction (and correspondingly higher core electron density) will tend to give a slight positive defect potential near the center, as observed in Supplementary Fig. 3 (inset). On the other hand, Pauli exclusion (or exchange-correlation interactions) forces the valence electrons to stay outside the core region (in the sense that the valence electron wavefunctions are orthogonal to core electrons). Therefore, when the dopant ion has a smaller size than the host, propagating electron states (which are composed of mainly valence electrons) will have deeper penetration into the core, which is equivalent to creating an attractive short-range defect potential. For P-doped Si, because P^{+5} has a smaller ionic size than Si^{+4} , the defect potential becomes negative slightly away from the center (Supplementary Fig. 3, inset).

The above-mentioned effects lead to different trends of defect potential profile in the periodic table. The variation of core electron density mostly depends on the atomic number, while the Pauli exclusion effect is largely determined by the ionic size. While both will affect the defect potential, here we argue that the short-range core potential is better described by the ionic size difference between the dopant and host atom. This is because within and around the core region, the exchange-correlation interaction can be significant. When discussing the defect potential, especially the sign of its short-range part, we think Pauli exclusion is a more important governing factor, which is reasonably captured by the ionic radius picture to the first order. In general, when the dopant ion has a smaller size than the host, the defect potential tends to be dominantly negative (attractive potential to allow electrons to penetrate deeper into the core, Fig. 1b). Conversely, if a dopant ion has a larger size than the host, it will tend to repel electrons due to Pauli exclusion and tend to be dominantly positive (repulsive potential, Fig. 1c). We will later show that in most cases our simulation results agree better with this ionic size picture. We therefore will use the ionic radius as the major indicator to infer the sign of short-range defect potential.”

In later sections (page 13) when we establish the correlation between electron-defect scattering matrix and the ionic radius, we also added

“Supplementary Figure 5 further shows the defect potentials for a few dopant/host pairs in different materials, demonstrating that indeed whenever the dopant has a larger ionic size the defect potential becomes relatively more positive (this trend is also observed comparing dopants before and after the lanthanide contraction, e.g. Ta and Nb).”

and have included Figure R1 as an additional supplementary figure (supplementary Figure 5) in the supplementary materials.

This description appears consistent with the calculations in Figs S2 to S5. Thus, the defect potential trend may be mainly an atomic number effect, which confounds with size, but not always in same

direction. Further, the calculations of Fig S5 compares a dopant before the lanthanide series contraction, and a dopant after, which may miss this effect.

Response: We would like to acknowledge that the mechanism the reviewer mentioned about the core electron density effect is a contributing factor. However, the potential profile the reviewer pointed out does not always agree with our calculations as we explained above. Instead, we found that our simulation results in most cases instead agree well with the ionic size picture. We think the reason is that the defect potential is better captured by the ionic radius due to exchange-correlation interactions near the core instead of the atomic number, as we explained above. While we have shown the correlation between electron-defect scattering matrix and the ionic radius in Figure 3a, one additional evidence can be found by comparing dopants before and after the lanthanide series contraction (Ta-doped ZrNiSn in Fig. R1a and Hf-doped NbFeSb in revised Fig. S7c). The reviewer pointed this out as an exception, but we instead think this is further evidence supporting our claim. For example, Ta has a larger atomic number than Zr, and if it were for the atomic number, Ta-doped ZrNiSn should exhibit a strong positive defect potential outside the nucleus. Instead, ZrNiSn:Ta defect potential is dominantly negative (Fig. R1a), and this is because Ta has a smaller ionic radius than Zr (due to lanthanide series contraction). Similar behaviors can also be seen comparing Hf and Zr dopants in NbFeSb (Hf has a more negative potential due to its smaller ionic radius, Figure S7c). Considering all above, we think the ionic radius difference instead of the atomic number is a better indicator for the sign of the defect potential. This additional point regarding dopants before and after the lanthanide contraction has been mentioned in the revision (see revision to the previous comment).

Consider the following from the manuscript:

(i) n-doped Si with P or As. P^{5+} core is smaller than Si^{4+} but As^{5+} core is bigger. By the rule in the manuscript (bigger core makes better dopant for n-type), the latter should give higher mobility at high doping. In actual case, the former gives the higher mobility (marginally), and Fig S3b does show P gives the positive defect potential.

Response: First we would like to mention that near the center As dopant has a more positive potential than P dopant, consistent with our ionic radius picture. The original scale in Fig. S3 was small to make it clear that the long-range tail follows the Coulomb potential behavior, and in Figure R2 below we re-plotted the defect potentials with a larger energy scale. Slightly away from the center, the defect potential for As dopant indeed starts to have a more negative value, as the reviewer mentioned. This is possibly due to the electron response to the strong potential perturbation.

Figure R2. Defect potential of P, As and Sb dopants in silicon. The trend of the short-range portion of the defect potential agrees with the ionic radius picture – dopants with larger ionic size tend to create more positive defect potentials near the center.

Our picture was intended to describe the qualitative trend between different dopants. For silicon however, it is known that the dopant effects are marginal. We note that the electron-defect scattering strengths are ultimately determined by the scattering matrix (not the defect potential alone), which are spatial products of electron wavefunctions and the defect potential (we will have more explanations about this point in the response to the last comment). The marginal dopant effects in silicon suggest that these defect scattering matrices have small values. In such case, the sign of the defect potential may no longer be a good indicator for the defect scattering, and the mobility with different dopants will also depend on the actual overlap between electrons and the defect potential.

Revision: We have revised supplementary Figure 3 and included Figure R2 as an inset. In the manuscript we revised the corresponding sentence (page 11) to

“The key observation that enabled our first-principles computation is that the long-range part of the defect potential can be well described by an analytic Coulomb potential profile (Supplementary Note and Supplementary Fig. 3, where the trend of the short-range defect potentials of dopants in silicon are also found to agree with the ionic radius picture).”

(ii) n-doped SrTiO₃ with Sr²⁺:La³⁺ or Ti⁴⁺:Nb⁵⁺. La³⁺ is smaller than Sr²⁺ (103Å vs 118Å; Shannon effective radii, 1976), but Nb⁵⁺ is larger than Ti⁴⁺ (64Å vs 60.5Å). By the rule in the manuscript, Nb doping should be better, but in fact, La doping is better.

Response: Our ionic radius picture was intended to describe the qualitative trend between different dopants and used as an indicator. When the ionic radius difference is small, the sign and magnitude of the defect potential can be more sensitive to the actual electron density profile. For Nb and Ti, their ionic radii are close. The actual defect potential in such case should be calculated from first principles. Our first principles calculation indicate Nb dopant has a negative defect potential, which adds to the negative Coulomb potential that further increase the electron scatterings.

In addition, we should mention that when comparing dopants on different atomic sites, the relative projections of the electron wavefunctions on the corresponding atomic sites are also important (we will have more explanations about this point in the response to the last comment). From the

projected electronic density-of-states (Figure R3) we know that electrons near the conduction band edge mostly consist of d orbitals on Ti atom. Therefore, any perturbation due to dopants on Ti would be more significant than dopants on other atoms. This is another contributing factor that leads to the higher defect scattering rates in Nb dopant (on Ti site) compared to La dopant (on Sr site).

Figure R3. Projected electronic density-of-states in SrTiO₃. The energy zero is taken at the conduction band edge. Only projections onto the d orbitals of Ti and Sr atoms are shown.

Revision: We have included additional discussion to clarify that the electronic projections also contribute to the defect scattering when comparing dopants on different atomic sites (page 12):

“We should mention that in both cases the dopant has an ionic size similar to the host, and therefore the sign of the defect potential is more sensitive to the actual electron density profile and needs to be determined by the calculation. In addition, La and Nb dopants are located at different atomic sites. In such case, the actual projection of valence electron wavefunctions on given atomic sites also influence the magnitude of the electron-defect scattering matrix (more discussions about this will be given later). For SrTiO₃, electrons near the conduction band edge mostly consist of d orbitals on Ti (Supplementary Fig. 4), which strongly overlap with the perturbation caused by Nb dopant. This further increases the electron-defect scatterings from Nb dopant.”

and have included Figure R3 into the revised Supplementary Figure 4.

(iii) Fig 3a shows very weak global correlation between electron-defect interaction matrix with ionic size difference, apart from the two extreme points, but correlation is better within each semicond family...

Response: The reason that the correlation between electron-defect interaction matrix and the ionic size difference is better within each material family is that the interaction matrix also depends on how the electron wavefunctions spatially overlap with the defect site (and therefore on the actual spatial variation of the wavefunction). To understand this point, we first note that the electron-

defect interaction matrix is a spatial product of the defect potential and the electronic wavefunctions

$$\langle \psi_{k'} | \Delta \hat{V} | \psi_k \rangle = \int d\mathbf{r} \psi_{k'}^*(\mathbf{r}) \Delta \hat{V}(\mathbf{r}) \psi_k(\mathbf{r}) \quad (1)$$

Usually, the electronic wavefunction can be approximately expressed by a linear combination of atomic orbitals on different atomic sites

$$\psi_{\mathbf{k}}(\mathbf{r}) = e^{i\mathbf{k}\cdot\mathbf{r}} \sum c_{\alpha i} \phi_{\alpha}^i(\mathbf{r}) \quad (2)$$

where ϕ_{α}^i denotes the i -th atomic orbital on atom α and has significant non-zero values only around atom α . With this, we have

$$\langle \psi_{k'} | \Delta \hat{V} | \psi_k \rangle = e^{i(\mathbf{k}-\mathbf{k}')\cdot\mathbf{r}} \sum_{\alpha i, \beta j} c_{\alpha i} c_{\beta j}^* \int d\mathbf{r} \phi_{\beta}^{j*}(\mathbf{r}) \Delta \hat{V}(\mathbf{r}) \phi_{\alpha}^i(\mathbf{r}) \quad (3)$$

We denote the defect site as γ . Because the defect potential $\Delta \hat{V}(\mathbf{r})$ is significant only around the defect site, it is then clear that in addition to the magnitude of the defect potential, the electron-defect interaction also depends on the projection of wavefunctions on the defect site (otherwise ϕ_{α}^i and $\Delta \hat{V}$ would have minimal spatial overlap and the spatial integration will be close to zero), which are described by the pre-factors $c_{\alpha i} c_{\beta j}^*$. For the same defect potential, wavefunctions with larger projected density-of-states on the defect site (namely large non-zero values of $c_{\gamma i}$) will lead to stronger electron-defect interactions, and vice versa.

Within each material family, the wavefunctions for the host material are the same (namely the projections on the defect site, described by $c_{\gamma i}$, are the same), and therefore a better correlation can be observed between the interaction matrix and the ionic size difference. For different materials, the wavefunctions can be very different, and therefore even for defect potential with similar strengths the interaction matrix can also be different (due to different projections, or $c_{\gamma i}$). The important point of Fig. 3a is that, even though different materials have different wavefunctions, it turns out the defect interaction matrix can be reasonably described by a single ionic size difference descriptor. We think this is strong evidence that the ionic size is indeed a good indicator for the general understanding of electron-defect interaction and can be used as a starting point for dopant selection in semiconductor materials.

Revision: We have revised relevant paragraphs (page 13-14) in the manuscript to emphasize that the projection of electronic wavefunction on a given atomic site also contributes to the electron-defect scattering strength, and is the reason why the correlation appears better within each semiconductor family:

“Here, we mention that because the sEDI matrix is a spatial product of defect potential and the electronic wavefunction, sEDI values will also depend on the projection of electronic wavefunctions on the dopant atom (see more discussions in Supplementary Note). For different materials, even when defect potentials are similar, the varying electronic wavefunctions and their spatial profiles would lead to variation in the sEDI values, and this is the reason why the correlation appears better within each material family.”

and have added additional discussions in the Supplementary Note about the spatial projection of the electron-defect interaction

“Spatial projection of electron-defect interaction

While in the main text we mainly discussed the defect potential and its trend in the periodic table based on the ionic radius, the actual electron-defect interaction strength is governed by the electron-defect interaction matrix, which is a spatial product of the defect potential and the electronic wavefunctions

$$\langle \psi_{k'} | \Delta \hat{V} | \psi_k \rangle = \int d\mathbf{r} \psi_{k'}^*(\mathbf{r}) \Delta \hat{V}(\mathbf{r}) \psi_k(\mathbf{r}) \quad (12)$$

Usually, the electronic wavefunction can be approximately expressed by a linear combination of atomic orbitals on different atomic sites

$$\psi_k(\mathbf{r}) = e^{ik \cdot \mathbf{r}} \sum c_{\alpha i} \phi_{\alpha}^i(\mathbf{r}) \quad (13)$$

where ϕ_{α}^i denotes the i -th atomic orbital on atom α and has significant non-zero values only around atom α . With this, we have

$$\langle \psi_{k'} | \Delta \hat{V} | \psi_k \rangle = e^{i(k-k') \cdot \mathbf{r}} \sum_{\alpha i, \beta j} c_{\alpha i} c_{\beta j}^* \int d\mathbf{r} \phi_{\beta}^j(\mathbf{r}) \Delta \hat{V}(\mathbf{r}) \phi_{\alpha}^i(\mathbf{r}) \quad (14)$$

We denote the defect site as γ . Because the defect potential $\Delta \hat{V}(\mathbf{r})$ is significant only around the defect site, it is then clear that in addition to the magnitude of the defect potential, the electron-defect interaction also depends on the projection of wavefunctions on given atomic sites, which are described by the pre-factors $c_{\alpha i} c_{\beta j}^*$. For the same defect potential, wavefunctions with larger projected density-of-states on the defect site (namely large non-zero values of $c_{\gamma i}$) will lead to stronger electron-defect interactions, and vice versa.”

Mobility enhancement in heavily doped semiconductors *via* electron cloaking

Jiawei Zhou^{1*}, Hangtian Zhu², Qichen Song¹, Zhiwei Ding¹, Jun Mao², Zhifeng Ren², Gang Chen^{1*}

¹*Department of Mechanical Engineering, Massachusetts Institute of Technology, Cambridge, MA 02139, USA*

²*Department of Physics and Texas Center for Superconductivity at the University of Houston (TcSUH), University of Houston, Houston, TX 77204, USA*

Authors to whom correspondence should be addressed.

* Electronic mail: jwzhou@stanford.edu, gchen2@mit.edu

Doping is central for solid-state devices from transistors to thermoelectric energy converters^{1,2}. The interaction between electrons and dopants plays a pivotal role in carrier transport^{3,4}. Conventional theory suggests that the Coulomb field of the ionized dopants^{5,6} limits the charge mobility at high carrier densities, and that either the atomic details of the dopants are unimportant or the mobility can only be further degraded, while experimental results often show that dopant choice affects mobility. In practice, the selection of dopants is still mostly a trial-and-error process. Here we demonstrate, *via* first-principles simulation and comparison with experiments, that a large short-range perturbation created by selected dopants can in fact counteract the long-range Coulomb field, leading to electron transport that is nearly immune to the presence of dopants. Such “cloaking” of dopants leads to enhanced mobilities at high carrier concentrations close to the intrinsic electron-phonon scattering limit. We show that the ionic radius can be used to guide dopant selection in order to achieve such an electron-cloaking effect. Our finding provides guidance to the selection of dopants for solid-state conductors to achieve high mobility for electronic, photonic, and energy conversion applications.

Doping is a fundamental strategy employed to control the electrical conductivity of semiconductors¹. Conventional understanding based on the theory originally proposed by Brooks and Herring^{5,6} states that electrons are strongly scattered by the long-range Coulomb field of the charged dopant, leading to reduced mobility. With further generalization to consider multiple scatterings, electron-electron interactions, and dielectric screening^{3,7,8}, the Brooks-Herring theory has been successfully used to explain the reduced charge mobility in conventional semiconductors such as silicon and III-V semiconductors at low to intermediate doping concentrations. An

important consequence of this theory is that different dopants with the same charge have the same impact on the electron transport, as the theory neglects the atomic details³. While the effect of a dopant's chemical nature has been recognized in the past, most models have assumed an empirical potential profile and treated such a chemical effect as a perturbation to the Coulomb field, as indicated by the “central cell correction”^{9–11}. A prevailing view is that such correction could only further reduce charge mobility due to the strong local interactions between electrons and defects. Experimentally, different impacts on carrier mobility resulting from different dopants are often observed^{9,12,13}, suggesting that their atomic details play a significant role in governing the electron-defect interactions. The impact of dopant scattering on mobility can be particularly large at high carrier concentrations, as is often observed in solid oxide materials¹⁴, transparent conductors¹⁵, and thermoelectric compounds¹⁶, with a carrier concentration close to or above $\sim 10^{20} \text{ cm}^{-3}$. Despite the crucial role of dopants in governing the carrier transport, dopant selection has thus far been mostly a trial-and-error process.

Recent advancements in *ab initio* simulations have enabled quantification of electron energy or potential changes induced by charged defects, allowing the engineering of defects from first principles^{17–19}. These studies have revealed that a charged defect potential can significantly deviate from the conventional Coulomb field assumption^{17–19}. However, how such atomic details impact charge mobility remains unknown, mainly due to the lack of capability to treat the long-range Coulomb field and short-range electron interactions on an equal footing. Here we employ a new computational approach to treat Coulomb and short-range interactions simultaneously and apply it to heavily doped semiconductors. Building on the recent development in the formation energy calculations for charged defects⁴, we take into account short-range perturbations from *ab initio* calculations while incorporating the long-range potential to far distances *via* analytic

expression. Our approach overcomes the obstacles that have previously prevented direct quantification of electron interactions with charged defects, allowing us to examine the impact of short-range interactions on electron transport. We discovered that the deviation of defect potential from the Coulomb field at short range can lead to strong electron-defect interactions, particularly at high carrier concentrations. We further demonstrate, in contrast to the conventional belief that charge mobility in extrinsic semiconductors is limited by Coulomb scatterings, that the chemical nature of dopants can be harnessed to break this barrier, leading to effective electron cloaking and enhanced mobility close to the intrinsic electron-phonon scattering limit. While modeling suggested that a core-shell nanoparticle with proper potential can be cloaked²⁰, we found here that the central cell effect of selected dopants allows them to actually cloak themselves, achieving electron cloaking *via* doping. We show that ionic radius can guide the choice of proper dopants to realize the cloaking effect, hence providing direction in the dopant selection to achieve high mobility.

Electron-defect interaction

Central to the electron-defect interactions is the perturbed electronic potential $\Delta\hat{V}$ resulting from the presence of defects. Representative defect potential profiles are shown in Fig. 1a, in which the potential at longer distances can be approximated by the Coulomb field of a point charge (here we use a representative n-type dopant, which gives rise to an attractive Coulomb potential, as an example), while deviations occur close to the defect (represented by a simplified rectangular profile). Two scenarios, corresponding to attractive and repulsive short-range potentials, are depicted. Such short-range deviations are traditionally treated using empirical potentials, known as central cell correction^{3,9-11}, to investigate their effect on electron dynamics. It has been found that the dominant electron-defect interactions are due to the long-range Coulomb field and the

atomic details mostly introduce perturbations to the electron binding energy or scattering rates^{3,9-11}, which only further degrades the mobility. The central cell effects are usually weak, and have not been harnessed to help achieve high electron mobility.

However, if a large central cell potential exists and has a sign opposite to that of the Coulomb potential, it could counteract the scatterings due to the Coulomb field and enhance electron mobility. The reason is that the electron defect scattering is governed by the magnitude of the electron-defect interaction (EDI) matrix (to the first order), given by the spatial integration of the defect potential – $\langle \psi_{\mathbf{k}'} | \Delta \hat{V} | \psi_{\mathbf{k}} \rangle$, where $\psi_{\mathbf{k}}$ is the electronic wavefunction with wave vector \mathbf{k} . Both central cell part and long-range Coulomb part of the defect potential $\Delta \hat{V}$ contributes to the EDI matrix. When the central cell potential has an opposite sign to that of the Coulomb potential, it will lead to cancelling terms in the above spatial integration, and correspondingly a small EDI matrix, which implies weak or negligible electron-defect scattering. Note that some of past literature calculated the scattering rate due to the central cell and the long-range Coulomb contribution separately, and added the two rates according to the Matthiessen's rule. Such a treatment neglects the coherence effect of electron waves. In this treatment, the inclusion of the central cell potential always leads to a higher scattering rate. Conceptually, this is not the correct treatment since it is the net effect from the central cell and the long-range Coulomb interaction that impacts the electron scattering.

It is then clear that the sign of the central cell potential relative to that of Coulomb potential plays a large role in how charged defects scatter electrons. Here we show that the ionic radii of the dopant and host atoms can be used to indicate the sign of the central cell potential (Fig. 1a). The ionic radius here refers to the size of the ion without valence electrons. For example, a silicon atom can be considered as a silicon ion with +4 charge and four electrons, which we denote as $\text{Si}^{+4}[2s2p]$.

Similarly, a phosphorus atom can be denoted as $P^{+5}[2s3p]$. The reason we separate the valence electrons from the core is that valence electrons participate in chemical bonding and will be more strongly affected by the lattice structure, while the core electrons are more localized. When silicon is doped with phosphorus (P), phosphorus atom will lose one electron (which becomes free) and becomes positively charged. The remaining four electrons in the outer shell of P atom would participate in the chemical bonding, just as the valence electrons of Si would do, except that the core potential is different from that of Si. The defect potential (perturbed potential due to replacement of one Si by one P) thus results from the difference in the core potential (P^{+5} compared to Si^{+4}).

The first major difference between the core potential of P^{+5} and Si^{+4} is that their charges differ by one, which lead to a long-range Coulomb field when Si^{+4} is replaced by P^{+5} . In addition, as the electron approaches close to the center, it will experience difference in its interactions with the core electrons as the latter now are being held by a different atom. Such interactions involve both Coulomb interactions and exchange-correlation interactions, the latter of which are inherent to the many-body electron system and a cause of this is Pauli exclusion. On one hand, the Coulomb interactions between the nucleus and electrons will cause the core electrons to contract or expand depending on the atomic number of the dopant relative to the host, which would lead to variations in the core potential. In the case of P-doped Si, P^{+5} has a higher proton number than Si^{+4} , which leads to a slight contraction of the core electrons. This core electron contraction (and correspondingly higher core electron density) will tend to give a slight positive defect potential near the center, as observed in Supplementary Fig. 3 (inset). On the other hand, Pauli exclusion (or exchange-correlation interactions) forces the valence electrons to stay outside the core region (in the sense that the valence electron wavefunctions are orthogonal to core electrons). Therefore,

when the dopant ion has a smaller size than the host, propagating electron states (which are composed of mainly valence electrons) will have deeper penetration into the core, which is equivalent to creating an attractive short-range defect potential. For P-doped Si, because P^{+5} has a smaller ionic size than Si^{+4} , the defect potential becomes negative slightly away from the center (Supplementary Fig. 3, inset).

The above-mentioned effects lead to different trends of defect potential profile in the periodic table. The variation of core electron density mostly depends on the atomic number, while the Pauli exclusion effect is largely determined by the ionic size. While both will affect the defect potential, here we argue that the short-range core potential is better described by the ionic size difference between the dopant and host atom. This is because within and around the core region, the exchange-correlation interaction can be significant. When discussing the defect potential, especially the sign of its short-range part, we think Pauli exclusion is a more important governing factor, which is reasonably captured by the ionic radius picture to the first order. In general, when the dopant ion has a smaller size than the host, the defect potential tends to be dominantly negative (attractive potential to allow electrons to penetrate deeper into the core, Fig. 1b). Conversely, if a dopant ion has a larger size than the host, it will tend to repel electrons due to Pauli exclusion and tend to be dominantly positive (repulsive potential, Fig. 1c). We will later show that in most cases our simulation results agree better with this ionic size picture. We therefore will use the ionic radius as the major indicator to infer the sign of short-range defect potential.

Now we consider short-range and long-range parts together for the defect potential. For n-type dopant, the long-range Coulomb potential is attractive. A dopant with smaller ionic radius than the host would tend to create a short-range attractive potential, which adds to the Coulomb part and further increases EDI strength, leading to stronger electron-defect scatterings (electron

scattering scenario). In contrast, dopants with larger ionic radius will tend to create repulsive short-range potential, which counteracts the Coulomb potential. When the cancellation effect is maximal, the defects will appear to have negligible scatterings for electrons, and this realizes electron cloaking (electron cloaking scenario). As seen from the computed scattered electron wavefunction, the probability flux is distorted for the electron scattering scenario (Fig. 1b), while undistorted probability flux is recovered away from the defect for the electron cloaking scenario (Fig. 1c). In short, to achieve electron cloaking effect, dopants with larger ionic radius are favored for n-type materials. In addition to the electron cloaking, the ionic radius provides a useful guidance for understanding and selecting dopants in terms of the electron mobility. Effectively, the ionic radius can be seen as a scale bar for dopant selection: in n-type materials, dopants with larger ionic radius are more desired than those with smaller ionic radius because they are more repulsive to electrons and tend to counteract the Coulomb scatterings (Fig. 1a).

The above argument can also be applied to p-type materials, except that the Coulomb potential now becomes repulsive to electrons. As a result, the desired dopants that counteract the Coulomb scattering need to have attractive short-range potentials. The relation between ionic radius and short-range potentials remains the same. Therefore, dopants with smaller ionic radius are more desired for p-type materials (Supplementary Fig. 1).

To illustrate the consequence of different defect potentials on electron transport, we first study a model semiconductor as an example (see modeling details in Methods). Shown in Fig. 1d-e are scattered electron phase shifts computed for a n-type model semiconductor with different defect potentials. Phase shifts quantify how scattered electron waves differ from the incoming waves, and their magnitudes represent the strength of electron-defect scatterings²¹. In general, attractive potentials lead to negative phase shifts, while repulsive potentials lead to positive phase

shifts. When attractive short-range potential coexists with the attractive long-range Coulomb potential in n-type materials, the phase shifts add up in magnitude (Fig. 1d), leading to increased electron scattering rates and reduced mobility (Fig. 1f,h). In contrast, if the short-range potential is repulsive, it counteracts the attractive long-range Coulomb field and reduces the overall phase shifts (Fig. 1e), decreasing the total electron-defect scattering rates (Fig. 1g). When the defect scatterings become weaker than the intrinsic electron-phonon scatterings, the charge mobility becomes nearly immune to the presence of defects and high mobility can be achieved (Fig. 1i).

Electron scattering due to atomic distortions

Having discussed the effects of different dopants on mobility in the model semiconductor, now we turn to practical materials. Below we will first present simulation results and comparison with experiments corresponding to the electron scattering scenario, and then discuss possible evidences that demonstrate the electron cloaking effect. First, we note that mobility variations due to different dopants become significant when the electron-dopant interactions from the central cell potential are strong and comparable to those from the Coulomb field. To find out the parameter space where dopant selection is more critical, Figure 2a displays the ratio of the characteristic electron scattering rates due to central cell scattering and Coulomb scattering in the model semiconductor. In general, the central cell scattering becomes stronger for materials with larger dielectric constant and at higher carrier concentrations. This is because with larger dielectric constant the magnitude of Coulomb potential becomes relatively smaller and at higher carrier concentrations the Coulomb potential is weakened *via* screening. Conventional semiconductors usually have low to intermediate doping concentrations and small dielectric constant, which together indicate the dominance of the Coulomb potential in governing their electron scatterings and the relative unimportance of the dopant selection (Fig. 2a). In contrast, materials with a high

carrier density, *e.g.*, thermoelectrics¹⁶ and solid oxide conductors¹⁴, are likely to be more sensitive to the central cell potential due to their large carrier concentrations. Among them, many oxide and thermoelectric materials also possess a high dielectric constant associated with soft phonon modes. Thus, for these materials, the central cell effect could be significant and thereby potentially be harnessed to counteract the Coulomb scatterings and break the conventional limit in charge mobility at high carrier densities.

Chalcogenide compounds are a class of materials that have received wide attention due to their potential for optoelectronic, photovoltaic, and thermoelectric applications. In particular, the pursuit of high thermoelectric energy conversion efficiency has driven studies of heavily doped chalcogenide semiconductors¹⁶. Experiments have shown that different dopants with the same ionization charge can lead to drastically different mobility values, suggesting strong central cell effects¹³. The significance of the central cell effect can be described by the short-range electron-defect interaction (sEDI) matrix, $\langle \psi_{\mathbf{k}'} | \Delta \hat{V}_{cent} | \psi_{\mathbf{k}} \rangle$, where $\Delta \hat{V}_{cent}$ is the central cell part of the defect potential. For comparison, we computed the defect potential for both Si-doped GaAs, a conventional semiconductor, and Bi-doped PbTe, a representative chalcogenide material. Significant deviations from the Coulomb potential are seen at short range in both cases (Supplementary Fig. 2). We note that the planar averaged defect potentials generally vary around 0.1 – 1 eV at the short range, on the same order of magnitude with previous reports^{18,22}. Nonetheless, because of the strong covalent bond between Ga and As atoms and the larger spread of the charge density, the sEDI matrix in GaAs is generally small, as shown in Fig. 2b, which displays a three-dimensional contour colormap of the sEDI matrix in the real space, with the shape exhibiting the hybridized *s-p* orbital feature of the electron state. In comparison, PbTe consists of

resonantly bonded p orbitals, which are sensitive to the defect-induced local distortion that breaks the crystal symmetry. As a result, the sEDI matrix is significantly larger (Fig. 2c).

In order to evaluate the electron transport based on EDI, we need to account for the long-range Coulomb potential, which extends beyond the finite supercell calculations. The key observation that enabled our first-principles computation is that the long-range part of the defect potential can be well described by an analytic Coulomb potential profile (Supplementary Note and Supplementary Fig. 3, where the trend of short-range defect potentials of dopants in silicon are also found to agree with the ionic radius picture). Therefore, we are able to express the defect potential to far distances and to treat short-range and long-range potentials on an equal footing (see details in Methods). To validate our approach, Fig. 2d-e show comparisons between calculations of electron mobility and experimental results for GaAs and PbTe, respectively, at different carrier densities. For each material, two dopants at different atomic sites are considered. In GaAs, we observed that Si doping reduces the mobility slightly more than Te doping does (Fig. 2d). The dopant effects are generally small and the major scattering is due to the Coulomb potential, consistent with past electron transport studies on conventional semiconductors¹². On the other hand, the two dopants in PbTe have clearly different impacts. Bismuth (Bi) doping greatly reduces the electron mobility while iodine (I) doping is able to maintain a high mobility value (Fig. 2e). This large discrepancy occurs in part because electron states in the conduction band of PbTe are mostly formed by orbitals from Pb, and therefore I doping at the Te site only slightly disturbs the electron state while Bi doping at the Pb site overlaps with the electron states and has a large EDI matrix. In addition, because Bi has a smaller ionic radius than Pb²³, the short-range potential of Bi dopant is attractive and adds to the long-range Coulomb potential in n-type PbTe. Therefore, the large EDI from Bi doping contributes to strong electron-defect scattering, leading to decreased mobility as

seen in Fig. 2e. For both GaAs and PbTe, good agreement between simulation and experiment has been achieved.

Mobility variation with dopants has also been observed in other materials, e.g. in SrTiO₃, an oxide with perovskite structure. The computed defect potentials and corresponding electron transport properties for lanthanum (La) doping (on Sr site) and niobium (Nb) doping (on Ti site) are shown in Supplementary Fig. 4. In order to correctly describe the phonon modes in SrTiO₃ with perovskite phase, *ab initio* molecular dynamics was used to extract effective force constants at finite temperature^{24,25}. Neither dopant creates significant repulsive potentials that can counteract the Coulomb field. Instead, Nb dopant creates a strong attractive short-range potential that further enhances the electron-defect scatterings. We should mention that in both cases the dopant has an ionic size similar to the host, and therefore the sign of the defect potential is more sensitive to the actual electron density profile and needs to be determined by the calculation. In addition, La and Nb dopants are located at different atomic sites. In such case, the actual projection of valence electron wavefunctions on given atomic sites also influence the magnitude of the electron-defect scattering matrix (more discussions about this will be given later). For SrTiO₃, electrons near the conduction band edge mostly consist of *d* orbitals on Ti (Supplementary Fig. 4), which strongly overlap with the perturbation caused by Nb dopant. This further increases the electron-defect scatterings from Nb dopant. As a result, the computed mobility for Nb-doped SrTiO₃ is lower than La-doped one. We caution that the computed mobilities are not accurate due to the use of quasiparticle picture in Boltzmann transport theory²⁶, which ignores the polaron nature of charge transport in SrTiO₃. Nonetheless, the general trend of mobility with different dopants (La and Nb) agrees with experiments²⁷ (Supplementary Fig. 4e). The above examples confirmed the electron scattering scenario due to strong short-range electron-defect interactions and further demonstrate

the ability of our computational approach to distinguish the impact of different charged defects on electron transport.

Electron cloaking

We now discuss in what circumstances the short-range electron-defect interactions can instead enhance the mobility and possible evidences that demonstrate electron cloaking, using half-Heusler materials as examples. Half-Heuslers are a promising material family for spintronic and thermoelectric applications. Their large compositional variability has opened up a wide space for exploring different dopants to optimize their charge transport. We will use the ionic radius as a scale bar for the sign of the short-range defect potential, which determines whether it would be detrimental or beneficial to the charge mobility. First, we establish the correlation between ionic radius and the sEDI matrix. As explained above, a dopant with a large (small) ionic radius compared to that of the host atom tends to create a strong repulsive (attractive) short-range potential $\Delta\hat{V}_{cent}$, and thereby a large positive (negative) value in the sEDI matrix $\langle\psi_{\mathbf{k}'}|\Delta\hat{V}_{cent}|\psi_{\mathbf{k}}\rangle$. We therefore expect the sEDI matrix to correlate with the ionic radius difference between the dopant and host atoms. Figure 3a shows a comparison between computed sEDI matrices in half-Heusler materials (for electrons/holes at the band edge) and the ionic radius difference between the dopant and host atoms, which indeed demonstrates this correlation. Here, the ionic radii are taken from theoretical calculations based on Slater orbitals excluding the valence electrons²³. Supplementary Figure 5 further shows the defect potentials for a few dopant/host pairs in different materials, demonstrating that indeed whenever the dopant has a larger ionic size the defect potential becomes relatively more positive (this trend is also observed comparing dopants before and after the lanthanide contraction, e.g. Ta and Nb). Here, we mention that because the sEDI matrix is a spatial product of defect potential and the electronic wavefunction, sEDI values will

also depend on the projection of electronic wavefunctions on the dopant atom (see more discussions in Supplementary Note). For different materials, even when defect potentials are similar, the varying electronic wavefunctions and their spatial profiles would lead to variation in the sEDI values, and this is the reason why the correlation appears better within each material family. Nonetheless, based on Fig. 3a, we believe the variation in the ionic radius captures the major effect of short-range perturbation and provides a reasonable estimation for the sign and strength of the central cell potential. Based on this plot, dopants in the left region will lead to electron scatterings while dopants in the right region would favor electron cloaking (for n-type materials). Examples of defect potentials from dopants shown in Fig. 3a leading to electron-scattering or -cloaking scenarios can be found in Supplementary Fig. 6.

Classification of dopants based on their ionic radius allows us to further understand the electron transport behavior in heavily doped semiconductors, and is a potential guide in the selection of dopants to enhance mobility. This can be seen in the experimental results of two example compounds – ZrNiSn and NbFeSb, which are well known for their thermoelectric performance among n-type and p-type semiconductors, respectively. For n-type ZrNiSn, the typical dopants (V, Nb, or Ta) each have a smaller ionic radius than the host atom (Zr, Fig. 3a), resulting in an attractive central cell potential. This attractive potential adds to the attractive Coulomb potential (for n-type material) and increases the defect scattering. As a result, the compiled experimental mobility data mostly show a monotonic decreasing trend with increasing carrier concentration (Fig. 3b). The non-monotonic trend in one data has been attributed to other extrinsic defect scatterings and is not directly related to dopants²⁸. In Fig. 3d, we further show highest mobility values for V, Nb and Ta doping from past work with respect to the ionic radius difference between dopant and host ($r_{dopant} - r_{host}$, based on Fig. 3a). A general trend of

increasing mobility with increasing dopant ionic radius is observed. This is consistent with our defect scattering picture (Fig. 1a), because for n-type materials a dopant with larger ionic radius is less attractive to electrons and therefore contributes less to total electron-defect scatterings.

On the other hand, for p-type NbFeSb, while the typical dopants (Ti, Hf) each still have a smaller ionic radius than the host atom (Nb), the Coulomb potential is now repulsive (for p-type material), so the two potentials counteract each other. In this case, there will be a carrier concentration range in which the dopant scattering is weak and the mobility approaches the intrinsic electron-phonon scattering limit, manifesting as a peak with increasing carrier concentration (Fig. 1i). Such peaks are indeed observed in the compiled experimental mobility data for p-type NbFeSb materials (Fig. 3c), suggesting that a partial electron cloaking effect is at play. However, we acknowledge that other extrinsic effects, such as existence of compensated charged defects, may also lead to non-monotonic mobility variation. Another mechanism that is responsible for mobility peaks is the screening of polar optical phonon scattering, an intrinsic scattering mechanism²⁹. Still, if the scattering from the dopant is strong, the mobility would instead be limited by electron-defect interactions and a monotonically decreasing trend would be expected. The general observation of mobility peaks in NbFeSb suggests electron cloaking effect likely exists. Another evidence supporting electron cloaking effect is shown in Fig. 3e, which plots the highest mobility values from past work for different dopants with respect to the ionic radius difference. In contrast to the case with ZrNiSn, dopant with the largest ionic radius difference from the host (Ti dopant on Nb) has highest mobility, consistent throughout many studies. This contradicts simple defect scattering picture, which would suggest such dopants should create strongest short-range electron-dopant scattering and decrease the mobility. However, because small ionic radius dopants actually create attractive short-range potential that counteracts the

repulsive Coulomb field in p-type materials, the overall electron-defect scattering should decrease (and correspondingly the mobility increases) as the dopant's ionic radius becomes smaller, which is the trend observed in Fig. 3e. The electron dopant interaction picture based on the ionic radius is consistent with both n-type and p-type materials (Fig. 3c,e).

In order to quantify the extent to which electron cloaking can benefit the electron transport, and in particular the thermoelectric performance in the case of half-Heusler materials, we computed the electron transport properties in p-type NbFeSb. Ti-doped NbFeSb was recently reported to possess a high power factor at room temperature³⁰. The analysis above suggests that Ti doping in p-type NbFeSb facilitates electron cloaking (Fig. 3c,e). Our calculations show that despite the large doping concentrations, the electron scattering rates are indeed close to the intrinsic limit (determined by electron-phonon scattering) due to the counteracting Coulomb potential and the short-range defect potential of Ti (Supplementary Fig. 7). This partial electron cloaking effect leads to higher electrical conductivity and a larger thermoelectric power factor in the range from 300 K to 1000 K, bringing the simulation results into better agreement with the experiment (Fig. 3f-g). Moreover, the relative magnitudes of the optimal power factors in NbFeSb with the various dopants experimentally investigated thus far³¹ are also consistent with our calculations (Fig. 3h), and the trend with the ionic radius agrees with our electron dopant interaction picture (compare Fig. 3h with Fig. 3e). The higher power factors achieved with Ti and Hf dopants can be understood by their defect potentials: their short-range potentials counterbalance the Coulomb potential and create a partial cloaking effect that enhances the charge mobility (Supplementary Fig. 7). In a similar compound (Ti-doped TaFeSb) we also observed a large power factor enhancement due to electron cloaking (Supplementary Fig. 8). While these enhancements may not seem large, the thermoelectric figure of merit zT is directly proportional to the power factor. In this regard, rational

dopant selection that improves the charge mobility and power factor will be beneficial for the thermoelectric efficiency, whose improvement has been a challenging task. Besides, we note that such enhancement due to electron cloaking is expected to become stronger at lower temperatures, due to the increasing importance of defect scatterings compared to intrinsic electron-phonon scatterings as the temperature decreases. Our computation shows that Ti dopant in NbFeSb and TaFeSb can potentially lead to significant enhancement of the power factor by as much as 80% at 150 K compared to the conventional Coulomb-limited case due to the cloaking effect of the dopant (Supplementary Fig. 9). This thus also provides an opportunity to optimize thermoelectric materials for cooling and refrigeration applications³² through electron cloaking.

In summary, we have demonstrated that the chemical details of certain dopants can be harnessed to counteract the strong electron scatterings resulting from their long-range Coulomb field. Consequently, in contrast to the conventional belief that charge mobility is always limited by extrinsic Coulomb scatterings, we have shown how high intrinsic mobility can be achieved by rationally selecting dopants based on their ionic radius. While our study focuses on point defects, the first-principles computational approach can be applied to other short-range interactions such as those due to defect clusters, and even dislocations or grain boundaries, when the long-range Coulomb interactions and short-range perturbations are comparable in strength. Our results provide guidelines for dopant selection, which thus far has been mostly based on trial-and-error. The insights provided here on the impact of often-neglected atomic details of defects on charge transport in heavily doped materials will stimulate the search for high-efficiency thermoelectric materials, as well as the development of high-mobility materials for microelectronic and optoelectronic applications.

Methods

Model study of electron-defect scattering

To illustrate the general impact of central cell potential on charge transport, we have used a model semiconductor with a parametrized isotropic band structure and scattering information corresponding to that of silicon, and we represent the defect potential by a simplified profile as shown in Fig. 1a. The charge mobility is given by

$$\mu_e = \left[\frac{N_v e}{3} \int v^2 \tau \left(-\frac{\partial f^0}{\partial E} \right) D(E) dE \right] / n \quad (1)$$

where the integration spans over electron states close to the Fermi level, N_v is the band degeneracy, e is the electronic charge, v is the electron group velocity and is related to electron energy *via* the conductivity effective mass $m_{eff,c}$, as $v^2 = 2E/m_{eff,c}$, τ is the electron relaxation time, $f^0 = 1/(1 + \exp(\frac{E-\mu}{k_B T}))$ is the Fermi-Dirac distribution function with μ being the Fermi level, E is the electron energy, $D(E)$ is the electronic density of states and is related to the density-of-states effective mass $m_{eff,DOS}$ *via* $D = (\frac{2m_{eff,DOS}}{\hbar^2})^{3/2} \frac{\sqrt{E}}{2\pi^2}$ with \hbar being the reduced Planck constant, and n is the carrier concentration.

The electron relaxation time τ , the inverse of the scattering rate, is determined *via* Matthiessen's rule considering both intrinsic electron-phonon interactions and extrinsic electron-defect interactions: $1/\tau = 1/\tau_{e-ph} + 1/\tau_{e-d}$. The electron-phonon interactions consider both acoustic phonon and optical phonon scatterings *via* corresponding deformation potentials, and the full parametrization is given in Supplementary Note. The electron-defect scattering rate $1/\tau_{e-d}$ is calculated based on the partial wave analysis which evaluates the scattering of electrons by a spherically symmetric potential

$$\frac{1}{\tau_{e-d}} = N_d \frac{4\pi}{\hbar^2} \frac{1}{m_{eff,DOS} \sqrt{2m_{eff,DOS} E}} \sum_{l=0}^{\infty} (l+1) \sin^2(\delta_l - \delta_{l+1}) \quad (2)$$

where N_d is the volume density of the defect and δ_l is the phase shift of the electron wave with quantum number l . The defect potential (in units of energy) has been taken to have the form $\Delta\hat{V} = \begin{cases} V_0 & r \leq r_0 \\ -\frac{e^2}{4\pi\epsilon\epsilon_0 r} & r > r_0 \end{cases}$, with r_0 characterizing the range of the short-range potential, and where V_0 is the short-range potential energy and ϵ_0 is the vacuum permittivity. For the results in Fig. 1, we have assumed $V_0 = 9.7$ eV and $r_0 = 1.6$ Å. The scattering rates and mobility in Fig. 1d-g are obtained assuming $\epsilon = 11.7$ (dielectric constant corresponding to silicon) while the dielectric constant is varied in Fig. 2a.

Details of the calculation of the partial wave phase shift are provided in the Supplementary Note. In brief, because the defect potential is spherically symmetric, the orbital angular momentum operator becomes a constant of motion for electrons, and the electron wavefunctions can be represented by an additional quantum number l in addition to its energy E , called partial waves.

Each partial wave is scattered by the potential and the scattered wave acquires a phase shift δ_l compared to the case in which no defect is present. Intuitively, the phase shift represents how strongly the defect potential attracts or repels the electrons, both of which will lead to a large phase shift and thus strong scattering. The total scattering rates are obtained *via* the above formula by considering all values of l . The partial wave analysis considers multiple scatterings between the electron wave and the defect, and is thus exact under the assumption that the electron wave is a plane wave²⁰. Although the electron wave actually has a more complex profile modulated by the periodic potential in the crystal, the above results provide an estimation of the effect of the central cell potential on charge transport, particularly in comparison to the conventional Coulomb scatterings.

First-principles calculation of defect potential

The defect potential, $\Delta\hat{V}$, is defined as the difference in the total electronic potential between the system containing one defect and the pristine bulk material, $\Delta\hat{V} = V_d - V_0$, where the calculation is performed for a cubic supercell containing 96 atoms for half-Heusler materials. In the calculation of the structure with one defect, a net charge is given corresponding to the charged state of the dopant. In our study, the dopants were chosen from a column in the periodic table adjacent to that of the atom being substituted, and thus the net charge is assumed to be +1 for n-type and -1 for p-type. The extracted defect potential contains both the long-range Coulomb potential and the central cell part which deviates from the Coulomb potential profile. This defect potential cannot be directly used to compute the EDI matrix $\langle\psi_{k'}|\Delta\hat{V}|\psi_k\rangle$ since the finite size of the supercell does not correctly capture the large span of the long-range Coulomb field. However, because the long-range tail of the defect potential agrees well with the analytic Coulomb potential profile (Fig. 1b-c), we can first subtract the long-range part from the defect potential, leaving only a short-range component $\Delta\hat{V}_{sr}$. In figures where we show the defect potential extracted from first principles simulation, we have also plotted the Coulomb potential to show that first principles defect potential indeed recovers the correct asymptotic trend given by the Coulomb field when one moves away from the defect, as has been shown in previous work²². When evaluating the EDI matrix later, we add the long-range Coulomb potential back to the defect potential, $\Delta\hat{V} = \Delta\hat{V}_{sr} + \Delta\hat{V}_{lr}$. Because the second contribution to EDI due to the long-range part can be computed using an analytic expression for the Coulomb potential extending to a longer distance (to be detailed below), we circumvent the difficulty of the finite size of the supercell and are able to treat both long-range and short-range potentials on an equal footing. This workflow is similar to the recent development in incorporating the long-range polar optical phonon scattering into the Wannier interpolation method for electron-phonon interaction calculations^{33,34}.

When subtracting the long-range part from the defect potential, we compute the long-range potential according to the following formula³⁵ based on the Ewald summation, which represents the Coulomb potential at location \mathbf{r} due to an infinite periodic array of charge Ze at locations \mathbf{R}_i (given by the supercell size).

$$\Delta\hat{V}_{lr} = -\sum_i \frac{ze^2}{\sqrt{|\bar{\epsilon}|}} \frac{\text{erfc}\left(\gamma\sqrt{(\mathbf{R}_i-\mathbf{r})\cdot\bar{\epsilon}^{-1}\cdot(\mathbf{R}_i-\mathbf{r})}\right)}{\sqrt{(\mathbf{R}_i-\mathbf{r})\cdot\bar{\epsilon}^{-1}\cdot(\mathbf{R}_i-\mathbf{r})}} - \sum_{\mathbf{G}_i \neq 0} \frac{4\pi ze^2}{\Omega} \frac{\exp\left(-\mathbf{G}_i\cdot\bar{\epsilon}\cdot\frac{\mathbf{G}_i}{4\gamma^2}\right)}{\mathbf{G}_i\cdot\bar{\epsilon}\cdot\mathbf{G}_i} \exp(i\mathbf{G}_i\cdot\mathbf{r}) + \frac{\pi ze^2}{\Omega\gamma^2} \quad (3)$$

Here $\bar{\epsilon}$ is the dielectric tensor computed from first principles, Ω is the supercell volume, and γ is a convergence parameter for the Ewald summation. In Supplementary Note, we show that the remaining potential after the subtraction is indeed short-range. Because a short-range potential should approach zero at distances away from the defect, we further align the potential at far distances to zero.

Electron transport calculation

The first-principles electron transport properties for half-Heusler materials are calculated by summing over all electron states according to the Boltzmann transport theory³⁶. For example, the electrical conductivity is given by

$$\sigma = \frac{e^2}{3\Omega_0 N_k} \sum_{\mathbf{k}\alpha} v_{\mathbf{k}\alpha}^2 \tau_{\mathbf{k}\alpha} \left(-\frac{\partial f_{\mathbf{k}\alpha}^0}{\partial E} \right) \quad (4)$$

in which we have explicitly written out the summation of the discrete mesh points \mathbf{k} in the Brillouin zone, and where Ω_0 is the unit cell volume and α is the band index. Isotropic materials are assumed and the factor 1/3 appears because the conductivity is the same in all three directions. The equilibrium properties of electrons are calculated from first principles using the QUANTUM ESPRESSO software package³⁷. We use the generalized gradient approximation (GGA) of Perdew, Burke and Ernzerhof³⁸ with the Troullier-Martins-type norm-conserving semilocal pseudopotential (corresponding to pbe-mt.UPF in the QUANTUM ESPRESSO pseudopotential library). A cutoff energy of 120 Ryd and a $6 \times 6 \times 6$ \mathbf{k} -mesh are used to determine the equilibrium lattice constant. The equilibrium properties of phonons and the electron-phonon interaction matrices are calculated *via* density functional perturbation theory³⁹ for a $6 \times 6 \times 6$ \mathbf{q} -mesh (with a $6 \times 6 \times 6$ \mathbf{k} -mesh for the electron-phonon interaction matrix). We then use the EPW software package⁴⁰ to interpolate the electronic information and the phonon information, as well as the electron-phonon coupling matrices to a fine mesh *via* the Wannier interpolation method⁴¹. In determining the Fermi level μ , we assumed that the dopants are fully ionized and therefore the Fermi level is such that the doping concentration N_d equals the total carrier concentration, which is given by $n = \frac{1}{\Omega_0 N_k} \sum_{\mathbf{k}\alpha} f_{\mathbf{k}\alpha}^0$. The electron relaxation time τ is determined *via* Matthiessen's rule considering both intrinsic electron-phonon interactions and extrinsic electron-defect interactions: $1/\tau = 1/\tau_{e-ph} + 1/\tau_{e-d}$. The electron-phonon interaction matrices are first calculated within density functional perturbation theory and then interpolated *via* the Wannier interpolation scheme to the fine mesh⁴⁰. This includes electron scatterings by polar optical phonons^{33,34}, as well as taking the carrier screening effect into account. More details on this can be found in our previous work³⁶ and in Supplementary Note.

The calculation of electron-defect scattering rates considering the full defect potential is the key step in our study, and is given by the following formula under the momentum relaxation approximation

$$\frac{1}{\tau_k^{e-d}} = N_d \Omega_0 \frac{2\pi}{\hbar} \frac{1}{N_k} \sum_{\mathbf{k}'} \left(1 - \frac{\mathbf{v}_k \cdot \mathbf{v}_{\mathbf{k}'}}{|\mathbf{v}_k| |\mathbf{v}_{\mathbf{k}'}} \right) |g_{e-d}(\mathbf{k}, \mathbf{k}')|^2 \delta(E_{\mathbf{k}} - E_{\mathbf{k}'}) \quad (5)$$

where N_d is the volume density of dopants, N_k is the number of \mathbf{k} points, the factor $1 - \frac{\mathbf{v}_k \cdot \mathbf{v}_{\mathbf{k}'}}{|\mathbf{v}_k| |\mathbf{v}_{\mathbf{k}'}}$ takes into account the fact that scatterings between electrons with similar velocity directions do not contribute much to momentum loss and thus less to electrical resistance, and $\delta(E_{\mathbf{k}} - E_{\mathbf{k}'})$ indicates that the defect scattering is an elastic process. $g_{e-d}(\mathbf{k}, \mathbf{k}') = \langle \psi_{\mathbf{k}'} | \Delta \hat{V} | \psi_{\mathbf{k}} \rangle$ is the EDI matrix. As explained above, the defect potential $\Delta \hat{V}$ contains both long-range and short-range parts, leading to two contributions to EDI. To compute these contributions, the EDI matrix is first rewritten as follows¹⁹

$$\langle \psi_{\mathbf{k}} | \Delta \hat{V} | \psi_{\mathbf{k}'} \rangle = \int d^3r u_{\mathbf{k}'}^* e^{-i\mathbf{k}' \cdot \mathbf{r}} \Delta \hat{V} u_{\mathbf{k}} e^{i\mathbf{k} \cdot \mathbf{r}} = \sum_{\mathbf{G}} \Delta V(\mathbf{k}' - \mathbf{k} + \mathbf{G}) \langle u_{\mathbf{k}'} | e^{i\mathbf{G} \cdot \mathbf{r}} | u_{\mathbf{k}} \rangle \quad (6)$$

where the Fourier transform of the defect potential is defined as $\Delta V(\mathbf{q}) = \frac{1}{\Omega_0} \int d^3r \Delta \hat{V}(\mathbf{r}) e^{-i\mathbf{q} \cdot \mathbf{r}}$, with Ω_0 being the unit cell volume and the integration spanning the entire space. This form separates the defect potential from the wave functions, and the factor containing wavefunctions can be computed readily once the periodic components $u_{\mathbf{k}}$ of the wavefunctions are known¹⁹: $\langle u_{\mathbf{k}'} | e^{i\mathbf{G} \cdot \mathbf{r}} | u_{\mathbf{k}} \rangle = \int d^3r u_{\mathbf{k}'}^* e^{-i\mathbf{G} \cdot \mathbf{r}} u_{\mathbf{k}}$, where the integration spans over the unit cell. To evaluate the EDI matrix, we then compute the Fourier component of the defect potential $\Delta V(\mathbf{q})$, which again contains both long-range and short-range parts. The short-range part can be calculated readily within the supercell based on $\Delta V_{sr}(\mathbf{q}) = \frac{1}{\Omega_0} \int d^3r \Delta \hat{V}_{sr}(\mathbf{r}) e^{-i\mathbf{q} \cdot \mathbf{r}}$ since the potential has negligible contributions at far distances. The long-range part can be obtained by performing the integration analytically to infinity, yielding $\Delta V_{lr}(\mathbf{q}) = -\frac{Ze^2}{\Omega_0 \epsilon \epsilon_0} \frac{1}{|\mathbf{q}|^2 + (1/L_D)^2}$. For this expression, we have assumed the Coulomb potential energy is given by $\Delta \hat{V}_{lr}(\mathbf{r}) = -\frac{Ze^2}{4\pi \epsilon \epsilon_0} \frac{\exp(-r/L_D)}{r}$, where the factor $\exp(-r/L_D)$ considers the carrier screening at high carrier concentrations with the Debye screening length L_D given by

$$L_D = \left(\frac{e^2}{\epsilon \epsilon_0} \int \left(-\frac{\partial f}{\partial E} \right) D(E) dE \right)^{-1/2} \quad (7)$$

Adding both long-range and short-range components as $\Delta V(\mathbf{q}) = \Delta V_{sr}(\mathbf{q}) + \Delta V_{lr}(\mathbf{q})$ allows us to evaluate the EDI matrix completely. The electron-defect scattering rates thus obtained are then combined with electron-phonon scatterings to give the total electron scattering rates, which are used to evaluate the electron transport properties, as in Eq. (4).

Data Availability

The data that support the findings of this study are available from the corresponding authors on reasonable request.

Code Availability

The code for computing electron scattering rates through first principles electron transport calculation is a modified version of the EPW code⁴⁰, originally released within the QUANTUM ESPRESSO package³⁷. Our modified EPW code is available at <http://doi.org/10.24435/materialscloud:5a-7s>.

Acknowledgement

We thank T.-H. Liu, M. Li, and Q. Zhang for helpful discussions on the first-principles calculations and electron-defect interactions. The work performed at MIT is supported by the DARPA MATRIX program under Grant No. HR0011-16-2-0041.

Author Contributions

J.Z. and G.C. conceived the project. J.Z. performed the theoretical analysis and first-principles computation. J.Z., H.Z., Q.S., Z.R., and G.C. contributed to the data analysis. J.Z. and G.C. wrote the manuscript. All authors commented on, discussed, and edited the manuscript.

Competing Financial Interests

The authors declare no competing financial interests.

Reference

1. Sze, S. M. *Physics of Semiconductor Devices*. (John Wiley & Sons, 1981).
2. Minnich, A. J., Dresselhaus, M. S., Ren, Z. F. & Chen, G. Bulk nanostructured thermoelectric materials: current research and future prospects. *Energy Environ. Sci.* **2**, 466–479 (2009).
3. Chattopadhyay, D. & Queisser, H. J. Electron scattering by ionized impurities in semiconductors. *Rev. Mod. Phys.* **53**, 745–768 (1981).
4. Freysoldt, C. *et al.* First-principles calculations for point defects in solids. *Rev. Mod. Phys.* **86**, 253–305 (2014).
5. Brooks, H. Theory of the electrical properties of germanium and silicon. in *Advances in Electronics and Electron Physics* (ed. Marton, L.) vol. 7 85–182 (Academic Press, 1955).
6. Herring, C. Transport properties of a many-valley semiconductor. *The Bell System Technical Journal* **34**, 237–290 (1955).
7. Fischetti, M. V. Effect of the electron-plasmon interaction on the electron mobility in silicon. *Phys. Rev. B* **44**, 5527–5534 (1991).
8. Sanborn, B. A., Allen, P. B. & Mahan, G. D. Theory of screening and electron mobility: Application to n-type silicon. *Phys. Rev. B* **46**, 15123–15134 (1992).
9. Ralph, H. I., Simpson, G. & Elliott, R. J. Central-cell corrections to the theory of ionized-impurity scattering of electrons in silicon. *Phys. Rev. B* **11**, 2948–2956 (1975).
10. El-Ghanem, H. M. A. & Ridley, B. K. Impurity scattering of electrons in non-degenerate semiconductors. *J. Phys. C: Solid State Phys.* **13**, 2041 (1980).
11. Sankey, O. F., Dow, J. D. & Hess, K. Theory of resonant scattering in semiconductors due to impurity central-cell potentials. *Appl. Phys. Lett.* **41**, 664–666 (1982).
12. Szymyd, D. M., Hanna, M. C. & Majerfeld, A. Heavily doped GaAs:Se. II. Electron mobility. *Journal of Applied Physics* **68**, 2376–2381 (1990).
13. Wang, H., Cao, X., Takagiwa, Y. & Snyder, G. J. Higher mobility in bulk semiconductors by separating the dopants from the charge-conducting band – a case study of thermoelectric PbSe. *Mater. Horiz.* **2**, 323–329 (2015).
14. Reagor, D. W. & Butko, V. Y. Highly conductive nanolayers on strontium titanate produced by preferential ion-beam etching. *Nature Materials* **4**, 593–596 (2005).
15. Hitosugi, T., Yamada, N., Nakao, S., Hirose, Y. & Hasegawa, T. Properties of TiO₂-based transparent conducting oxides. *physica status solidi (a)* **207**, 1529–1537 (2010).
16. Liu, W. *et al.* New trends, strategies and opportunities in thermoelectric materials: A perspective. *Materials Today Physics* **1**, 50–60 (2017).
17. Lany, S. & Zunger, A. Assessment of correction methods for the band-gap problem and for finite-size effects in supercell defect calculations: Case studies for ZnO and GaAs. *Phys. Rev. B* **78**, 235104 (2008).
18. Freysoldt, C., Neugebauer, J. & Van de Walle, C. G. Fully ab initio finite-size corrections for charged-defect supercell calculations. *Phys. Rev. Lett.* **102**, 016402 (2009).
19. Lu, I.-T., Zhou, J.-J. & Bernardi, M. Efficient ab initio calculations of electron-defect scattering and defect-limited carrier mobility. *Phys. Rev. Materials* **3**, 033804 (2019).

20. Liao, B., Zebarjadi, M., Esfarjani, K. & Chen, G. Cloaking core-shell nanoparticles from conducting electrons in solids. *Phys. Rev. Lett.* **109**, 126806 (2012).
21. Cohen-Tannoudji, C., Diu, B. & Laloe, F. *Quantum Mechanics*. (Wiley, 1992).
22. Rurali, R., Markussen, T., Suñé, J., Brandbyge, M. & Jauho, A.-P. Modeling transport in ultrathin Si nanowires: Charged versus neutral impurities. *Nano Lett.* **8**, 2825–2828 (2008).
23. Ghosh, D. C. & Biswas, R. Theoretical calculation of absolute radii of atoms and ions. Part 2. The ionic radii. *International Journal of Molecular Sciences* **4**, 379–407 (2003).
24. Hellman, O., Steneteg, P., Abrikosov, I. A. & Simak, S. I. Temperature dependent effective potential method for accurate free energy calculations of solids. *Phys. Rev. B* **87**, 104111 (2013).
25. Tadano, T., Gohda, Y. & Tsuneyuki, S. Anharmonic force constants extracted from first-principles molecular dynamics: applications to heat transfer simulations. *J. Phys.: Condens. Matter* **26**, 225402 (2014).
26. Zhou, J.-J. & Bernardi, M. Predicting charge transport in the presence of polarons: The beyond-quasiparticle regime in SrTiO₃. *Phys. Rev. Research* **1**, 033138 (2019).
27. Han, W. *et al.* Spin injection and detection in lanthanum- and niobium-doped SrTiO₃ using the Hanle technique. *Nat. Comm.* **4**, 2134 (2013).
28. Xie, H. *et al.* The intrinsic disorder related alloy scattering in ZrNiSn half-Heusler thermoelectric materials. *Scientific Reports* **4**, 6888 (2014).
29. Ren, Q. *et al.* Establishing the carrier scattering phase diagram for ZrNiSn-based half-Heusler thermoelectric materials. *Nature Communications* **11**, 3142 (2020).
30. He, R. *et al.* Achieving high power factor and output power density in p-type half-Heuslers Nb_{1-x}Ti_xFeSb. *PNAS* **113**, 13576–13581 (2016).
31. Ren, W. *et al.* Ultrahigh power factor in thermoelectric system Nb_{0.95}M_{0.05}FeSb (M = Hf, Zr, and Ti). *Advanced Science* **5**, 1800278 (2018).
32. Mao, J. *et al.* High thermoelectric cooling performance of n-type Mg₃Bi₂-based materials. *Science* **365**, 495–498 (2019).
33. Sjakste, J., Vast, N., Calandra, M. & Mauri, F. Wannier interpolation of the electron-phonon matrix elements in polar semiconductors: Polar-optical coupling in GaAs. *Phys. Rev. B* **92**, 054307 (2015).
34. Verdi, C. & Giustino, F. Fröhlich Electron-phonon vertex from first principles. *Phys. Rev. Lett.* **115**, 176401 (2015).
35. Kumagai, Y. & Oba, F. Electrostatics-based finite-size corrections for first-principles point defect calculations. *Phys. Rev. B* **89**, 195205 (2014).
36. Zhou, J., Liao, B. & Chen, G. First-principles calculations of thermal, electrical, and thermoelectric transport properties of semiconductors. *Semicond. Sci. Technol.* **31**, 043001 (2016).
37. Giannozzi, P. *et al.* QUANTUM ESPRESSO: a modular and open-source software project for quantum simulations of materials. *J. Phys.: Condens. Matter* **21**, 395502 (2009).
38. Perdew, J. P., Burke, K. & Ernzerhof, M. Generalized gradient approximation made simple. *Phys. Rev. Lett.* **77**, 3865–3868 (1996).

39. Baroni, S., de Gironcoli, S., Dal Corso, A. & Giannozzi, P. Phonons and related crystal properties from density-functional perturbation theory. *Rev. Mod. Phys.* **73**, 515–562 (2001).
40. Giustino, F., Cohen, M. & Louie, S. Electron-phonon interaction using Wannier functions. *Phys. Rev. B* **76**, 165108 (2007).
41. Marzari, N., Mostofi, A. A., Yates, J. R., Souza, I. & Vanderbilt, D. Maximally localized Wannier functions: Theory and applications. *Rev. Mod. Phys.* **84**, 1419–1475 (2012).
42. LaLonde, A. D., Pei, Y. & Snyder, G. J. Reevaluation of PbTe_{1-x}I_x as high performance n-type thermoelectric material. *Energy Environ. Sci.* **4**, 2090–2096 (2011).
43. Gelbstein, Y., Dashevsky, Z. & Dariel, M. P. Synthesis of n-type PbTe by powder metallurgy. in *Proceedings ICT2001 20th International Conference on Thermoelectrics* 143–149 (2001).
44. Rogacheva, E. I., Lyubchenko, S. G., Vodoretz, O., Kuzmenko, A. M. & Dresselhaus, M. Thermoelectric properties of PbTe crystals and thin films. in *2006 25th International Conference on Thermoelectrics* 656–661 (2006).
45. Chauhan, N. S. *et al.* Vanadium-doping-induced resonant energy levels for the enhancement of thermoelectric performance in Hf-free ZrNiSn half-Heusler alloys. *ACS Appl. Energy Mater.* **1**, 757–764 (2018).
46. Muta, H., Kanemitsu, T., Kurosaki, K. & Yamanaka, S. High-temperature thermoelectric properties of Nb-doped MNiSn (M=Ti, Zr) half-Heusler compound. *Journal of Alloys and Compounds* **469**, 50–55 (2009).
47. Zhang, H. *et al.* Thermoelectric properties of n-type half-Heusler compounds (Hf_{0.25}Zr_{0.75})_{1-x}Nb_xNiSn. *Acta Materialia* **113**, 41–47 (2016).
48. Yang, X. *et al.* Enhanced thermoelectric performance of Zr_{1-x}Ta_xNiSn half-Heusler alloys by diagonal-rule doping. *ACS Appl. Mater. Interfaces* **12**, 3773–3783 (2020).
49. Zhao, D., Zuo, M., Wang, Z., Teng, X. & Geng, H. Synthesis and thermoelectric properties of tantalum-doped ZrNiSn half-Heusler alloys. *Funct. Mater. Lett.* **07**, 1450032 (2014).
50. Fu, C., Zhu, T., Liu, Y., Xie, H. & Zhao, X. Band engineering of high performance p-type FeNbSb based half-Heusler thermoelectric materials for figure of merit $zT > 1$. *Energy Environ. Sci.* **8**, 216–220 (2015).
51. Fu, C. *et al.* Realizing high figure of merit in heavy-band p-type half-Heusler thermoelectric materials. *Nat. Comm.* **6**, 8144 (2015).
52. Tavassoli, A. *et al.* On the Half-Heusler compounds Nb_{1-x}{Ti,Zr,Hf}_xFeSb: Phase relations, thermoelectric properties at low and high temperature, and mechanical properties. *Acta Materialia* **135**, 263–276 (2017).

Figure Legends

Figure 1. Impact of charged defects on electron transport. **a**, Illustration of a propagating electron wave scattered by the perturbed electronic potential due to the presence of charged defects. The example of an n-type dopant is depicted. The defect potential in general contains two parts, a long-range part due to the Coulomb field of the charge which is attractive to electrons for a n-type dopant, and a short-range part that depends on the bonding environment of the defect, known as the central cell potential. The ionic radius can be used as a good indicator for the sign of the central cell potential. Depending on the ionic radius of the dopant atom, one of two general scenarios can play out. If the dopant atom has a smaller ionic radius than the host atom, the perturbation tends to create an additional attractive force for electrons (electron-scattering scenario); if the dopant atom has a larger ionic radius, it tends to create a repulsive force which then opposes the long-range Coulomb potential (electron-cloaking scenario). The above discussions pertain to the case of n-type dopant. **b-c**, Defect potentials and scattered electron wavefunctions corresponding to two different scenarios (**b**: electron scattering, **c**: electron cloaking). Here the central cell potential is represented by a simplified rectangular profile. The streamlines show the probability flux and the colors indicate the real part of the wavefunctions. In the case of electron cloaking (**c**), unperturbed streamlines are recovered away from the defect (located at the center). The domain size is $158.8 \text{ \AA} \times 158.8 \text{ \AA}$ and the electron energy is 0.1 eV . **d-e**, Phase shifts of scattered wave functions with angular momentum quantum number of $l = 1$ for two scenarios (**d**: electron scattering, **e**: electron cloaking). **f-g**, Modeled intrinsic electron scattering rates due to electron-phonon interactions, in comparison to those considering electron-defect scatterings (**f**: electron scattering, **g**: electron cloaking). The calculation assumes a parabolic band to illustrate the general impact of defect scatterings and uses partial wave analysis to calculate the scattering rates due to defects (see details in Methods). In (**g**), because the central cell potential opposes the Coulomb potential, their effects on electron scatterings are partially canceled. **h-i**, Electron mobility with respect to the carrier concentration (**h**: electron scattering, **i**: electron cloaking). While mobility at high carrier concentrations is traditionally believed to be limited by Coulomb scattering, a strong opposing central cell potential can be harnessed to break this limit, leading to high mobility limited only by the intrinsic electron-phonon interactions, as shown in (**i**).

Figure 2. Electron-defect interaction. **a**, Ratio between characteristic electron scattering rates due to central cell scattering and those due to Coulomb scattering, $\eta = \gamma_{cent}/\gamma_{Coul}$, shown as a color map (with red indicating high η , blue indicating low η , and white indicating unity, as shown in the scale at right), with respect to the dielectric constant and carrier concentration. The characteristic electron scattering rate γ is defined based on the mobility μ as $\gamma = e/(m^*\mu)$. Studied materials in this work are also labeled in the plot. Both PbTe and SrTiO $_3$ have large dielectric constants and the arrows indicate their actual locations lie outside the given dielectric constant range. Regions with $\eta \approx 1$ represent cases in which the central cell potential has a comparable impact on charge transport to that of the Coulomb potential, and thus could be harnessed to counteract the Coulomb scatterings. The magnitude of the central cell potential is taken to be 6 eV (with a width of 6 Bohr radius) in this simulation. **b-c**, Contour plots of the electron-defect interaction matrix $\langle \psi_{\mathbf{k}} | \Delta \hat{V} | \psi_{\mathbf{k}} \rangle$ for **(b)** Si-doped GaAs and **(c)** Bi-doped PbTe, respectively, where \mathbf{k} is taken to be at the conduction band edge state (at the Γ point for GaAs and at the L point for PbTe). Significant electron-defect interaction is seen for PbTe. The unit of the coordinates is \AA and the center is at the defect location. **d-e**, Computed electron mobility as a function of the carrier density for **(d)** n-type GaAs with two different dopants (Ga:Si, As:Te) and **(e)** n-type PbTe with two different dopants (Pb:Bi, Te:I) in comparison to experimental results [experimental sources: As:Te and Ga:Si¹²; Te:I (squares)⁴²; Te:I (diamonds)⁴³; and Pb:Bi⁴⁴]. For PbTe, Bi doping strongly reduces the mobility compared to I doping, whereas the dopant effects on GaAs are comparatively smaller.

Figure 3. Defect-mediated electrical and thermoelectric transport. **a**, Correlation between ionic radius difference and the short-range electron-defect interaction matrix in half-Heusler materials with different dopants. Inset: relative sizes of the related atoms illustrated based on their ionic radius²³, with the charge of each shown at the top of each column. The electron-defect interaction matrix is calculated for the band edge state. **b-c**, Compiled experimental data for mobility dependence on the carrier concentration in representative half-Heusler materials with different dopants: **(b)** n-type ZrNiSn [experimental sources: ZrNiSn:V⁴⁵; ZrNiSn:Nb⁴⁶; Zr_{0.75}Hf_{0.25}NiSn:Nb⁴⁷; ZrNiSn:Ta (violet)⁴⁸; and ZrNiSn:Ta (green)⁴⁹], and **(c)** p-type NbFeSb [experimental sources: NbFeSb: Ti (blue)³⁰; NbFeSb:Ti (red)⁵⁰; NbFeSb:Zr and NbFeSb:Hf⁵¹]. **d-e**, Compiled highest mobility data from past studies with respect to the ionic radius difference between the dopant and host atoms for **(d)** n-type ZrNiSn and **(e)** p-type NbFeSb [experimental sources: ZrNiSn:V⁴⁵; ZrNiSn:Ta^{48,49}; ZrNiSn:Nb^{28,46}; NbFeSb:Ti^{30,31,50,52}; NbFeSb:Hf^{31,51}; NbFeSb:Zr^{31,51,52}]. The ionic radius differences are taken from **(a)**. The shaded regions indicate the standard deviation of these extracted mobility data. **f-g**, Comparisons between simulations and experimental results for the **(f)** electrical conductivity and **(g)** thermoelectric power factor of Ti-doped NbFeSb from 300 K to 1000 K. Consideration of Coulomb scatterings alone underestimates the power factor, while the consideration of full defect scattering with a partial cloaking effect leads to better agreement with the experiment. **(h)** Comparison between simulations and experimental results³¹ for the optimal room-temperature thermoelectric power factor in p-type NbFeSb with respect to the ionic radius difference between dopant and host atoms. The trend of power factor with respect to the ionic radius agrees between the experiments and the simulations.

Supplementary Information

Mobility enhancement in heavily doped semiconductors *via* electron cloaking

Jiawei Zhou¹, Hangtian Zhu², Qichen Song¹, Zhiwei Ding¹, Jun Mao², Zhifeng Ren², Gang
Chen¹

¹*Department of Mechanical Engineering, Massachusetts Institute of Technology, Cambridge,
MA 02139, USA*

²*Department of Physics and Texas Center for Superconductivity, University of Houston,
Houston, TX 77204, USA*

Supplementary Note

Model study of defect scattering

Figure 1 of the manuscript showed our electron transport calculation results for a model semiconductor, which illustrates the general effect of electron-defect interaction on electron transport, and demonstrates the possibility of achieving ideal electron cloaking (Fig. 1c,e,g). The electron mobility is calculated as¹

$$\mu_e = \left[\frac{N_v e}{3} \int v^2 \tau \left(-\frac{\partial f^0}{\partial E} \right) D(E) dE \right] / n \quad (1)$$

Here the integration spans over electron states close to the Fermi level, $N_v = 6$ is the band degeneracy and e is the electronic charge. E is the electron energy. v is the electron group velocity and is related to the conductivity effective mass $m_{eff,c}$ ($0.268 m_e$, where m_e is the free electron mass) via $v^2 = 2E/m_{eff,c}$. τ is the electron relaxation time. $f^0 = 1/(1 + \exp(\frac{E-\mu}{k_B T}))$ is the Fermi-Dirac distribution function with μ being the Fermi level. $D(E)$ is the electronic density of states, related to the density-of-states effective mass $m_{eff,DOS}$ ($0.33 m_e$) via $D = \left(\frac{2m_{eff,DOS}}{\hbar^2} \right)^{3/2} \frac{\sqrt{E}}{2\pi^2}$ with \hbar being the reduced Planck constant. n is the carrier concentration.

The electron relaxation time τ , the inverse of the scattering rate, is determined via Matthiessen's rule considering both intrinsic electron-phonon interactions and extrinsic electron-defect interactions: $1/\tau = 1/\tau_{e-ph} + 1/\tau_{e-d}$. The intrinsic electron-phonon interactions consider both acoustic phonon and optical phonon scatterings via corresponding deformation potentials¹:

$$\frac{1}{\tau_{e-ph,acoustic}} = \frac{\pi D_A^2 k_B T}{\hbar C_l} D(E) \quad (2)$$

where $D_A = 9.6$ eV is the acoustic deformation potential and $C_l = 190.7$ GPa is the elastic constant, and

$$\frac{1}{\tau_{e-ph,optical}} = (N_v - 1) \frac{\pi D_O^2}{2\rho\omega_O} \left[\frac{1}{e^{\frac{\hbar\omega_O}{k_B T}} - 1} D(E + \hbar\omega_O) + \left(\frac{1}{e^{\frac{\hbar\omega_O}{k_B T}} - 1} + 1 \right) D(E - \hbar\omega_O) \right] \quad (3)$$

where $D_O = 6.5$ eV/Å is the optical deformation potential, $\rho = 2330$ kg/m³ is the material density, and ω_O is the angular frequency of the optical phonon ($\omega_O = 2\pi f_O$, with $f_O = 472$ cm⁻¹). The sum of these two scattering rates leads to the intrinsic electron-phonon scattering rates: $1/\tau_{e-ph} = 1/\tau_{e-ph,acoustic} + 1/\tau_{e-ph,optical}$.

The electron-defect scattering rate $1/\tau_{e-d}$ is calculated based on the partial wave analysis which evaluates the scattering of electrons by a spherically symmetric potential²

$$\frac{1}{\tau_{e-d}} = N_d \frac{4\pi}{\hbar^2} \frac{1}{m_{eff,DOS} \sqrt{2m_{eff,DOS} E}} \sum_{l=0}^{\infty} (l+1) \sin^2(\delta_l - \delta_{l+1}) \quad (4)$$

where N_d is the volume density of the defect and δ_l is the phase shift of the electron wave with quantum number l . The maximum number of l in the summation is taken to be 10, which is found to be sufficient to achieve convergence. The scattering of electron waves is obtained by solving the Schrödinger equation. For spherically symmetric potential V , the wavefunction takes the form of $\Psi(\mathbf{r}) = \sum_{l,m} a_{l,m} \frac{u_l(r)}{r} Y_l^m(\theta, \phi)$, where $a_{l,m}$ are constants, and $Y_l^m(\theta, \phi)$ are the normalized spherical harmonics. The radial function $u_l(r)$ satisfies²

$$\frac{d^2 u_l}{dx^2} + \left[\frac{2ma_0^2 E}{\hbar^2} - \frac{2ma_0^2 V(r)}{\hbar^2} - \frac{l(l+1)}{x^2} \right] u_l = 0 \quad (5)$$

where the coordinate has been re-scaled by a characteristic length a_0 ($x = r/a_0$). Two forms of defect potential have been considered. The first corresponds to a pure Coulomb potential

$$\Delta \hat{V} = -\frac{e^2}{4\pi\epsilon\epsilon_0 r} \quad (6)$$

where $\epsilon = 11.7$ is the dielectric constant, and ϵ_0 is the vacuum permittivity. The second corresponds to a more practical case in which the central part is replaced with a central potential

$$\Delta \hat{V} = \begin{cases} V_0 & r \leq r_0 \\ -\frac{e^2}{4\pi\epsilon\epsilon_0 r} & r > r_0 \end{cases} \quad (7)$$

with $r_0 = 1.6 \text{ \AA}$ characterizing the range of the short-range potential, and where $V_0 = 9.7 \text{ eV}$ is the short-range potential energy. The solution to Eq. (5) is obtained by numerical integration based on the Gauss-Jackson method³. The phase shift is determined by matching the obtained solution to the expected asymptotic form of $u_l \approx x B_l \cos(\delta_l) \left[j_l \left(\sqrt{\frac{2ma_0^2 E}{\hbar^2}} x \right) - \tan(\delta_l) y_l \left(\sqrt{\frac{2ma_0^2 E}{\hbar^2}} x \right) \right]$, where $j_l(z)$ and $y_l(z)$ are the regular and irregular spherical Bessel function of order l , respectively. Once the phase shifts are determined, the electron-defect scattering rates can be readily computed using Eq. (4).

Extraction of central cell defect potential

The impurity potential is defined as the difference in the total electronic potential from first-principles calculations between the system with the defect and the original pristine system $\Delta \hat{V} = V_d - V_{bulk}$. One typically builds a large supercell and calculates the defect potential for the pure and defected systems separately. However, as the supercell size is often limited to be no more than a few lattice vectors long, the long-range Coulomb potential is cut off at the supercell boundary and cannot be correctly represented. The challenge of the electron-defect scattering calculation is thus to obtain a correct full profile of the defect potential including both its long-range and short-range parts.

While the central cell potential can vary significantly with the defect and lattice type, we recognize that the long-range portion of the defect potential can be well described by an analytic Coulomb potential profile. This fact has been utilized to correct the defect formation energy in finite size supercell calculations⁴. Here we use this fact to recover the short-range part of the defect potential, which essentially is the central cell potential. To illustrate this, we consider n-type silicon as an example. We built a silicon supercell with a $3 \times 3 \times 3$ conventional unit cell and replaced one silicon atom with a dopant atom (phosphorous, arsenic, or antimony). The defect potential is obtained by subtracting the potential of pure silicon from the one with the dopant atom. For the latter calculation, the total charge of the supercell is taken to be $+1e$, making the dopant positively charged (corresponding to n-type). The defect potentials corresponding to the different dopants in silicon are shown in Fig. S3a.

If the defect potential is short-ranged, the potential profile far away from the defect atom (corresponding to the middle point in the plot) should be flat. However, it is clear from Fig. S3a that while different dopants show different short-range profiles, all exhibit a gradually decaying profile far away from the defect. This gradually decaying profile is due to the long-range Coulomb potential of the defect charge. If we consider the Coulomb potential of an infinite periodic array of charge Ze at locations \mathbf{R}_i corresponding to the corners of the periodic supercell⁵, the Coulomb potential at \mathbf{r} is

$$\Delta\hat{V}_{lr} = -\sum_i \frac{Ze^2}{\sqrt{|\bar{\epsilon}|}} \frac{\text{erfc}\left(\gamma\sqrt{(\mathbf{R}_i-\mathbf{r})\cdot\bar{\epsilon}^{-1}\cdot(\mathbf{R}_i-\mathbf{r})}\right)}{\sqrt{(\mathbf{R}_i-\mathbf{r})\cdot\bar{\epsilon}^{-1}\cdot(\mathbf{R}_i-\mathbf{r})}} - \sum_{\mathbf{G}_i \neq 0} \frac{4\pi Ze^2}{\Omega} \frac{\exp\left(-\mathbf{G}_i\cdot\bar{\epsilon}\cdot\frac{\mathbf{G}_i}{4\gamma^2}\right)}{\mathbf{G}_i\cdot\bar{\epsilon}\cdot\mathbf{G}_i} \exp(i\mathbf{G}_i\cdot\mathbf{r}) + \frac{\pi Ze^2}{\Omega\gamma^2} \quad (8)$$

Here $\bar{\epsilon}$ is the dielectric tensor computed from first principles, Ω is the supercell volume, $Z = 1$ is the defect charge, and γ is a convergence parameter for the Ewald summation. This long-range Coulomb potential is also plotted in Fig. S3a as a reference and matches well with the asymptotic trend of all defect potentials extracted from first principles. If we subtract this long-range term from the defect potential, we are then left with only a short-range component (Fig. S3b), which becomes flat away from the defect. In Fig. S3b we have also aligned the potential at the farthest distance from the defect to zero, based on its short-range nature. This shows that the above procedure enabled the extraction of the short-range defect potential from the finite-size supercell calculations. The effects of long-range Coulomb potential on electron-defect interactions can then be later added into the scattering matrix *via* an analytic expression, as explained in detail in Methods.

First-principles electron transport calculation

First-principles calculations of electron transport properties (specifically, the electrical conductivity σ and the Seebeck coefficient S) are based on the Boltzmann transport theory¹:

$$\begin{cases} \sigma = \frac{e^2}{3\Omega_0 N_k} \sum_{\mathbf{k}\alpha} v_{\mathbf{k}\alpha}^2 \tau_{\mathbf{k}\alpha} \left(-\frac{\partial f_{\mathbf{k}\alpha}^0}{\partial E}\right) \\ S = \frac{e}{3\sigma\Omega_0 N_k T} \sum_{\mathbf{k}\alpha} (E - \mu) v_{\mathbf{k}\alpha}^2 \tau_{\mathbf{k}\alpha} \left(-\frac{\partial f_{\mathbf{k}\alpha}^0}{\partial E}\right) \end{cases} \quad (9)$$

where e is the electronic charge, Ω_0 is the unit cell volume, $N_{\mathbf{k}}$ is the number of \mathbf{k} points, α is the band index, $v_{\mathbf{k}\alpha}$ is the electron group velocity, $\tau_{\mathbf{k}\alpha}$ is the electron relaxation time, E is the electron energy, μ is the Fermi level, and $f_{\mathbf{k}\alpha}^0$ is the Fermi-Dirac distribution. The electron energy and group velocity are derived from the electronic band structure. The equilibrium properties of electrons of half-Heusler materials are calculated from first principles using the QUANTUM ESPRESSO software package⁶. We use the generalized gradient approximation (GGA) of Perdew, Burke and Ernzerhof with the Troullier-Martins-type norm-conserving semilocal pseudopotential⁷ (corresponding to pbe-mt.UPF in the QUANTUM ESPRESSO pseudopotential library). A cutoff energy of 120 Ryd and a $6 \times 6 \times 6$ \mathbf{k} -mesh are used to determine the equilibrium lattice constant. The equilibrium properties of phonons and the electron-phonon interaction matrices are calculated *via* density functional perturbation theory⁸ for a $6 \times 6 \times 6$ \mathbf{q} -mesh (with a $6 \times 6 \times 6$ \mathbf{k} -mesh for the electron-phonon interaction matrix). We then use the EPW software package⁹ to interpolate the electronic information and the phonon information, as well as the electron-phonon coupling matrices to a fine mesh. A fine mesh is required to ensure that the calculation of transport properties based on Eq. (9) is converged. In Eq. (9), the electron relaxation time is determined *via* Matthiessen's rule considering both intrinsic electron-phonon scattering rates and extrinsic electron-defect scattering rates: $1/\tau = 1/\tau_{e-ph} + 1/\tau_{e-d}$. The intrinsic electron-phonon scattering rates are related to the electron-phonon interaction matrix $g(\mathbf{k}, \mathbf{k} + \mathbf{q}, \mathbf{q})$ *via*¹⁰

$$\frac{1}{\tau_k^{e-ph}} = \frac{2\pi}{\hbar} \frac{1}{N_q} \sum_{\mathbf{q}} |g(\mathbf{k}, \mathbf{k} + \mathbf{q}, \mathbf{q})|^2 \cdot \left[\begin{array}{l} (n_q + f_{\mathbf{k}+\mathbf{q}}) \delta(E_{\mathbf{k}} - E_{\mathbf{k}+\mathbf{q}} + \hbar\omega_{\mathbf{q}}) \\ + (n_q + 1 - f_{\mathbf{k}+\mathbf{q}}) \delta(E_{\mathbf{k}} - E_{\mathbf{k}+\mathbf{q}} - \hbar\omega_{\mathbf{q}}) \end{array} \right] \quad (10)$$

which sums over all possible scattering processes that satisfy momentum and energy conservations using a tetrahedral integration method, where N_q is the number of \mathbf{q} points, n_q is the Bose-Einstein distribution for phonons, and the delta functions indicate the energy conservation. The electron-defect scattering rates are determined by the electron-defect interaction matrix $g_{e-d}(\mathbf{k}, \mathbf{k}')$ *via*

$$\frac{1}{\tau_k^{e-d}} = N_d \Omega_0 \frac{2\pi}{\hbar} \frac{1}{N_k} \sum_{\mathbf{k}'} \left(1 - \frac{\mathbf{v}_{\mathbf{k}} \cdot \mathbf{v}_{\mathbf{k}'}}{|\mathbf{v}_{\mathbf{k}}| |\mathbf{v}_{\mathbf{k}'}} \right) |g_{e-d}(\mathbf{k}, \mathbf{k}')|^2 \delta(E_{\mathbf{k}} - E_{\mathbf{k}'}) \quad (11)$$

where N_d is the volume density of defects. The calculation of $g_{e-d}(\mathbf{k}, \mathbf{k}')$ has been detailed in Methods. By adding all scattering rates together, we obtain the total electron scattering rates, which are then inserted into Eq. (9) to yield the electron transport properties.

Spatial projection of electron-defect interaction

While in the main text we mainly discussed the defect potential and its trend in the periodic table based on the ionic radius, the actual electron-defect interaction strength is governed by the electron-defect interaction matrix, which is a spatial product of the defect potential and the electronic wavefunctions

$$\langle \psi_{\mathbf{k}'} | \Delta \hat{V} | \psi_{\mathbf{k}} \rangle = \int d\mathbf{r} \psi_{\mathbf{k}'}^*(\mathbf{r}) \Delta \hat{V}(\mathbf{r}) \psi_{\mathbf{k}}(\mathbf{r}) \quad (12)$$

Usually, the electronic wavefunction can be approximately expressed by a linear combination of atomic orbitals on different atomic sites

$$\psi_{\mathbf{k}}(\mathbf{r}) = e^{i\mathbf{k}\cdot\mathbf{r}} \sum_{\alpha i} c_{\alpha i} \phi_{\alpha}^i(\mathbf{r}) \quad (13)$$

where ϕ_{α}^i denotes the i -th atomic orbital on atom α and has significant non-zero values only around atom α . With this, we have

$$\langle \psi_{\mathbf{k}'} | \Delta \hat{V} | \psi_{\mathbf{k}} \rangle = e^{i(\mathbf{k}-\mathbf{k}')\cdot\mathbf{r}} \sum_{\alpha i, \beta j} c_{\alpha i} c_{\beta j}^* \int d\mathbf{r} \phi_{\beta}^{j*}(\mathbf{r}) \Delta \hat{V}(\mathbf{r}) \phi_{\alpha}^i(\mathbf{r}) \quad (14)$$

We denote the defect site as γ . Because the defect potential $\Delta \hat{V}(\mathbf{r})$ is significant only around the defect site, it is then clear that in addition to the magnitude of the defect potential, the electron-defect interaction also depends on the projection of wavefunctions on given atomic sites, which are described by the pre-factors $c_{\alpha i} c_{\beta j}^*$. For the same defect potential, wavefunctions with larger projected density-of-states on the defect site (namely large non-zero values of $c_{\gamma i}$) will lead to stronger electron-defect interactions, and vice versa.

Supplementary Figure

Figure S1. Illustration of a propagating electron wave scattered by the perturbed electronic potential due to the presence of a charged p-type dopant. For a p-type dopant, the long-range Coulomb potential is repulsive to electrons (contrary to the n-type case). In this case, if the dopant atom has a smaller ionic radius than the host atom, the perturbation tends to create an attractive force for electrons, which counteracts the Coulomb potential (electron-cloaking scenario). On the other hand, if the dopant atom has a larger ionic radius, it tends to create a repulsive force, which then adds to the long-range Coulomb potential (electron-scattering scenario).

Figure S2. Defect potential extracted from first principles calculations. **a-b**, Directly extracted defect potentials for **(a)** Si-doped GaAs, and **(b)** Bi-doped PbTe, plotted from the location of the defect. In each case, the long-range part can be approximately described by the Coulomb potential. The distortions observed in the range between ~ 2 Å and ~ 7 Å are due to relaxation of atomic positions in the presence of the defect. Significant deviation from the Coulomb potential can be seen at short range (< 1 Å). The asymptotic trend given by the Coulomb field is recovered as one moves away from the defect. **c-d**, Planar averaged defect potentials for Si-doped GaAs **(c)** and Bi-doped PbTe **(d)**. The planar averaged potential is calculated by $\overline{\Delta V}(z) = \frac{1}{A} \iint_0^a dx dy \Delta \hat{V}(x, y, z)$ where a is the lattice vector length of the supercell, and A is the area of the x - y plane of the supercell, as indicated by the inset of **(c)**. The z axis starts from the plane containing the defect.

Figure S3. Illustration of subtraction of the long-range Coulomb potential from *ab initio* defect potential profiles. **a**, Uncorrected defect potentials for silicon doped with different n-type dopants (P, As, or Sb, as indicated in the figure legend) directly extracted from first-principles calculations, compared with the analytic long-range Coulomb potential calculated based on Eq. (8) in Supplementary Note. The three-dimensional defect potentials are projected as a function of the distance from the defect, located at the center of the supercell. The asymptotic trend given by the Coulomb field is recovered as one moves away from the defect. **The inset shows the short-range parts with a larger energy scale.** **b**, Corrected defect potentials, showing their short-range nature. Here, in calculating the defect potential, we have ignored the atomic relaxation to emphasize the long-range decaying part. The potentials at the farthest location away from the defect have also been aligned to zero.

Figure S4. Electron transport simulation of SrTiO₃. **a**, Computed phonon dispersion of SrTiO₃. Dashed lines are results from density functional perturbation theory, which gives rise to imaginary phonon frequencies. The solid lines are obtained by fitting force constants to a force-displacement dataset from *ab initio* molecular dynamics study¹¹ using VASP package. The fitting was performed using ALAMODE package¹². The latter method correctly reproduces the stable phonon modes at 300 K. **b**, Projected electronic density-of-states near the band edge, showing that electron states near the conduction band edge have dominant contributions from *d* orbitals on Sr atom. **c-d**, Defect potentials for (b) La dopant on Sr site, and (c) Nb dopant on Ti site. **e**, Comparison of electron-defect scattering rates for La and Nb dopants at the carrier concentration of $6 \times 10^{19} \text{ cm}^{-3}$. The defect potential includes both the short-range perturbation and long-range Coulomb potential. **f**, Mobilities considering different scattering conditions (Coulomb scattering only, and those that consider the full defect potential corresponding to La and Nb dopants), and comparison between simulation and experiment¹³. The calculated mobilities assume a carrier concentration of $6 \times 10^{19} \text{ cm}^{-3}$.

Figure S5. Calculated defect potentials for selected dopant/host pairs. Results are shown for n-type Ta-doped ZrNiSn (a), Nb-doped ZrNiSn (b), V-doped TiNiSn (c), Nb-doped TiNiSn (d), and p-type Sc-doped ZrCoSb (e), Y-doped ZrCoSb (f), Sc-doped TiCoSb (g), and Y-doped TiCoSb (h). The ionic sizes are drawn in the inset (same as Figure 3a). Here, in presenting the defect potential, we have ignored the atomic relaxation to focus on the short-range defect potential.

Figure S6. Illustrations of defect potentials in n-type and p-type materials leading to electron-scattering and electron-cloaking scenarios. **a**, The electron-cloaking effect is achieved in n-type material when the Coulomb potential is counterbalanced by a repulsive central cell potential from a dopant. **b-c**, Calculated defect potential for n-type **(b)** V-doped ZrNiSn (electron-scattering scenario) and **(c)** Nb-doped TiNiSn (electron-cloaking scenario). **d**, The electron-cloaking effect is achieved in p-type material when the Coulomb potential is counterbalanced by an attractive central cell potential from a dopant. **e-f**, Calculated defect potential for p-type **(e)** Sc-doped ZrCoSb (electron-cloaking scenario) and **(f)** Y-doped TiCoSb (electron-scattering scenario). Here, in presenting the defect potential, we have ignored the atomic relaxation to focus on the short-range defect potential. Coulomb potentials are also plotted in **(b-c)** and **(e-f)**, to show that the asymptotic trend given by the Coulomb field is recovered as one moves away from the defect.

Figure S7. Electron transport details in p-type NbFeSb. **a**, Scattering rates for holes in p-type Ti-doped NbFeSb, showing that the short-range defect potential leads to a significant reduction in the scattering rates compared to those limited by Coulomb scattering. The energy is relative to the valence band edge. **b**, Charge carrier mean free paths as a function of energy in Ti-doped NbFeSb at a carrier concentration of $2 \times 10^{20} \text{ cm}^{-3}$. While Coulomb scatterings severely limit the mean free paths, the central cell potential can counteract this effect and increase the carrier mean free paths by almost a factor of two near the band edge. **c**, Calculated defect potential of p-type dopants (Ti, Zr, and Hf) in NbFeSb. The larger power factors obtained with Ti and Hf dopants can be understood based on their attractive short-range potential which counterbalances the long-range Coulomb scatterings.

Figure S8. Thermoelectric power factor of p-type Ti-doped TaFeSb at room temperature. Both curves consider intrinsic electron-phonon interactions. The dashed curve includes electron scatterings by the Coulomb potential, while the solid curve considers electron scatterings by the full defect potential. Due to the counteraction between the strong central cell potential of the Ti dopant and the Coulomb potential, the full defect potential leads to an enhanced optimal power factor.

Figure S9. Thermoelectric power factor at 150 K. a, p-type Ti-doped NbFeSb. **b,** p-type Ti-doped TaFeSb. For both plots, the red curve only considers electron scatterings by phonons and by the Coulomb potential, while the blue curve considers electron scatterings by phonons and the full defect potential (including the short-range part). Due to the counteraction between the strong central cell potential of the Ti dopant and the Coulomb potential, a larger enhancement in power factor is observed at lower temperatures.

Table S1. Dielectric constants of representative oxides and thermoelectric materials.

Material	Dielectric constant (relative)
ZrO ₂	29 ¹⁴
HfO ₂	25 ¹⁴
Ta ₂ O ₅	26 ¹⁴
La ₂ O ₃	30 ¹⁴
LaAlO ₃	30 ¹⁴
Nb ₂ O ₅	35 ¹⁴
TiO ₂	95 ¹⁴
Bi ₂ Te ₃	290 (, 15K) ¹⁵
PbTe	414 ¹⁵
SnSe	42 (c axis), 45 (a axis) ¹⁵
NbFeSb	45 ¹⁶
Mg ₃ Sb ₂	32 ¹⁶

Reference

1. Lundstrom, M. *Fundamentals of Carrier Transport*. (Cambridge University Press, 2009).
2. Cohen-Tannoudji, C., Diu, B. & Laloe, F. *Quantum Mechanics*. (Wiley, 1992).
3. Fetterman, W., Osborne, E. & Saxon, D. S. A numerical solution of Schrodinger's equation in the continuum. *Journal of Research of the National Bureau of Standards* **52**, (1954).
4. Freysoldt, C. *et al.* First-principles calculations for point defects in solids. *Rev. Mod. Phys.* **86**, 253–305 (2014).
5. Kumagai, Y. & Oba, F. Electrostatics-based finite-size corrections for first-principles point defect calculations. *Phys. Rev. B* **89**, 195205 (2014).
6. Giannozzi, P. *et al.* QUANTUM ESPRESSO: a modular and open-source software project for quantum simulations of materials. *J. Phys.: Condens. Matter* **21**, 395502 (2009).
7. Perdew, J. P., Burke, K. & Ernzerhof, M. Generalized gradient approximation made simple. *Phys. Rev. Lett.* **77**, 3865–3868 (1996).
8. Baroni, S., de Gironcoli, S., Dal Corso, A. & Giannozzi, P. Phonons and related crystal properties from density-functional perturbation theory. *Rev. Mod. Phys.* **73**, 515–562 (2001).
9. Giustino, F., Cohen, M. & Louie, S. Electron-phonon interaction using Wannier functions. *Phys. Rev. B* **76**, 165108 (2007).
10. J. M. Ziman. *Electrons and Phonons: The Theory of Transport Phenomena in Solids*. (Clarendon Press, 1960).
11. Hellman, O., Steneteg, P., Abrikosov, I. A. & Simak, S. I. Temperature dependent effective potential method for accurate free energy calculations of solids. *Phys. Rev. B* **87**, 104111 (2013).
12. Tadano, T., Gohda, Y. & Tsuneyuki, S. Anharmonic force constants extracted from first-principles molecular dynamics: applications to heat transfer simulations. *J. Phys.: Condens. Matter* **26**, 225402 (2014).
13. Han, W. *et al.* Spin injection and detection in lanthanum- and niobium-doped SrTiO₃ using the Hanle technique. *Nat. Comm.* **4**, 2134 (2013).
14. Azadmanjiri, J. *et al.* A review on hybrid nanolaminate materials synthesized by deposition techniques for energy storage applications. *J. Mater. Chem. A* **2**, 3695–3708 (2014).
15. *Non-Tetrahedrally Bonded Elements and Binary Compounds I*. vol. 41C (Springer-Verlag, 1998).
16. J. Slade, T. *et al.* Understanding the thermally activated charge transport in NaPb_mSbQ_{m+2} (Q = S, Se, Te) thermoelectrics: weak dielectric screening leads to grain boundary dominated charge carrier scattering. *Energy & Environmental Science* **13**, 1509–1518 (2020).

REVIEWERS' COMMENTS

Reviewer #1 (Remarks to the Author):

I have no further comments or queries for the authors. Please publish.

Revision report for MS# NCOMMS-21-20659B

Mobility enhancement in heavily doped semiconductors via electron cloaking

Jiawei Zhou, Hangtian Zhu, Qichen Song, Zhiwei Ding, Jun Mao, Zhifeng Ren, Gang Chen

We thank the reviewer for his/her positive and constructive comments on the manuscript. In this revised version we have mainly changed the manuscript to meet the format requirements. All the changes are tracked.

Reviewer #1:

I have no further comments or queries for the authors. Please publish.

Response: We thank the reviewer for his/her positive and constructive comments.